# Towards Minimax Optimality
# of Model-based Robust Reinforcement Learning

**Pierre Clavier** [1,2,3]       **Erwan Le Pennec**[1]       **Matthieu Geist**[4]

[1]CMAP, CNRS, Ecole Polytechnique, Institut Polytechnique de Paris, 91120 Palaiseau, France,
[2]INRIA Paris, HeKA, France,
[3]Centre de Recherche des Cordeliers, INSERM, Universite de Paris, Sorbonne Universite, F-75006 Paris, France,
[4]Cohere.

## Abstract

We study the sample complexity of obtaining an $\epsilon$-optimal policy in *Robust* discounted Markov Decision Processes (RMDPs), given only access to a generative model of the nominal kernel. This problem is widely studied in the non-robust case, and it is known that any planning approach applied to an empirical MDP estimated with $\tilde{\mathcal{O}}(\frac{H^3|S||A|}{\epsilon^2})$ samples provides an $\epsilon$-optimal policy, which is minimax optimal. Results in the robust case are much more scarce. For $sa$- (resp $s$-) rectangular uncertainty sets, until recently the best-known sample complexity was $\tilde{\mathcal{O}}(\frac{H^4|S|^2|A|}{\epsilon^2})$ (resp. $\tilde{\mathcal{O}}(\frac{H^4|S|^2|A|^2}{\epsilon^2})$), for specific algorithms and when the uncertainty set is based on the total variation (TV), the KL or the Chi-square divergences. In this paper, we consider uncertainty sets defined with an $L_p$-ball (recovering the TV case), and study the sample complexity of *any* planning algorithm (with high accuracy guarantee on the solution) applied to an empirical RMDP estimated using the generative model. In the general case, we prove a sample complexity of $\tilde{\mathcal{O}}(\frac{H^4|S||A|}{\epsilon^2})$ for both the $sa$- and $s$-rectangular cases (improvements of $|S|$ and $|S||A|$ respectively). When the size of the uncertainty is small enough, we improve the sample complexity to $\tilde{\mathcal{O}}(\frac{H^3|S||A|}{\epsilon^2})$, recovering the lower-bound for the non-robust case for the first time and a robust lower-bound. Finally, we also introduce simple and efficient algorithms for solving the studied $L_p$ robust MDPs.

## 1 INTRODUCTION

Reinforcement learning (RL) [Sutton and Barto, 2018], often modelled as learning and decision-making in a Markov decision process (MDP), has attracted increasing interest in recent years due to its remarkable success in practice. A major goal of RL is to find a strategy or policy, based on a collection of data samples, that can predict the expected cumulative rewards in an MDP, without direct access to a detailed description of the underlying model. However, Mannor et al. [2004] showed that the policy and the value function could sometimes be sensitive to estimation errors of the reward and transition probabilities, meaning that a very small perturbation of the reward and transition probabilities could lead to a significant change in the value function.

Robust MDPs [Iyengar, 2005, Nilim and El Ghaoui, 2005] (RMDPs) have been proposed to handle these problems by letting the transition probability vary in an uncertainty (or ambiguity) set. In this way, the solution of robust MDPs is less sensitive to model estimation errors with a properly chosen uncertainty set. An RMDP problem is usually formulated as a max-min problem, where the objective is to find the policy that maximizes the value function for the worst possible model that lies within an uncertainty set around a nominal model. Initially, RMPDs [Iyengar, 2005, Nilim and El Ghaoui, 2005] were developed because the solution of MDPs can be very sensitive to the model parameters [Zhao et al., 2019, Packer et al., 2018]. However, as the solution of robust MDPs is NP-hard for general uncertainty sets Nilim and El Ghaoui [2005], the uncertainty set is usually assumed to be rectangular (meaning that it can be decomposed as a product of uncertainty sets for each state or state-action pair), which allows tractability Iyengar [2005], Ho et al. [2021]. These two kinds of sets are called respectively $s$- and $sa$-rectangular sets. A fundamental difference between them is that the greedy and optimal policy in $sa$-rectangular robust MDPs is deterministic, as in non-robust MDPs, but can be stochastic in the $s$-rectangular case Wiesemann et al. [2013]. Compared to $sa$-rectangular robust MDPs, $s$-rectangular robust MDPs are less restrictive but much more difficult to handle. Under this rectangularity assumption, many structural properties of MDPs remain intact Iyengar [2005] and methods such as robust value it-

eration, robust modified policy iteration, or partial robust policy iteration Ho et al. [2021] can be used to solve them. It is also known that the uncertainty in the reward can be easily handled, while handling uncertainty in the transition kernel is much more difficult Kumar et al. [2022], Derman et al. [2021]. Finally, Deep Robust RL algorithms Pinto et al. [2017], Clavier et al. [2022], Tanabe et al. [2022] have been proposed to tackle the problem of Robust MDPS with continuous state-action space.

In this work, we consider robust MDPs, with both $sa$- and $s$-rectangular uncertainty sets, consisting of $L_p$-balls centered around the nominal model $P_0$. We assume access to a generative model, which can sample a next state from any state-action pair from the nominal model. The question we address is to know how many samples are required to compute an $\epsilon$-optimal policy. This classic abstraction, which allows studying the sample complexity of planning over a long horizon, is widely studied in the non-robust setting Singh and Yee [1994], Sidford et al. [2018], Azar et al. [2013], Agarwal et al. [2020], Li et al. [2020], Kozuno et al. [2022], but much less in the robust setting [Yang et al., 2021, Panaganti and Kalathil, 2022, Shi and Chi, 2022, Xu et al., 2023, Shi et al., 2023]. We consider more specifically model-based robust RL. We call the generative model the same number of times for each state-action pair, to build a maximum likelihood estimate of the nominal model, and use any planning algorithm for robust MDPs (with high accuracy guarantee on the solution) on this empirical model. This setting will be discussed further later, but we insist right away that it is especially meaningful in the robust setting, as it is a good abstraction of sim2real. The research question we address is:

*How many samples are required for guaranteeing an $\epsilon$-optimal policy with high probability?*

Our **first contribution** is to prove that for both $s$ and $sa$-rectangular sets based on $L_p$-balls, the sample complexity the proposed approach is $\tilde{\mathcal{O}}(\frac{H^4 |S||A|}{\epsilon^2})$, with $H = (1-\gamma)^{-1}$ being the horizon term. Previous works [Yang et al., 2021, Panaganti and Kalathil, 2022, Shi and Chi, 2022, Xu et al., 2023] study different sets, based on the Kullback-Leibler (KL) divergence, Chi-square divergence, and total variation (TV). We have the TV in common ($L_1$-ball up to a normalizing factor), and, in this case, we improve these existing results by $|S|$ for the $sa$-rectangular case, and by $|S||A|$ for the $s$-rectangular case, which is significant for large state-action spaces. On the technical side, our results build heavily upon the dual view of robust Bellman operators [Derman et al., 2021, Kumar et al., 2022]. However, we deviate from this line of work by enforcing the uncertainty set to belong to the simplex. This allows ensuring that the robust operators are overly conservative while ensuring they are $\gamma$-contractions, which is important for the theoretical analysis. On the negative side, the algorithms they introduce are no longer applicable, which calls for new algorithmic design.

Our **second contribution** is to show that, if the uncertainty set is small enough, then we have a sample complexity of $\tilde{\mathcal{O}}(\frac{H^3 |S||A|}{\epsilon^2})$. This is a further improvement by $H$ of the previous bound, and it matches the known lower bound for the non-robust case [Azar et al., 2013]. On the technical side, it again builds upon the dual view of robust Bellman operators with the deviation mentioned above.[Derman et al., 2021, Kumar et al., 2022]. In addition to that, it adapts two proof techniques of the non-robust case: The total variance technique of Azar et al. [2013] to reduce the dependency to the horizon, and the *absorbing MDP* construction of Agarwal et al. [2020] to allow for a wider range of valid $\epsilon$.As mentioned earlier,[Derman et al., 2021, Kumar et al., 2022] algorithms are not applicable to the more realistic uncertainty sets we consider.

Our **third contribution** is an algorithm DRVI L$_P$ (see Alg. 1, for Distributionally Robust Value Iteration for $L_P$ in $sa$rectangular case that solves exactly RMDPs in the case of valid robust transition that belongs to the simplex contrary to Kumar et al. [2022].

## 2 RELATED WORK

The question of sample complexity when having access to a generative model has been widely studied in the non-robust setting Singh and Yee [1994], Sidford et al. [2018], Azar et al. [2013], Agarwal et al. [2020], Li et al. [2020], Kozuno et al. [2022]. Notably, Azar et al. [2013] provide a lower-bound of this sample complexity, $\tilde{\Omega}(\frac{|S||A|H^3}{\epsilon^2})$, and show that (tabular) model-based RL reaches this lower-bound, making it minimax optimal (up to polylog factors). This bound relies on the so-called total variance technique, that we adapt to the robust setting. However, their result is only true for small enough $\epsilon$, in the range $(0, \sqrt{H/|S|})$. This was later improved to $(0, \sqrt{H})$ by Agarwal et al. [2020], thanks to a novel *absorbing MDP* construction, that we also adapt to the robust setting.

Closer to our contributions are the works that study the sample complexity in the *robust* setting Yang et al. [2021], Panaganti and Kalathil [2022], Xu et al. [2023], Shi and Chi [2022]. The study of sample complexity of specific algorithms (respectively either empirical robust value or Robust Phased Value Learning) is studied by Panaganti and Kalathil [2022], Xu et al. [2023], while our results apply to any oracle planning (applied to the empirical model), as long as it provides a solution with enough accuracy. We consider both $s$- and $sa$-rectangular uncertainty sets, as Yang et al. [2021], while Panaganti and Kalathil [2022], Xu et al. [2023], Shi and Chi [2022] only consider the simpler $sa$-rectangular sets. They all study either TV, KL or Chi-square balls, while we study $L_p$-balls. Shi and Chi [2022] improved the KL bound compared to Yang et al. [2021], Panaganti and Kalathil [2022] in the $sa$ rectangular case.

The framework of Xu et al. [2023] is slightly different as they consider finite horizon which adds a factor $H$ in all bounds. All previous results are not minimax optimal in terms of the horizon factor.

We rely more specifically on a simple optimization dual expression of the minimization problem over models. As such, we do not cover the KL and Chi-square cases, which do not have such a simple form even if there can also be written as simple scalar optimization problem. However, we have in common with Yang et al. [2021], Panaganti and Kalathil [2022] the total variation case, which corresponds to a (scaled) $L_1$-ball. For this case, we can compare our sample complexities. Without assumption on the size of the uncertainty set, we improve the existing sample complexities by $|S|$ and $|S||A|$ respectively (for $sa$- or $s$-rectangularity). Also, our bounds have no dependency on the size of the uncertainty set. Notice that as we consider a generic oracle planning algorithm, our bounds apply to the algorithms they consider in Panaganti and Kalathil [2022], Xu et al. [2023]. If we further assume that the uncertainty set is small enough, then we improve the bound by an additional $H$ factor, reaching the minimax sample complexity of the non-robust case. Table 1 summarizes the difference in sample complexity, and we'll discuss them again after stating our theorems.

Finally, the archival version of this contribution predates the concurrent work of Shi et al. [2023] that studies the sample complexity of RMDPs for $TV$ and $\chi^2$ divergence. In the very specific case of $sa$- rectangular for $TV$ which in this case coincides with $L_1$ norm, Shi et al. [2023] retrieves our upper bound which is minimax optimal in the regime where the radius of the uncertainty set is small and improves our result in the regime where the radius of the uncertainty set is bigger than $1 - \gamma$. However, our results hold more generally for the $s$-rectangular case are still state-of-the-art for $s$-rectangular case with $p \geq 1$ and for $sa-$rectangular with $p > 1$. Notice also that the proof techniques are very different, and it is an interesting research direction to know if their bound for the regime where the radius of the uncertainty set is bigger than $1 - \gamma$ or their lower-bound would extend to the more general case studied here.

# 3  PRELIMINARIES

For finite sets $S$ and $A$, we write respectively $|S|$ and $|A|$ their cardinality. We write $\Delta_A := \{p : A \to \mathbb{R} \mid p(a) \geq 0, \sum_{a \in A} p(a) = 1\}$ the simplex over $A$. For $v \in \mathbb{R}^S$ the classic $L_q$ norm is $\|v\|_q^q = \sum_s v(s)^q$. The unitary vector of dimension $|S|$ is denoted $1_S$. Finally, we denote $\tilde{\mathcal{O}}$ the $\mathcal{O}$ notation up to logarithm factor.

## 3.1  MARKOV DECISION PROCESS

A Markov Decision Process (MDP) is defined by $M = (\mathcal{S}, \mathcal{A}, P, R, \gamma, \mu)$ where $S$ and $A$ are the finite state and action spaces, $P : \mathcal{S} \times \mathcal{A} \to \Delta_\mathcal{S}$ is the transition kernel, $R : \mathcal{S} \times \mathcal{A} \to [0, 1]$ is the reward function, $\mu \in \Delta_\mathcal{S}$ is the initial distribution over states and $\gamma \in [0, 1)$ is the discount factor. A stationary policy $\pi : \mathcal{S} \to \Delta_\mathcal{A}$ maps states to probability distributions over actions. We write $P_{s,a}$ the vector $P(\cdot|s, a)$. We also define $P^\pi$ to be the transition matrix on state-action pairs induced by a policy $\pi$: $P^\pi_{(s,a),(s',a')} = P(s' \mid s, a)\pi(a' \mid s')$. Slightly abusing notations, for $V \in \mathbb{R}^S$, we define the vector $\mathrm{Var}_P(V) \in \mathbb{R}^{\mathcal{S} \times A}$ as $\mathrm{Var}_P(V)(s, a) := \mathrm{Var}_{P(\cdot|s,a)}(V)$, so that $\mathrm{Var}_P(V) = P(V)^2 - (PV)^2$ (with the square understood component-wise). Usually, the goal is to estimate the value function defined as:

$$V^\pi_{P,R}(s) := \mathbb{E}\left[\sum_{n=0}^\infty \gamma^n R(s_n, a_n) \mid s_0 = s, \pi, P\right].$$

The value function $V^\pi_{P,R}$ for policy $\pi$, is the fixed point of the Bellmen operator $\mathcal{T}^\pi_{P,R}$, defined as

$$\mathcal{T}^\pi_{P,R} V(s) = \sum_a \pi(a|s)[R(s, a) + \gamma \sum_{s'} P(s'|s, a) V(s')].$$

We also define the optimal Bellman operator: $\mathcal{T}^*_{P,R} V(s) = \max_{\pi_s \in \Delta_\mathcal{A}} \left(\mathcal{T}^{\pi_s}_{P,R} V\right)(s)$. Both optimal and classical Bellman operators are $\gamma$-contractions Sutton and Barto [2018]. This is why sequences $\{V^\pi_n \mid n \geq 0\}$, and $\{V^*_n \mid n \geq 0\}$, defined as

$$V^\pi_{n+1} := \mathcal{T}^\pi_{P,R} V^\pi_n \quad \text{and} \quad V^*_{n+1} := \mathcal{T}^*_{P,R} V^*_n,$$

converge linearly to $V^\pi_{P,R}$ and $V^*_{P,R}$, respectively the value function following $\pi$ and the optimal value function. Finally, we can define the Q-function,

$$Q^\pi_{P,R}(s, a) := \mathbb{E}\left[\sum_{n=0}^\infty \gamma^n R(s_n, a_n) \mid s_0 = s, a_0 = a, \pi, P\right].$$

The value function and Q-function are linked with the relation $V^\pi_{P,R}(s) = \langle (\pi_s, Q^\pi_{P,R}(s) \rangle_A$. With these notations, we can define Q-functions for transition probability transition $P$ following policy $\pi$ such as

$$Q^\pi_{P,R} = R + \gamma P V^\pi_{P,R} = R + \gamma P^\pi Q^\pi_{P,R} = (I - \gamma P^\pi)^{-1} R.$$

## 3.2  ROBUST MARKOV DECISION PROCESS

Once classical MDPs defined, we can define robust (optimal) Bellman operators $\mathcal{T}^\pi_\mathcal{U}$ and $\mathcal{T}^*_\mathcal{U}$,

$$\mathcal{T}^\pi_\mathcal{U}(s) := \min_{R, P \in \mathcal{U}} \left(\mathcal{T}^\pi_{P,R} V\right)(s)$$

Table 1: Sample Complexity of TV for $s$- or $sa$ rectangular with $\beta$ (see Def 3.2) the radius of uncertainty set (see also Tab. 2 in the appendix for a complete table with different norms)

| | Panaganti and Kalathil [2022] | Yang et al. [2021] | Our $\beta \geq 0$ | Our $1/(2H\gamma) > \beta > 0$ | Shi et al. [2023] |
|---|---|---|---|---|---|
| $sa$-rect. | $\tilde{\mathcal{O}}\left(\frac{|S|^2|A|H^4}{\epsilon^2}\right)$ | $\tilde{\mathcal{O}}\left(\frac{|S|^2|A|H^4(2+\beta)^2}{\epsilon^2\beta^2}\right)$ | $\tilde{\mathcal{O}}\left(\frac{|S||A|H^4}{\epsilon^2}\right)$ | $\tilde{\mathcal{O}}\left(\frac{|S||A|H^3}{\epsilon^2}\right)$ | $\tilde{\mathcal{O}}\left(\frac{|S||A|H^2}{\epsilon^2\min(1/H,\beta)}\right)$ |
| $s$-rect. | $\times$ | $\tilde{\mathcal{O}}\left(\frac{|S|^2|A|^2H^4(2+\beta)^2}{\epsilon^2\beta^2}\right)$ | $\tilde{\mathcal{O}}\left(\frac{|S||A|H^4}{\epsilon^2}\right)$ | $\tilde{\mathcal{O}}\left(\frac{|S||A|H^3}{\epsilon^2}\right)$ | $\times$ |

$$\left(\mathcal{T}_{\mathcal{U}}^* V\right)(s) := \max_{\pi_s \in \Delta_{\mathcal{A}}} \min_{R,P \in \mathcal{U}} \left(\mathcal{T}_{P,R}^{\pi_s} V\right)(s),$$

where $P$ and $R$ belong to the uncertainty set $\mathcal{U}$. The optimal robust Bellman operator $\mathcal{T}_{\mathcal{U}}^*$ and robust Bellman operator $\mathcal{T}_{\mathcal{U}}^\pi$ are $\gamma$-contraction maps for any policy $\pi$ [Iyengar, 2005, Thm. 3.2] if the adversarial kernel $P \in \Delta_s$ to obtain a valid transition kernel :

$$\|\mathcal{T}_{\mathcal{U}}^* v - \mathcal{T}_{\mathcal{U}}^* u\|_\infty \leq \gamma \|u - v\|_\infty,$$
$$\|\mathcal{T}_{\mathcal{U}}^\pi v - \mathcal{T}_{\mathcal{U}}^\pi u\|_\infty \leq \gamma \|u - v\|_\infty, \quad \forall \pi.$$

Finally, for any initial values $V_0^\pi, V_0^*$, sequences defined as $V_{n+1}^\pi := \mathcal{T}_{\mathcal{U}}^\pi V_n^\pi$ and $V_{n+1}^* := \mathcal{T}_{\mathcal{U}}^* V_n^*$ converge linearly to their respective fixed points, that is $V_n^\pi \to V_{\mathcal{U}}^\pi$ and $V_n^* \to V_{\mathcal{U}}^*$. This makes robust value iteration an attractive method for solving robust MDPs. In order to obtain tractable forms of RMDPs, one has to make assumptions about the uncertainty sets and give them a rectangularity structure Iyengar [2005]. In the following, we will use an $L_p$ norm as the distance between distributions. The $s$- and $sa$-rectangular assumptions can be defined as follows, with $R_0$ and $P_0$ being called the nominal reward and kernel.

**Assumption 3.1.** *($sa$-rectangularity) We define $sa$-rectangular $L_p$-constrained uncertainty set as*

$$\mathcal{U}_p^{sa} := (R_0 + \mathcal{R}) \times (P_0 + \mathcal{P}), \mathcal{R} = \times_{s \in \mathcal{S}, a \in \mathcal{A}} \mathcal{R}_{s,a},$$
$$\mathcal{P} = \times_{s \in \mathcal{S}, a \in \mathcal{A}} \mathcal{P}_{s,a}, \mathcal{R}_{s,a} = \{r_{s,a} \in \mathbb{R} \mid |r_{s,a}| \leq \alpha_{s,a}\}$$
$$\mathcal{P}_{s,a} = \{P_{s,a} : \mathcal{S} \to \mathbb{R} \mid \sum_{s'} P_{s,a}(s') = 0,$$
$$P_{0,s,a} + P_{s,a} \geq 0, \|P_{s,a}\|_p \leq \beta_{s,a}\}$$

**Assumption 3.2.** *($s$-rectangularity) We define $s$-rectangular $L_p$-constrained uncertainty set as*

$$\mathcal{U}_p^{\mathbf{s}} = (R_0 + \mathcal{R}) \times (P_0 + \mathcal{P}), \mathcal{P} = \times_{s \in \mathcal{S}} \mathcal{P}_s,$$
$$\mathcal{R} = \times_{s \in \mathcal{S}} \mathcal{R}_s, \quad \mathcal{R}_s = \left\{r_s : \mathcal{A} \to \mathbb{R} \mid \|r_s\|_p \leq \alpha_s\right\}$$
$$\mathcal{P}_s = \{P_s : \mathcal{S} \times \mathcal{A} \to \mathbb{R} \mid \sum_{s'} P_s(s', a) = 0,$$
$$\forall a \in A, P_s(.,a) + P_{0,s} \geq 0, \|P_s\|_p \leq \beta_s\}$$

We write $\beta = \sup_{s,a} \beta_{s,a}$ for $sa$-rectangular assumptions or $\beta = \sup_s \beta_s$ for $s$-rectangular assumptions and with the same manner $\alpha = \sup_{s,a} \alpha_{s,a}$. Moreover, we write

$P \in \mathcal{P}_{0,s,a}$ for $P = P_{0,s,a} + P'$ with $P' \in \mathcal{P}_{s,a}$ and $P \in \mathcal{P}_{0,s}$ for $P = P_{0,s}^\pi + P'$ with $P' \in \mathcal{P}_s$, $P_{0,s}^\pi(s') = \sum_a \pi(a|s) P_{0,s,a}(s') \in \mathbb{R}^S$.

In comparison to $sa$-rectangular robust MDPs, $s$-rectangular robust MDPs are less restrictive but much more difficult to deal with. Using rectangular assumptions and constraints defined with $L_p$-balls, it is possible to derive simple dual forms for the (optimal) robust Bellman operators for the minimization problem that involves the seminorm defined below:

**Definition 3.1** (Span seminorm [Puterman, 1990])**.** *Let $q$ be such that it satisfies the Holder's equality, i.e. $\frac{1}{p} + \frac{1}{q} = 1$. Let $q$-variance or span-seminorm function $\mathrm{sp}_q(.) : \mathcal{S} \to \mathbb{R}$ and $q$-mean function $\omega_q : \mathcal{S} \to \mathbb{R}$ be defined as*

$$\mathrm{sp}_q(v) := \min_{\omega \in \mathbb{R}} \|v - \omega \mathbf{1}\|_q, \quad \omega_q(v) := \arg\min_{\omega \in \mathbb{R}} \|v - \omega \mathbf{1}\|_q.$$

One can think of those span-seminorms as semi-mean-centered-norms. The main problem is that these quantities represent the dispersion of a distribution around its mean, and there are no order relations for this type of object. Seminorms appear in the (non-robust) RL community for other reasons Puterman [1990], Scherrer [2013]. For $p =1$, 2 and $\infty$, a closed form can be derived, corresponding to median, variance and range. This is not the case for arbitrary $p$ but span-seminorms can be efficiently computed in practice, see Kumar et al. [2022]. Once span-seminorms defined, we introduced the dual of the inner minimization problem.

**Lemma 3.3** (Duality for $sa$ rectangular case with $L_p$ norm)**.** *For any $V \in \mathbb{R}^S, P_{0,s,a} = P_0(.|s,a) \in \mathbb{R}^S$ and $\mu \in \mathbb{R}^S$*

$$\min_{P \in \mathcal{P}_{0,s,a}} PV = \max_{\mu \geq 0} P_{0,s,a}(V - \mu) - \beta_{s,a} \mathrm{sp}_q(V - \mu)$$

**Lemma 3.4** (Duality for $s$ rectangular case.)**.** *Consider the probability kernel $P_{0,s}^\pi = \Pi^\pi P_{0,s,a} \in \mathbb{R}^s$ with $\Pi^\pi$ a projection matrix associated with a given policy $\pi$ such that $P_{0,s}^\pi(s') = \sum_a \pi(a|s) P_{0,s,a}(s') \in \mathbb{R}^S$. For any $V \in \mathbb{R}^S$ :*

$$\min_{P \in \mathcal{P}_{0,s}} PV = \max_{\mu \geq 0} P_{0,s}^\pi(V - \mu) - \beta_s \|\pi_s\|_q \mathrm{sp}_q(V - \mu)$$

Proofs car be found in Appendix B.5 ,3.4. These results allow computing robust value and Q-functions. Close to our work, Derman et al. [2021], Kumar et al. [2022] do not assume that robust kernel belongs to the simplex and in that sense, their formulation is a relaxation of the framework of RMPDs. Using this relaxation, closed form of robust Bellman operator can be obtained, see Th. 1 in Kumar et al. [2022]. In our work, we assume a valid transition kernel in the simplex ($P_{s,a} \geq 0$ or $P_s \geq 0$ for respectively $sa-$ or $s-$ rectangular case.) that leads to dual form that has not a closed form but which is a simple scalar optimization problem. A complete discussion can be found in Appendix A.2.

Finally, we denote robust $Q$ function for $sa-$ and $s-$ rectangular respectively $Q_{sa}^\pi$ and $Q_s^\pi$ and we define them from robust value function $V_{sa}^\pi$, $V_s^\pi$ as :

$$V_s^\pi(s) = \sum_a \pi(a|s)Q_s^\pi(s,a), V_{sa}^\pi(s)$$
$$= \sum_a \pi(a|s)Q_{sa}^\pi(s,a)$$

**Lemma 3.5.** *For $sa-$ and $s-$ rectangular,*

$$Q_{sa}^\pi(s,a) = r_{Q_{sa}^\pi}^{(s,a)} + \gamma P_{0,s,a}V_{sa}^\pi,$$
$$Q_s^\pi(s,a) = r_{Q_s^\pi}^s + \gamma P_{0,s,a}V_s^\pi$$

*with*

$$r_{Q_{sa}^\pi}^{(s,a)} = R_0(s,a) - \alpha_{s,a} + \gamma \min_{P \in \mathcal{P}_{s,a}} PV_{sa}^\pi$$

$$r_{Q_s^\pi}^s = R_0(s,a) - \left(\frac{\pi_s(a)}{\|\pi_s\|_q}\right)^{q-1}\alpha_s + \gamma \min_{P^\pi \in \mathcal{P}_s} P^\pi V_s^\pi$$

Robust $Q$ functions and dual forms of the robust Bellman operators will be central to our analysis of the sample complexity of model-based robust RL. They allow improving the bound by a factor $|S|$ or $|S||A|$ compared to existing results (Sec. 4). With additional technical subtleties, adapted from the non-robust setting, and assuming the uncertainty set is small enough, they even allow improving the bound by a factor $|S|H$ or $|S||A|H$ (Sec. 5).

## 3.3 GENERATIVE MODEL FRAMEWORK

We consider the setting where we have access to a generative model, or sampler, that gives us samples $s' \sim P_0(\cdot \mid s, a)$, from the nominal model and from arbitrary state-action couples. Suppose we call our sampler $N$ times on each state-action pair $(s, a)$. Let $\widehat{P}$ be our empirical model, the maximum likelihood estimate of $P_0$,

$$\widehat{P}(s' \mid s, a) = P_{s,a}(s') = \frac{\text{count}(s', s, a)}{N},$$

where $\text{count}(s', s, a)$ represents the number of times the state-action pair $(s, a)$ transitions to state $s'$. Moreover, we define $\widehat{M}$ as the empirical RMDP identical to the original $M$ except that it uses $\widehat{P}$ instead of $P_0$ for the transition kernel. We denote by $\widehat{V}^\pi$ and $\widehat{Q}^\pi$ the value functions of a policy $\pi$ in $\widehat{M}$, and $\widehat{\pi}^\star$, $\widehat{Q}^\star$ and $\widehat{V}^\star$ denote the optimal policy and its value functions in $\widehat{M}$. It is assumed that the reward function $R_0$ is known and deterministic and therefore exactly identical in $M$ and $\widehat{M}$. Moreover, we write $P \in \hat{\mathcal{P}}_{s,a}$ for $P = \hat{P}_{s,a} + P'$ with $P' \in \mathcal{P}_{s,a}$ and $P \in \hat{\mathcal{P}}_s$ for $P = \hat{P}_s^\pi + P'$ with $P' \in \mathcal{P}_s$, $\hat{P}_s^\pi(s') = \sum_a \pi(a|s)\hat{P}_{s,a}(s') \in \mathbb{R}^S$.

Notice that our analysis would easily account for an estimated reward (the hard part being handling the estimated transition model). This generative model framework, when we can only sample from the nominal kernel, is classic and appears for both non-robust and robust MDPs [Agarwal et al., 2020, Panaganti et al., 2022, Azar et al., 2013, Xu et al., 2023]. In the robust case, it is especially relevant as an abstraction of "sim-to-real", the simulator giving access to the nominal kernel for learning a robust policy to be deployed in the real world (assumed to belong to the uncertainty set).

The question of how to solve RMDPs and the related computational complexity are complementary, but different from Theorems 4.1 and 5.1. Indeed, an important point that differentiates us from [Panaganti and Kalathil, 2022] is the use of a *robust optimization oracle*. In (model-based) sample complexity analysis, the goal is to determine the smallest sample size $N$ such that a planner executed in $\widehat{M}$ yields a near-optimal policy in the RMDP $M$. To decouple the statistical and computational aspects of planning with respect to an approximate model $\widehat{M}$, we will use an optimization oracle that takes as input an (empirical) RMDP and returns a policy $\hat{\pi}$ that satisfies $\|\hat{Q}^* - \hat{Q}^{\hat{\pi}}\|_\infty \leq \epsilon_{\text{opt}}$. Our final bound will depend on $\epsilon$, the error made from finite sample complexity, and $\epsilon_{\text{opt}}$. In practice, the error $\epsilon_{\text{opt}}$ is typically decreasing at a linear speed of $\gamma^k$ at the $k^{\text{th}}$ iteration of the algorithm, as in classical MDPs because (optimal) Bellman operators are $\gamma$-contraction in both classic and robust settings when robust kernel in assuming in the simplex.

The computational cost of RMDPs is addressed by Iyengar [2005] but not in the $L_p$. Kumar et al. [2022] address this question, in this case, using the regularized form of robust MDPs obtained with relaxed hypothesis on the kernel (See Appendix A.2). The conclusions of the latter are that $L_p$ robust MDPs are computationally as easy as non-robust MDPs for regularized forms, at least for some choices of $p$ for their relaxation. However, in their analysis, the use of $\gamma$-contraction of the Robust Bellman Operator is needed, whereas this is not always the case for sufficiently large $\beta$. Indeed, assuming robust kernel is not anymore in the simplex, Robust Bellman Operator is not anymore a $\gamma$-contraction but an $\epsilon-$contraction for $\epsilon$ close to 1 and only for a small range of $\beta$. (See Derman et al. [2021] Th. 5.1). We address the question of solving RMPDs in the $L_p$ case

with a valid robust kernel in Alg. 1 as it is required to obtain an $\epsilon_{ops}$ solution in our analysis.

# 4 SAMPLE COMPLEXITY WITH $L_p$-BALLS

The aim of this section is to obtain an upper-bound on the sample complexity of RMDPs. This result is true for $sa$- and $s$-rectangular sets and for any $L_p$ norm with $p \geq 1$.

**Theorem 4.1.** *Assume $\delta > 0$, $\epsilon > 0$ and $\beta > 0$. Let $\hat{\pi}$ be any $\epsilon_{opt}$-optimal policy for $\widehat{M}$, i.e. $\|\hat{Q}^{\hat{\pi}} - \hat{Q}^{\star}\|_\infty \leq \epsilon_{opt}$. With $N$ calls to the sampler per state-action pair, such that $N \geq \frac{C\gamma^2 L''}{(1-\gamma)^4 \epsilon^2}$, with $L'' = \log(\frac{32 SAN\|1_s\|_q}{\delta(1-\gamma)})$ we obtain the following guarantee for policy $\hat{\pi}$,*

$$\|Q^* - Q^{\hat{\pi}}\|_\infty \leq \epsilon + \frac{3\gamma\epsilon_{opt}}{1-\gamma}$$

*with probability at least $1 - \delta$, where $C$ is an absolute constant. Finally, for $N_{total} = N|\mathcal{S}||\mathcal{A}|$ and $H = 1/(1-\gamma)$, we get an overall complexity of*

$$N_{total} = \tilde{\mathcal{O}}\left(\frac{H^4|S||A|}{\epsilon^2}\right).$$

## 4.1 DISCUSSION

This result says that the policy $\hat{\pi}$ computed by the planner on the empirical RMDP $\hat{M}$ will be $(\epsilon_{opt} + \epsilon)$-optimal in the original RMDP $M$. As explained before, 1 planning algorithms for RMDPs that guarantee arbitrary small $\epsilon_{opt}$, such as robust value iteration considered by Panaganti and Kalathil [2022]. It will also apply to future planners, as long as they come with a convergence guarantee. The error term $\epsilon$ is controlled by the number of samples: $N_{tot} = \tilde{\mathcal{O}}(H^4|S||A|\epsilon^{-2})$ calls to the generative models allow guaranteeing an error $\epsilon$. This is a gain in terms of sample complexity of $|S|$ compared to Panaganti and Kalathil [2022], for the $sa$-rectangular assumption. Our bound also holds for both $s$- and $sa$-rectangular uncertainty sets. Panaganti et al. [2022] do not study the $s$-rectangular case, while Yang et al. [2021] do, but have a worst dependency to $|A|$ in this case. Their bounds also have additional dependencies on the size of the uncertainty set, which we do not have. We recall that we do not cover the same cases, we do not analyze the KL and Chi-Square robust set, while they do not analyze the $L_p$ robust set for $p > 1$. However, the above comparison holds for the total variation case that we have in common ($p = 1$). These bounds are clearly stated in Table 1. In the non-robust setting, Azar et al. [2013] show that there exist MDPs where the sample complexity is at least $\tilde{\Omega}\left(\frac{H^3|A||S|}{\epsilon^2}\right)$. Section 5 gives a new upper-bound in $H^3$ which matches this lower-bound for non-robust MDPs with an extra condition on the range of $\beta$ (the uncertainty set should be small enough).

## 4.2 SKETCH OF PROOF

This first proof is the simpler one, it relies notably on Ho-effding's concentration arguments. We provide a sketch, the full proof can be found in Appendix B. The resulting bound is not optimal in terms of the horizon $H$, but it also does not impose any condition on the range of $\epsilon$ or $\beta$, contrary to the (better) bound of Sec. 5. We would like to bound the supremum norm of the difference between the optimal Q-function and the one of the policy computed by the planner in the empirical RMDP, according to the true RMDP, $\|Q^* - Q^{\hat{\pi}}\|_\infty$. Using a simple decomposition and the fact that $\pi^*$ is not optimal in the empirical RMDP ($\hat{Q}^{\pi^*} \leq \hat{Q}^* = \hat{Q}^{\hat{\pi}}$), we have that

$$Q^* - Q^{\hat{\pi}} = Q^* - \hat{Q}^* + \hat{Q}^* - \hat{Q}^{\hat{\pi}} + \hat{Q}^{\hat{\pi}} - Q^{\hat{\pi}}.$$

As $Q^* - \hat{Q}^* \leq Q^* - \hat{Q}^{\pi^*}$, a triangle inequality yields

$$\|Q^* - Q^{\hat{\pi}}\|_\infty \leq \|Q^* - \hat{Q}^{\pi^*}\|_\infty + \|\hat{Q}^* - \hat{Q}^{\hat{\pi}}\|_\infty$$
$$+ \|\hat{Q}^{\hat{\pi}} - Q^{\hat{\pi}}\|_\infty.$$

The second term is easy to bound, by the assumption of the planning oracle we have $\|\hat{Q}^* - \hat{Q}^{\hat{\pi}}\|_\infty \leq \epsilon_{opt}$. The two other terms are similar in nature. They compare the Q-functions of the same policy (either $\pi^*$ the optimal one of the original RMDP, or $\hat{\pi}$ the output of the planning algorithm) but for different RMPDs, either the original one or the empirical one. For bounding the remaining terms, we need to introduce the following notation. For any set $\mathcal{D}$ and a vector $v$, let define $\kappa_{\mathcal{D}}(v) = \inf\{u^\top v : u \in \mathcal{D}\}$. This quantity corresponds to the inf form of the robust Bellman operator. The following lemma provides a data-dependent bound of the two terms of interest.

**Lemma 4.2.** *We have with $\mathcal{P}_{s,a}$ defined in Assumption 3.1 and $\hat{\mathcal{P}}_{s,a}$ the robust set centered around the empirical MDPs that*

$$\|Q^{\hat{\pi}} - \hat{Q}^{\hat{\pi}}\|_\infty \leq \frac{\gamma}{1-\gamma} \max_{s,a} |\kappa_{\hat{\mathcal{P}}_{s,a}}(\hat{V}^{\hat{\pi}}) - \kappa_{\mathcal{P}_{0,s,a}}(\hat{V}^{\hat{\pi}})|$$

$$\|Q^* - \hat{Q}^{\pi^*}\|_\infty \leq \frac{\gamma}{1-\gamma} \max_{s,a} |\kappa_{\hat{\mathcal{P}}_{s,a}}(V^*) - \kappa_{\mathcal{P}_{0,s,a}}(V^*)|.$$

For proving these inequalities, we rely on fundamental properties of the (robust) Bellman operator, such as $\gamma$-contraction. This lemma is written for $sa$-rectangular assumption but is also true for $s$-rectangular assumption, replacing notation of robust set $\mathcal{P}_{s,a}$ by $\mathcal{P}_s$. Now, we need to bound the resulting terms, which is done by the following lemma.

**Lemma 4.3.** *With probability at least $1 - \delta$, we have*

$$\max_{s,a} |\kappa_{\hat{\mathcal{P}}_{s,a}}(\hat{V}^{\hat{\pi}}) - \kappa_{\mathcal{P}_{0,s,a}}(\hat{V}^{\hat{\pi}})|$$
$$\leq \frac{10}{(1-\gamma)}\left(\sqrt{\frac{L''}{2N}} + \frac{L''|S|^{1/q}\|1_S\|_q(p-1)}{N}\right) + 2\epsilon_{opt}.$$

*with $L'' = \log(\frac{32SAN\|1\|_q}{\delta(1-\gamma)})$*

Again, this also holds for $s$-rectangular sets. This inequality relies on Hoeffding's based concentration argument coupled with absorbing MDPs of Agarwal et al. [2020] and smoothness of the $L_p$ norm. Putting everything together, we have just shown that :

$$\|Q^* - Q^{\hat\pi}\|_\infty \le \frac{3\gamma\epsilon_{\text{opt}}}{1-\gamma}$$
$$+ \frac{20\gamma}{(1-\gamma)^2}\left(\sqrt{\frac{L''}{2N}} + \frac{L''|S|^{1/q}\|1_S\|_q\,(p-1)}{N}\right)$$

Solving in $\epsilon$ for the second term of the right-hand side gives the stated result as the term proportional to $1/N$ is small compared to the second one for sufficiently small $\epsilon$.

# 5 TOWARD MINIMAX OPTIMAL SAMPLE COMPLEXITY

Now, we provide a better bound in terms of the horizon $H$, reaching (up to log factors) the lower-bound in $H^3$ for non-robust MDPs. Recall $\beta = \sup_{s,a} \beta_{s,a}$ for the $sa$-rectangular assumption or $\beta = \sup_s \beta_s$ for the $s$-rectangular assumption. For the following result to hold, we need to assume that the uncertainty set is small enough: we will require

$$\beta \le \frac{1-\gamma}{2\gamma|S|^{1/q}} = \frac{1}{2(H-1)|S|^{1/q}}.$$

The following theorem is true for both $sa$- and $s$-rectangular uncertainty sets, and for any $L_p$ norm with $p \ge 1$.

**Theorem 5.1.** *let $\beta_0 \in (0, \frac{1}{2(H-1)|S|^{1/q}}]$, for any $\kappa > 0$ and any $\epsilon_0 \le \kappa\sqrt{H}$ it exists a $C_{\beta_0,\epsilon_0} > 0$ independent of $H$ such that for any $\beta \in (0, \beta_0)$ and any $\epsilon \in (0, \epsilon_0)$, whenever $N$ the number of calls to the sampler per state-action pair satisfies $N \ge C_{\beta_0,\epsilon_0} \frac{L\gamma^2 H^3}{\epsilon^2}$ where $L = \log(8|\mathcal{S}||\mathcal{A}|/((1-\gamma)\delta))$, it holds that if $\widehat\pi$ is any $\epsilon_{opt}$ -optimal policy for $\widehat{M}$, that is when $\|\widehat{Q}^{\widehat\pi} - \widehat{Q}^\star\|_\infty \le \epsilon_{opt}$, then*

$$\|Q^* - Q^{\hat\pi}\|_\infty \le \epsilon + \frac{8\epsilon_{opt}}{1-\gamma}$$

*with probability at least $1 - \delta$.*

*So $N_{total} = N|\mathcal{S}||\mathcal{A}|$ as an overall sample complexity*

$$\tilde{\mathcal{O}}\left(\frac{H^3|S||A|}{\epsilon^2}\right)$$

*for any $\epsilon < \epsilon_0$.*

## 5.1 DISCUSSION

The constants of Theorem 5.1 are explicitly given in Appendix C. For instance, for $\beta_0 = \frac{1}{8(H-1)}$ and $\epsilon_0 = \sqrt{16H}$,

we have $C = 1024$, other choices being possible. Recall that in the non-robust case, the lower-bound is $\tilde{\Omega}\left(\frac{H^3|S||A|}{\epsilon^2}\right)$ Azar et al. [2013]. Our theorem states that any model-based robust RL approach, in the generative model setting, with an accurate enough planner applied to the empirical RMDP, reaches this lower bound, up to log terms. As far as we know, it is the first time that one shows that solving an RMDP in this setting does not require more samples than solving a non-robust MDP, provided that the uncertainty set is small enough. Our bound on $\epsilon$ is similar to the one of Agarwal et al. [2020] in the robust case with their range $[0, \sqrt{H})$, we differ only by giving more flexibility in the choice of the constant $C$. The best range of $\epsilon$ for non-robust MDPs is $(0, H)$ [Li et al., 2020], we let its extension to the robust case for future work. So far, we discussed the lower-bound for the non-robust case, that we reach. Indeed, non-robust MDPs can be considered as a special case of MDPs with $\beta = 0$. As far as we know, the only robust-specific lower-bounds on the sample complexity have been proposed by Yang et al. [2021]. They propose two lower-bounds accounting for the size of the uncertainty set, one for the Chi-square case, and one for the total variation case, which coincide with our $L_p$ framework for $p = 1$ This bound is

$$\tilde{\Omega}\left(\frac{|\mathcal{S}||\mathcal{A}|(1-\gamma)}{\varepsilon^2}\min\left\{\frac{1}{(1-\gamma)^4}, \frac{1}{\beta^4}\right\}\right).$$

This lower bound has two cases, depending on the size of the uncertainty set. If $\beta \le (1-\gamma) = 1/H$, we retrieve the non-robust lower bound $\tilde{\Omega}\left(\frac{|\mathcal{S}||\mathcal{A}|H^3}{\varepsilon^2}\right)$. Therefore, for a $L_1$-ball, our upper-bound matches the lower-bound, and we have proved that model-based robust RL in the generative model setting is minimax optimal for any accurate enough planner. Their condition for this bound, $\beta \le 1/H$, is close to our condition, $\beta < 1/(4(H-1))$. This suggests that our condition on $\beta$ is not just a proof artifact. In the second case, if $\beta > 1 - \gamma$, the lower-bound is $\tilde{\Omega}\left(\frac{|\mathcal{S}||\mathcal{A}|(1-\gamma)}{\varepsilon^2\beta^4}\right)$. In this case, our theorem does not hold, and we only currently get a bound in $H^4$ (see Sec. 4), which doesn't match this lower-bound.

In the case of $TV$, we know from posterior work Shi et al. [2023] that it is possible to get a tighter bound in the regime $\beta > 1 - \gamma$ but in the case of $L_P$ norm, it is still an open question. In the case where $\beta$ is too large, the question arises whether RMDPs are useful as long as there is little to control when the transition kernel can be too arbitrary.

To sum up, to the best of our knowledge, with a small enough uncertainty set, our work delivers the first-ever minimax-optimal guarantee for RMDPs according to the non-robust lower-bound for $L_p$-balls, and the first ever minimax-optimal guarantee according to the robust lower-bound for the total variation case for a sufficiently small radius of the uncertainty set, which has been later on the larger set of $\beta$ by Shi et al. [2023]. '

## 5.2 SKETCH OF PROOF

The full proof is provided in Appendix C. As in Sec. 4.2, we start from the inequality

$$\|Q^* - Q^{\hat{\pi}}\|_\infty \leq \|Q^* - \hat{Q}^{\pi^*}\|_\infty + \|\hat{Q}^* - \hat{Q}^{\hat{\pi}}\|_\infty + \|\hat{Q}^{\hat{\pi}} - Q^{\hat{\pi}}\|_\infty,$$

where the second term of the right-hand side can again be readily bounded, $\|\hat{Q}^* - \hat{Q}^{\hat{\pi}}\|_\infty \leq \epsilon_{\text{opt}}$. To bound the remaining two terms, if we want to obtain a tighter final bound, the contracting property of the robust Bellman operator will not be enough, we need a finer analysis. To achieve this, we rely on the total variance technique introduced by Azar et al. [2013] for the non-robust case, combined with the *absorbing MDP* construction of Agarwal et al. [2020], also for the non-robust case, which allows improving the range of valid $\epsilon$. The key underlying idea is to rely on a Bernstein concentration inequality rather than a Hoeffding one, therefore considering the variance of the random variable rather than its range, tightening the bound. Working with a Bernstein inequality will require controlling the variance of the return. A key result was provided by Azar et al. [2013], that we extend to the robust setting,

$$\left\|(I - \gamma P_0^\pi)^{-1} \sqrt{\text{Var}_{P_0}(V^\pi)}\right\|_\infty \leq \sqrt{\frac{2}{(1-\gamma)^3}}. \quad (1)$$

Naively bounding the left-hand side would provide a bound in $H^2$, while this (non-obvious) bound in $\sqrt{H^3}$ is crucial for obtaining on overall dependency in $H^3$ in the end. Now, we come back to the terms $\|Q^* - \hat{Q}^{\hat{\pi}^*}\|_\infty$ and $\|Q^{\hat{\pi}} - \hat{Q}^{\hat{\pi}}\|_\infty$ that we have to bound. This bound should involve a term proportional to $(I - \gamma P_0^\pi)^{-1}$ to leverage later Eq. (1). The following lemma is inspired by Agarwal et al. [2020], and its proof relies crucially on having a simple dual of robust Bellman operator.

**Lemma 5.2.**

$$\|Q^{\hat{\pi}} - \hat{Q}^{\hat{\pi}}\|_\infty \leq \gamma\|(I - \gamma P_0^{\hat{\pi}})^{-1}(P_0 - \hat{P})\hat{V}^{\hat{\pi}}\|_\infty$$
$$+ \frac{2\gamma\beta|S|^{1/q}}{1-\gamma}\|Q^{\hat{\pi}} - \hat{Q}^{\hat{\pi}}\|_\infty.$$

We see that the term $\beta$ appears in the bound. This comes from the need to control the difference in penalization between seminorms of value functions, from a technical viewpoint. Indeed, the terms $\frac{2\gamma\beta}{1-\gamma}\|Q^\pi - \hat{Q}^\pi\|_\infty$ (with $\pi$ being either $\hat{\pi}$ or $\pi^*$) are not present in the non-robust version of the bound, and are one of the main differences from the derivation of Agarwal et al. [2020]. The first term of the right-hand side of each bound $\|(I - \gamma P_0^\pi)^{-1}(P_0 - \hat{P})\hat{V}^\pi\|_\infty$ (with $\pi$ being either $\hat{\pi}$ or $\pi^*$, again) will be upper-bounded using a Bernstein argument, leveraging also Eq. (1). The resulting lemma is the following.

**Lemma 5.3.** *With probability at least $1 - \delta$, we have*

$$\left\|Q^{\hat{\pi}} - \widehat{Q}^{\hat{\pi}}\right\|_\infty < (C_N + C_\beta)\|Q^{\hat{\pi}} - \hat{Q}^{\hat{\pi}}\|_\infty$$
$$+ 4\gamma\sqrt{\frac{L}{N(1-\gamma)^3}} + \frac{\gamma\Delta'_{\delta,N}}{1-\gamma} + \frac{\gamma\epsilon_{\text{opt}}}{1-\gamma}\left(2 + \sqrt{\frac{8L}{N}}\right),$$

*with $C_\beta = \frac{2\gamma\beta|S|^{1/q}}{1-\gamma}$ and $C_N = \frac{\gamma}{1-\gamma}\sqrt{\frac{8L}{N}}$ and where $\Delta'_{\delta,N} = \sqrt{\frac{cL}{N}} + \frac{cL}{(1-\gamma)N}$ with $L = \log(8|S||A|/((1-\gamma)\delta))$.*

For this result to be exploitable, we have to ensure that $C_N + C_\beta < 1$, which leads to $\beta \leq \frac{1-\gamma}{2\gamma|S|^{1/q}}$, and then $C_N + C_\beta < 1$ leads to a constraint on $N$ in Theorem 5.1. Eventually, injecting the result of this last lemma in the initial bound, keeping the dominant term in $1/\sqrt{N}$ and solving for $\epsilon$ provides the stated result, cf Appendix C.

## 6 CONCLUSION

In this paper, we have studied the question of the sample complexity of model-based robust reinforcement learning. To decouple this from the problem of exploration, we have considered the classic (in non-robust RL) generative model setting, where a sampler can provide next-state samples from the nominal kernel and from arbitrary state-action couples. We focused our study more specifically on $sa$- and $s$-rectangular uncertainty sets corresponding to $L_p$-balls around the nominal.

Without any restriction on the size of uncertainty set ($\beta$), we have shown that the sample complexity of the studied general setting is $\tilde{\mathcal{O}}(\frac{|S||A|H^4}{\epsilon^2})$, already significantly improving existing results [Yang et al., 2021, Panaganti and Kalathil, 2022]. Our bound holds for both the $sa$- and $s$-rectangular cases, and improves existing results (for the total variation) by respectively $|S|$ and $|S||A|$. By assuming a small enough uncertainty set, and for a small enough $\epsilon$, we further improved this bound to $\tilde{\mathcal{O}}(\frac{|S||A|H^3}{\epsilon^2})$, adapting proof techniques from the non-robust case [Azar et al., 2013, Agarwal et al., 2020]. This is a significant improvement. Our bound again holds for both the $sa$- and $s$- rectangular cases, it matches the lower-bound for the non-robust case Azar et al. [2013], and it matches the total variation lower-bound for the robust case when the uncertainty set is small enough [Yang et al., 2021]. We think this is an important step towards minimax optimal robust reinforcement learning.

There are a number of natural perspectives, such as knowing if we could extend our results to other kinds of uncertainty sets, or to extend our last bound to larger uncertainty sets (despite the fact that if the dynamics are too unpredictable, there may be little left to be controlled). Our results build heavily on the simple dual form of the robust Bellman operator, which prevents us from considering, for the moment,

uncertainty sets based on the KL or Chi-square divergence. Beyond their theoretical advantages, these simple dual forms also provide practical and computationally efficient planning algorithms. Therefore, another interesting research direction would be to know if one could derive additional useful uncertainty sets relying primarily on the regularization viewpoint.

# 7 ACKNOWLEDGEMENTS

Pierre Clavier has been supported by a grant from Région Île-de-France.

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

# Towards Minimax Optimality
# of Model-based Robust Reinforcement Learning
# (Supplementary Material)

Pierre Clavier [1,2,3]        Erwan Le Pennec[1]        Matthieu Geist[4]

[1]CMAP, CNRS, Ecole Polytechnique, Institut Polytechnique de Paris, 91120 Palaiseau, France,
[2]INRIA Paris, HeKA, France,
[3]Centre de Recherche des Cordeliers, INSERM, Universite de Paris, Sorbonne Universite, F-75006 Paris, France,
[4]Cohere.

## A   OVERVIEW AND USEFUL INEQUALITIES

The appendix is organized as follows

- In Appendix A.1, a comprehensive table with state-of-the-art complexity for every distance.

- In Appendix A.2, we provide more details/explanations on the difference between our formulation on the one of Kumar et al. [2022] and Derman et al. [2021].

- In Appendix A.3, we give more details about our algorithm :DRVI L$_P$

- In Appendix A.4, we give some useful inequalities frequently used in the proofs.

- In Appendix B, we prove Theorem 4.1.

- In Appendix C, we prove Theorem 5.1.

Finally, the proofs for the $s$-rectangular and $sa$-rectangular cases are often very similar. If this is true, we will combine them in a single proof with the two cases detailed when needed.

### A.1   TABLE OF SAMPLE COMPLEXITY

Table 2: Sample Complexity for different metric and $s$- or $sa$ rectangular assumptions with $\beta$ the radius of uncertainty set, $H$ the horizon factor, $\epsilon$ the precicion, $\bar{p}$, $\beta_{0,p} = (1-\gamma)/(2\gamma|S|^{1/q})$. the smallest positive state transition probability of the nominal kernel visited by the optimal robust policy (see Yang et al. [2021]).

| | Panaganti and Kalathil [2022] | Yang et al. [2021] | Shi and Chi [2022] | Our $\beta \geq 0$ | Our $\beta_{0,p} > \beta > 0$ | Shi et al. [2023] $\beta > 1-\gamma$ | Shi et al. [2023] $0 < \beta < 1-\gamma$ |
|---|---|---|---|---|---|---|---|
| TV $(sa)$ | $\tilde{O}\left(\frac{|S|^2|A|H^4}{\epsilon^2}\right)$ | $\tilde{O}\left(\frac{|S|^2|A|H^4(2+\beta)^2}{\epsilon^2\beta^2}\right)$ | × | $\tilde{O}\left(\frac{|S||A|H^4}{\epsilon^2}\right)$ | $\tilde{O}\left(\frac{|S||A|H^3}{\epsilon^2}\right)$ | $\tilde{O}\left(\frac{|S||A|H^2}{\epsilon^2\beta}\right)$ | $\tilde{O}\left(\frac{|S||A|H^3}{\epsilon^2}\right)$ |
| TV $(s)$ | × | $\tilde{O}\left(\frac{|S|^2|A|H^4(2+\beta)^2}{\epsilon^2\beta^2}\right)$ | × | $\tilde{O}\left(\frac{|S||A|H^4}{\epsilon^2}\right)$ | $\tilde{O}\left(\frac{|S||A|H^3}{\epsilon^2}\right)$ | × | × |
| $L_p$ $(sa)$ | × | × | × | $\tilde{O}\left(\frac{|S||A|H^4}{\epsilon^2}\right)$ | $\tilde{O}\left(\frac{|S||A|H^3}{\epsilon^2}\right)$ | × | × |
| $L_p$ $(s)$ | × | × | × | $\tilde{O}\left(\frac{|S||A|H^4}{\epsilon^2}\right)$ | $\tilde{O}\left(\frac{|S||A|H^3}{\epsilon^2}\right)$ | × | × |
| $\chi^2$ $(sa)$ | $\tilde{O}\left(\frac{|S|^2|A|\beta H^4}{\epsilon^2}\right)$ | $\tilde{O}\left(\frac{|S|^2|\mathcal{A}|(1+\beta)^2 H^4}{\varepsilon^2(\sqrt{1+\beta}-1)^2}\right)$ | × | × | × | $\tilde{O}\left(\frac{|S||A|\beta H^4}{\epsilon^2}\right)$ | $\tilde{O}\left(\frac{|S||A|\beta H^4}{\epsilon^2}\right)$ |
| $\chi^2$ $(s)$ | × | $\tilde{O}\left(\frac{|S|^2|\mathcal{A}^3|(1+\beta)^2 H^4}{\varepsilon^2(\sqrt{1+\beta}-1)^2}\right)$ | × | × | × | | × |
| KL $(sa)$ | $\tilde{O}\left(\frac{|S|^2|\mathcal{A}|\exp(H)H^4}{\beta^2\varepsilon^2}\right)$ | $\tilde{O}\left(\frac{|S|^2|A|H^4}{\bar{p}^2\varepsilon^2\beta^2}\right)$ | $\tilde{O}\left(\frac{|S||A|H^4}{\bar{p}\epsilon^2\beta^4}\right)$ | × | × | × | × |
| KL $(s)$ | × | $\tilde{O}\left(\frac{|S|^2|A|^2H^4}{\bar{p}^2\varepsilon^2\beta^2}\right)$ | × | × | × | × | × |

## A.2 RELATION WITH THE WORK OF Kumar et al. [2022] AND Derman et al. [2021]

In the work of Derman et al. [2021] close forms for RMDPs with $L_p$ norms are derived assuming the following uncertainty set :

**Assumption A.1.** *(sa-rectangularity in Derman et al. [2021])*

$$\mathcal{U}_p^{sa} := (R_0 + \mathcal{R}) \times (P_0 + \mathcal{P}), \mathcal{R} = \times_{s \in \mathcal{S}, a \in \mathcal{A}} \mathcal{R}_{s,a}, \mathcal{R}_{s,a} = \left\{ r_{s,a} \in \mathbb{R} \mid \|r_{s,a}\|_p \leq \alpha_{s,a} \right\}$$

$$\mathcal{P} = \times_{s \in \mathcal{S}, a \in \mathcal{A}} \mathcal{P}_{s,a} \mathcal{P}_{s,a} = \{P_{s,a} : \mathcal{S} \rightarrow \mathbb{R}, \|P_{s,a}\|_p \leq \beta_{s,a}\}$$

Using these uncertainty sets leads to the following Bellman Operator :

**Theorem A.2** (Derman et al. [2021]). *The $sa$-rectangular Robust Bellman operator is equivalent to a regularized non-robust Bellman operator: for $r_{V,\pi}^{s,a}(s,a) = -\left(\alpha_s + \gamma\beta_{s,a} \|V\|_q\right) + R_0(s,a)$ as we have*

$$\mathcal{T}_{\mathcal{U}_p^{sa}}^{\pi} V(s) = \langle \pi_s, r_{V,\pi}^{s,a}(s,a) + \gamma \sum_{s'} P_0\left(s' \mid s, a\right) V\left(s'\right) \rangle_A$$

Using this formulation, they get a closed form for the inner minimization problem and for the Robust Bellman Operator

The work Kumar et al. [2022] modifies the work of Derman et al. [2021] using Kernel that sum to 1, $\sum_{s'} P_{s,a}(s') = 0$ in their definition, but using this uncertainty set, it is still possible to get a robust kernel out of the simplex. Using this formulation, they also get a closed form for the inner minimization problem and for the Robust Bellman Operator.

**Assumption A.3.** *(sa-rectangularity in Kumar et al. [2022])*

$$\mathcal{U}_p^{sa} := (R_0 + \mathcal{R}) \times (P_0 + \mathcal{P}), \mathcal{R} = \times_{s \in \mathcal{S}, a \in \mathcal{A}} \mathcal{R}_{s,a}, \mathcal{R}_{s,a} = \left\{ r_{s,a} \in \mathbb{R} \mid \|r_{s,a}\|_p \leq \alpha_{s,a} \right\}$$

$$\mathcal{P} = \times_{s \in \mathcal{S}, a \in \mathcal{A}} \mathcal{P}_{s,a} \mathcal{P}_{s,a} = \{P_{s,a} : \mathcal{S} \rightarrow \mathbb{R} \mid \sum_{s'} P_{s,a}\left(s'\right) = 0, \|P_{s,a}\|_p \leq \beta_{s,a}\}$$

Using these uncertainty sets where robust Kernel may not belong anymore to the simplex as they do not assume $P_0 + P_{s,a} \geq 0$. This leads to the following Bellman Operator :

**Theorem A.4** (Kumar et al. [2022]). *The $sa$-rectangular Robust Bellman operator is equivalent to a regularized non-robust Bellman operator: for $r_{V,\pi}^{s,a}(s,a) = -\left(\alpha_s + \gamma\beta_{s,a}\mathrm{sp}_q(V)\right) + R_0(s,a)$, as we have*

$$\mathcal{T}_{\mathcal{U}_p^{sa}}^{\pi} V(s) = \langle \pi_s, r_{V,\pi}^{s,a}(s,a) + \gamma \sum_{s'} P_0\left(s' \mid s, a\right) V\left(s'\right) \rangle_A$$

where $\mathrm{sp}_q(V)$ in defined in Def. 3.1. These results are due to the following lemma.

**Lemma A.5** ( Kumar et al. [2022]. Duality for the minimization problem for $sa$ rectangular case with $L_p$ norm without simplex constrain)**.**

$$\inf_{P:\sum_{s'} P(s')=0\|P-\hat{P}_{s,a}\|_p \leq \beta_{s,a}} PV = \widehat{P}_{s,a}V - \beta_{s,a}\mathrm{sp}_q(V)$$

Our analysis assumes the positivity of the kernel function, $P_0 + P_s \geq 0$ in s-rectangular or $P_0 + P_{s,a} \geq 0$ for $sa$-rectangular case. Using this more realistic assumption, we can not obtain a closed form of the robust Bellman operator. However, we are still able to compute a dual form for the inner minimization problem of RMDPs. With our definition of rectangularity in the simplex:

**Assumption A.6.** *(sa-rectangularity) We define $sa$-rectangular $L_p$-constrained uncertainty set as*

$$\mathcal{U}_p^{sa} := (R_0 + \mathcal{R}) \times (P_0 + \mathcal{P}), \mathcal{R} = \times_{s \in \mathcal{S}, a \in \mathcal{A}} \mathcal{R}_{s,a}, \mathcal{P} = \times_{s \in \mathcal{S}, a \in \mathcal{A}} \mathcal{P}_{s,a}, \mathcal{R}_{s,a} = \{r_{s,a} \in \mathbb{R} \mid |r_{s,a}| \leq \alpha_{s,a}\}$$

$$\mathcal{P}_{s,a} = \{P_{s,a} : \mathcal{S} \rightarrow \mathbb{R} \mid \sum_{s'} P_{s,a}(s') = 0, P_{0,s,a} + P_{s,a} \geq 0, , \|P_{s,a}\|_p \leq \beta_{s,a}\}$$

and using $\kappa_{\mathcal{D}}(v) = \inf\{u^\top v : u \in \mathcal{D}\}$ ., we obtain :

**Lemma A.7** (Duality for the minimization problem for $sa$ rectangular case with $L_p$ norm).

$$\kappa_{\widehat{\mathcal{P}}_{s,a}}(V) = \max_{\mu \geq 0}\{\widehat{P}_{s,a}(V - \mu) - \beta_{s,a}\mathrm{sp}_q(V - \mu)\}$$

Proof can be found on Appendix B.5

Contrary to previous lemma in Kumar et al. [2022], there is an additional $\max$ operator in our dual formulation. Interestingly, their formulation is a relaxation of our Lemmas 3.3 as their formulation does not assume the positivity of the kernel. Their relaxation allows practical algorithms with close form, but still suffer from non-exact formulation of RMDPs with robust Kernel that are not in the simplex.

One crucial point in our analysis is that Bellman Operator for RMDPs is a $\gamma$- contraction for robust kernel in the simplex for any radius $\beta$ (see Iyengar [2005]). For Kumar et al. [2022] and Derman et al. [2021] the range of $\beta$ where their Robust Bellman Operator is a contraction is smaller than $\frac{1-\gamma}{\gamma|S|^{1/q}}$ (see Proposition 4 of Derman et al. [2021]) which is the range where we have minimax optimality in our Theorem 5.1. For $\beta > \frac{1-\gamma}{\gamma|S|^{1/q}}$, there is no contraction anymore. In the following, we will assume that robust kernels belong to the simplex to use $\gamma$-contraction in our proof of sample complexity and ensure convergence of the following Distributionally Robust value Iteration for $L_p$ norms for any $\beta$ Algoritm 1.

## A.3   MODEL BASED DRVI L$_P$ ALGORITHM

---

**Algorithm 1:** DRVI L$_P$: Distributionally robust value iteration DRVI for $L_P$ norms with $sa-$rectangular assuptions

---
1  **input:** empirical nominal transition kernel $\widehat{P}_0$; reward function $r$; uncertainty level $\beta$.
2  **initialization:** $\widehat{Q}_0(s,a) = 0$, $\widehat{V}_0(s) = 0$ for all $(s,a) \in S \times A$.
3  **for** $t = 1, 2, \cdots, T$ **do**
4  $\quad$ **for** $\forall s \in S, a \in A$ **do**
5  $\quad\quad$ Set $\widehat{Q}_t(s,a)$ according to (2) for $sa-$rectangular ;
6  $\quad$ **for** $\forall s \in S$ **do**
7  $\quad\quad$ Set $\widehat{V}_t(s) = \max_a \widehat{Q}_t(s,a)$;
8  **output:** $\widehat{Q}_T$, $\widehat{V}_T$ and $\widehat{\pi}$ obeying $\widehat{\pi}(s) = \arg\max_a \widehat{Q}_T(s,a)$.

---

We propose Alg. 1 to solve robust MDPs in the case of $L_P$ norms using value Iteration with $sa$- rectangularity assumptions. First, we can remark that directly solving classical RMDPs formulation is computationally costly as it requires an optimization over an $S$-dimensional probability simplex at each iteration, especially when the dimension of the state space $S$ is large. However, using strong duality like Iyengar [2005] for the $TV$, one can also solve using the dual problem of this formulation. The equivalence between the two formulations can be found in Lemma 3.3. Using the dual form, the optimization (3) reduces to a 2-dimensional optimization problem that can be solved efficiently using any $2-$dimensional convex solver if there exists an analytic form of the span-semi norm. Then the iterates $\{\widehat{Q}_t\}_{t\geq 0}$ of DRVI for $L_P$ norms converge linearly to the fixed point $\widehat{Q}^\star$, owing to the appealing $\gamma$-contraction property of robust MDPs in the simplex. From an initialization $\widehat{Q}_0 = 0$, the update rule at the $t$-th ($t \geq 1$) iteration can be formulated as for $sa$-rectangular case as:

$$\forall (s,a) \in S \times A: \quad \widehat{Q}_t(s,a) = r(s,a) + \max_{\mu \geq 0}\widehat{P}(\hat{V}_{t-1} - \mu) - \beta_{s,a}\mathrm{sp}_q(\hat{V}_{t-1} - \mu) \tag{2}$$

$$= r(s,a) + \max_{\alpha_{\widehat{P}}^{\lambda,\omega} \in A_{\widehat{P}}^{\lambda,\omega}} \widehat{P}[\widehat{V}_{t-1}]_{\alpha_{\widehat{P}}^{\lambda,\omega}} - \beta_{s,a}\mathrm{sp}_q([\widehat{V}_{t-1}]_{\alpha_{\widehat{P}}^{\lambda,\omega}}) \tag{3}$$

where the variational family $A_{\widehat{P}}^{\lambda,\omega}$ is a $2-$dimensional variational family defined in (8). The specific form of the dual problem depends on the choice of the norm. In the case of $L_1$, $L_2$, or $L_\infty$, span semi-norms involved in dual problems have closed form (respectively equals to median, variance, or span), and equation 3 corresponds to a 2-D minimization problem.

But in general cases, one has to compute span-semi norms that can be easily computed using binary search solving

$$\sum_s \operatorname{sign}\left(v(s) - \omega_p(v)\right) |v(s) - \omega_p(v)|^{\frac{1}{p-1}} = 0$$

to compute $\omega_q$ and then setting the semi norm $\operatorname{sp}_q(v) = \|v - \omega_q\|$. Recall the $q$-variance function $\operatorname{sp}_q : \mathcal{S} \to \mathbb{R}$ and $q$-mean function $\omega_q : \mathcal{S} \to \mathbb{R}$ be defined as

$$\operatorname{sp}_q(v) := \min_{\omega \in \mathbb{R}} \|v - \omega \mathbf{1}\|_q, \quad \omega_q(v) := \arg\min_{\omega \in \mathbb{R}} \|v - \omega \mathbf{1}\|_q.$$

See Kumar et al. [2022] for discussion about computing span semi norms. So in the general case, we can also compute the maximum solving :

$$\forall (s, a) \in S \times A : \quad \widehat{Q}_t(s, a) = r(s, a) + \max_{\alpha_{\widehat{P}}^{\lambda, \omega} \in A_{\widehat{P}}^{\lambda, \omega}} \widehat{P}[\widehat{V}_{t-1}]_{\alpha_{\widehat{P}}^{\lambda, \omega}} - \beta_{s,a} \left\| [\widehat{V}_{t-1}]_{\alpha_{\widehat{P}}^{\lambda, \omega}} - w \right\|_q,$$

Using any $2-$D convex optimization algorithm solves the problem as this problem is jointly concave in $(\lambda, w)$ because $(\lambda, w) \to -\left\| [\widehat{V}_{t-1}]_{\alpha_{\widehat{P}}^{\lambda, \omega}} - w \right\|_q$ is concave using norm property and $(\lambda, w) \to \widehat{P}[\widehat{V}_{t-1}]_{\alpha_{\widehat{P}}^{\lambda, \omega}}$ also. Then the sum is concave.

Finally, in the $sa$-case we compute the best policy which is the greedy policy of the final Q-estimates $\widehat{Q}_T$ as the final policy $\widehat{\pi}$:

$$\forall s \in S : \quad \widehat{\pi}(s) = \arg\max_a \widehat{Q}_T(s, a).$$

## A.4 USEFUL INEQUALITIES AND NOTATIONS

Here we present some useful inequalities used frequently in the derivation. Consider any $P$ a transition matrix and $\beta_s$ for $s$ rectangular uncertain sets or $\beta_{sa}$ for $sa$- uncertainty sets, then for $\mathbb{I} = (1, 1, ..., 1)^\top$ :

$$(1 - \gamma P)^{-1} (\gamma \beta_s) \mathbb{I} < \frac{\beta}{1 - \gamma} \mathbb{I} \text{ and } (1 - \gamma P)^{-1} \mathbb{I} \leq \frac{1}{1 - \gamma} \mathbb{I} \tag{4}$$

$$\forall q \in \mathbb{N}^*, \quad \operatorname{sp}_q(.) \leq 2 \|.\|_q < 2|S|^{1/q} \|.\|_\infty, \quad \operatorname{sp}(.)_\infty \leq 2 \|.\|_\infty \tag{5}$$

$$\operatorname{sp}_q(.) \leq 2 \|.\|_q \leq 2 \|.\|_q \tag{6}$$

Eq. (4) is true, taking the supremum norm of the left-hand side inequality. Eq. (5) and Eq. (6) come from properties of norms, see Eq. (1) from Scherrer [2013].

Finally we denote the truncation operator for a vector $\alpha \in \mathbb{R}^S$,

$$[V]_\alpha := \begin{cases} \alpha(s), & \text{if } V(s) > \alpha(s) \\ V(s), & \text{otherwise.} \end{cases}$$

## A.5 ROBUST BELLMAN OPERATOR AND ROBUST Q VALUES

This is proof of Lemma 3.5:

**Lemma A.8.** *Robust Bellman Operator for $sa-$ and $s-$ rectangular are :*

$$\mathcal{T}_{\mathcal{U}_p^{sa}}^\pi V(s) = \sum_a \pi(a|s) \left( -\alpha_{s,a} + R_0(s, a) + \gamma \sum_{s'} P_0(s', s, a)v(s') + \gamma \min_{P \in \mathcal{P}_{s,a}} PV \right)$$

$$\mathcal{T}_{\mathcal{U}_p^s}^\pi V(s) = -\|\pi_s\|_q \alpha_s + \gamma \min_{P^\pi \in \mathcal{P}_s} P^\pi V + \sum_a \pi(a|s) \left( R_0(s, a) + \gamma P_0(s'|s, a)V(s') \right)$$

*Proof.* For $sa$-rectangular: by rectangularity

$$\mathcal{T}_{\mathcal{U}_p^{sa}}^\pi V(s) = \sum_a \pi(a|s)\left(-\alpha_{s,a} + R_0(s,a) + \gamma \min_{P \in P_0 + \mathcal{P}_{s,a}} PV\right)$$

$$= \sum_a \pi(a|s)\left(-\alpha_{s,a} + R_0(s,a) + \gamma \min_{P \in \mathcal{P}_{s,a}} PV + P_{0,s,a}V\right)$$

For $s-$rectangular case :

$$\mathcal{T}_{\mathcal{U}_p^s}^\pi V(s) = \min_{P^\pi \in P_0^\pi + \mathcal{P}_s,} \gamma PV + \min_{R \in R_0^\pi + \mathcal{R}_s} \sum_a \pi(a|s)R(s,a)$$

$$= \sum_a \pi(a|s)R_0(s,a) + \min_{R \in \mathcal{R}_s} \sum_a \pi(a|s)R(s,a) + \sum_a \pi(a|s)\gamma \sum_{s'} P_0(s'|s,a)V(s') + \min_{P^\pi \in \mathcal{P}_s,} \gamma P^\pi V$$

$$\stackrel{(a)}{=} \sum_a \pi(a|s)\left(R_0(s,a) + \sum_{s'} P_0(s'|s,a)V(s')\right) - \alpha_s \|\pi_s\|_q + \min_{P^\pi \in \mathcal{P}_s} \gamma P^\pi V$$

where (a) comes from Holder's inequality. $\qquad \square$

**Lemma A.9.** *For $sa-$ and $s-$ rectangular,*

$$Q_{sa}^\pi(s,a) = r_{Q_{sa}^\pi}^{(s,a)} + \gamma P_{0,s,a} V_{sa}^\pi,$$

$$Q_s^\pi(s,a) = r_{Q_s^\pi}^s + \gamma P_{0,s,a} V_s^\pi$$

*with*

$$r_{Q_{sa}^\pi}^{(s,a)} = R_0(s,a) - \alpha_{s,a} + \gamma \min_{P \in \mathcal{P}_{s,a}} PV_{sa}^\pi$$

$$r_{Q_s^\pi}^s = R_0(s,a) - \left(\frac{\pi_s(a)}{\|\pi_s\|_q}\right)^{q-1} \alpha_s + \gamma \min_{P^\pi \in \mathcal{P}_s} P^\pi V_s^\pi)$$

*Proof.* The result comes directly as for $sa$-rectangular the following relations hold,

$$V_{sa}^\pi(s) = \sum_a \pi(a|s)Q_{sa}^\pi(s,a) \quad \text{and}$$

and for $s$-rectangular case

$$V_s^\pi(s) = \sum_a \pi(a|s)Q_s^\pi(s,a).$$

Then using fixed point equation of Bellman operator: $\mathcal{T}_{\mathcal{U}_p^s}^\pi V_s^\pi(s) = V_s^\pi(s)$ or $\mathcal{T}_{\mathcal{U}_p^{sa}}^\pi V_{sa}^\pi(s) = V_{sa}^\pi(s)$ and previous Lemma A.8 for the expression of $\mathcal{T}_{\mathcal{U}_p^s}^\pi V_s^\pi(s)$, we can identify the robust $Q$ values that give the result

$\qquad \square$

# B   AN $H^4$ BOUND FOR $L_p$-BALLS

To lighten notations, we remove subscript s in most places and denote for example $V^\pi$ instead of $V_s^\pi$ for $s$-rectangular sets.

**Lemma B.1** (Decomposition of the bound).

$$\left\|Q^* - Q^{\hat\pi}\right\|_\infty \leq \left\|Q^* - \hat{Q}^{\pi^*}\right\|_\infty + \left\|\hat{Q}^{\pi^*} - \hat{Q}^{\hat\pi}\right\|_\infty + \left\|\hat{Q}^{\hat\pi} - Q^{\hat\pi}\right\|_\infty$$

*Proof.*

$$0 \leq Q^* - Q^{\hat{\pi}} = Q^* - \underbrace{\hat{Q}^*}_{\geq \hat{Q}^{\pi*}} + \hat{Q}^* - \hat{Q}^{\hat{\pi}} + \hat{Q}^{\hat{\pi}} - Q^{\hat{\pi}}$$

$$\leq Q^* - \hat{Q}^{\pi*} + \hat{Q}^* - \hat{Q}^{\hat{\pi}} + \hat{Q}^{\hat{\pi}} - Q^{\hat{\pi}}$$

$$\Rightarrow \|Q^* - Q^{\hat{\pi}}\|_\infty \leq \|Q^* - \hat{Q}^{\pi*}\|_\infty + \|\hat{Q}^* - \hat{Q}^{\hat{\pi}}\|_\infty + \|\hat{Q}^{\hat{\pi}} - Q^{\hat{\pi}}\|_\infty$$

$\square$

This decomposition is the starting point of our proofs for both Theorems 4.1 and 5.1. In this decomposition, the second term satisfies $\|Q^* - \hat{Q}^{\pi*}\|_\infty \leq \epsilon_{\text{opt}}$ by definition. This term goes to 0 exponentially fast as the robust Bellman operator is a $\gamma$-contraction. The two last terms $\|Q^* - \hat{Q}^{\pi*}\|_\infty$ and $\|\hat{Q}^{\hat{\pi}} - Q^{\hat{\pi}}\|_\infty$ need to be controlled using concentration inequalities between the true MDP and the estimated one. To do so, we need concentration inequalities such as the following Lemma B.2.

**Lemma B.2** (Hoeffding's inequality for $V$). *For any $V \in \mathbb{R}^{|\mathcal{S}|}$ with $\|V\|_\infty \leq H$, with probability at least $1 - \delta$, we have*

$$\max_{(s,a)} \left| P_0 V - \widehat{P} V \right| \leq H \sqrt{\frac{\log(2|\mathcal{S}||\mathcal{A}|/\delta)}{2N}}.$$

*Proof.* For any $(s, a)$ pair, assume a discrete random variable taking value $V(i)$ with probability $P_{0,s,a}(i)$ for all $i \in \{1, 2, \cdots, |\mathcal{S}|\}$. Using Hoeffding's inequality [Hoeffding, 1994] and $\|V\|_\infty \leq H$:

$$\mathbb{P}\left( P_0 V - \widehat{P} V \geq \varepsilon \right) \leq \exp\left(-N\varepsilon^2/(2H^2)\right) \quad \text{and} \quad \mathbb{P}\left( \widehat{P} V - P_0 V \geq \varepsilon \right) \leq \exp\left(-N\varepsilon^2/(2H^2)\right).$$

Then, taking $\varepsilon = H \sqrt{\frac{2\log(2|\mathcal{S}||\mathcal{A}|/\delta)}{N}}$, we get

$$\mathbb{P}\left( \left| P_0 V - \widehat{P} V \right| \geq H \sqrt{\frac{\log(2|\mathcal{S}||\mathcal{A}|/\delta)}{N}} \right) \leq \frac{\delta}{|\mathcal{S}||\mathcal{A}|}.$$

Finally, using a union bound:

$$\mathbb{P}\left( \max_{(s,a)} \left| P_0 V - \widehat{P} V \right| \geq H \sqrt{\frac{2\log(2|\mathcal{S}||\mathcal{A}|/\delta)}{N}} \right) \leq \sum_{s,a} \mathbb{P}\left( \left| P_0 V - \widehat{P} V \right| \geq H \sqrt{\frac{2\log(2|\mathcal{S}||\mathcal{A}|/\delta)}{N}} \right) \leq \delta.$$

$\square$

This completes the concentration proof. Next we will look at the contraction argument of the robust Bellman operator.

**Lemma B.3** (Contraction of infimum operator). *For $\mathcal{D} = \mathcal{P}_{s,a}$ or $\mathcal{P}_s$, the function*

$$\forall s, a, \quad v \mapsto \kappa_\mathcal{D}(v) = \inf\left\{ u^\top v : u \in \mathcal{D} \right\}$$

*is 1-Lipchitz.*

*Proof.* We have that

$$\forall (s,a) \in \mathcal{S} \times \mathcal{A}, \quad \kappa_{\mathcal{P}_{s,a}}(V_2) - \kappa_{\mathcal{P}_{s,a}}(V_1) = \inf_{p \in \mathcal{P}_{s,a}} p^\top V_2 - \inf_{\tilde{p} \in \mathcal{P}_{s,a}} \tilde{p}^\top V_1 = \inf_{p \in \mathcal{P}_{s,a}} \sup_{\tilde{p} \in \mathcal{P}_{s,a}} p^\top V_2 - \tilde{p}^\top V_1$$

$$\geq \inf_{p \in \mathcal{P}_{s,a}} p^\top (V_2 - V_1) = \kappa_{\mathcal{P}_{s,a}}(V_2 - V_1).$$

Then $\forall \varepsilon > 0$, there exists $P_{s,a} \in \mathcal{P}_{s,a}$ such that

$$P_{s,a}^\top (V_2 - V_1) - \varepsilon \leq \kappa_{\mathcal{P}_{s,a}}(V_2 - V_1).$$

Using those two properties,

$$\kappa_{\mathcal{P}_{s,a}}(V_1) - \kappa_{\mathcal{P}_{s,a}}(V_2) \le P_{s,a}^\top (V_1 - V_2) + \varepsilon \le \|P_{s,a}\|_1 \|V_1 - V_2\| + \varepsilon = \|V_1 - V_2\| + \varepsilon,$$

where we used the Holder's inequality. Since $\varepsilon$ is arbitrary small, we obtain, $\kappa_{\mathcal{P}_{s,a}}(V_1) - \kappa_{\mathcal{P}_{s,a}}(V_2) \le \|V_1 - V_2\|$. Exchanging the roles of $V_1$ and $V_2$ give the result.

The proof is similar for $\mathcal{P}_s$. $\qquad\square$

Note that an immediate consequence is the already known $\gamma$- contraction of the robust Bellman operator.

**Lemma B.4** (Upper-bounds of $\left\|Q^{\hat\pi} - \hat{Q}^{\hat\pi}\right\|_\infty$ and $\left\|Q^* - \hat{Q}^{\pi^*}\right\|_\infty$).

$$\left\|Q^{\hat\pi} - \hat{Q}^{\hat\pi}\right\|_\infty \le \frac{\gamma}{1-\gamma} \max_{s,a}\left|\kappa_{\hat{\mathcal{P}}_{0,s,a}}(\hat{V}^{\hat\pi}) - \kappa_{\mathcal{P}_{0,s,a}}(\hat{V}^{\hat\pi})\right|,$$

$$\left\|Q^* - \hat{Q}^{\pi^*}\right\|_\infty \le \frac{\gamma}{1-\gamma} \max_{s,a}\left|\kappa_{\hat{\mathcal{P}}_{s,a}}(V^*) - \kappa_{\mathcal{P}_{0,s,a}}(V^*)\right|.$$

*Proof.* For the first inequality, since we can rewrite the robust Q-function for any uncertainty sets on the dynamics as $Q^{\hat\pi}(s,a) = r - \alpha_{s,a} + \gamma\kappa_{\mathcal{P}_{0,s,a}}(V^{\hat\pi})$ (see Eq. (3.5)), or replacing $\alpha_{s,a}$ by $\alpha_s\left(\frac{\hat\pi_s(a)}{\|\hat\pi_s\|_q}\right)^{q-1}$ in the $s$- rectangular case:

$$Q^{\hat\pi}(s,a) - \hat{Q}^{\hat\pi}(s,a) \overset{(a)}{=} \gamma\kappa_{\mathcal{P}_{0,s,a}}(V^{\hat\pi}) - \gamma\kappa_{\hat{\mathcal{P}}_{s,a}}(\hat{V}^{\hat\pi})$$

$$= \gamma\left(\kappa_{\mathcal{P}_{0,s,a}}(V^{\hat\pi}) - \kappa_{\mathcal{P}_{0,s,a}}(\hat{V}^{\hat\pi})\right) + \gamma\left(\kappa_{\mathcal{P}_{0,s,a}}(\hat{V}^{\hat\pi}) - \kappa_{\hat{\mathcal{P}}_{s,a}}(\hat{V}^{\hat\pi})\right)$$

with $\mathcal{P}_{s,a}$ defined in Assumption 3.1 and $\hat{\mathcal{P}}_{s,a}$ with the same definition but centered around the empirical MDP. Hence, taking the supremum norm $\|.\|_\infty$,

$$\left\|Q^{\hat\pi} - \hat{Q}^{\hat\pi}\right\|_\infty = \max_{s,a}\left|\gamma\left(\kappa_{\mathcal{P}_{0,s,a}}(V^{\hat\pi}) - \kappa_{\mathcal{P}_{0,s,a}}(\hat{V}^{\hat\pi})\right) + \gamma\left(\kappa_{\mathcal{P}_{0,s,a}}(\hat{V}^{\hat\pi}) - \kappa_{\hat{\mathcal{P}}_{s,a}}(\hat{V}^{\hat\pi})\right)\right|$$

$$\overset{(b)}{\le} \gamma\left\|V^{\hat\pi} - \hat{V}^{\hat\pi}\right\|_\infty + \max_{s,a}\left|\gamma\left(\kappa_{\mathcal{P}_{0,s,a}}(\hat{V}^{\hat\pi}) - \kappa_{\hat{\mathcal{P}}_{s,a}}(\hat{V}^{\hat\pi})\right)\right|$$

$$\le \gamma\left\|V^{\hat\pi} - \hat{V}^{\hat\pi}\right\|_\infty + \gamma\max_{s,a}\left|\kappa_{\hat{\mathcal{P}}_{s,a}}(\hat{V}^{\hat\pi}) - \kappa_{\mathcal{P}_{0,s,a}}(\hat{V}^{\hat\pi})\right|$$

$$\overset{(c)}{\le} \gamma\left\|Q^{\hat\pi} - \hat{Q}^{\hat\pi}\right\|_\infty + \gamma\max_{s,a}\left|\kappa_{\hat{\mathcal{P}}_{s,a}}(\hat{V}^{\hat\pi}) - \kappa_{\mathcal{P}_{0,s,a}}(\hat{V}^{\hat\pi})\right|.$$

Line (a) comes from the rectangularity assumption, (b) uses the triangular inequality and the 1-contraction of the infimum in Lemma B.3, (c) uses the fact that $\|V^\pi - \widehat{V}^\pi\|_\infty \le \|Q^\pi - \widehat{Q}^\pi\|_\infty$ for any $\pi$. As $1 - \gamma < 1$, we get the first stated result.

One can note that the proof is true for any policy, so it is also true for both $\hat\pi$ and $\pi^*$ which concludes the proof. This proof is written for the $sa$-rectangular assumption, it is also true for the $s$-rectangular case with slightly different notations, replacing $\mathcal{D} = \mathcal{P}_{0,s,a}$ by $\mathcal{D} = \mathcal{P}_{0,s}$. Now we need to find new form for $\kappa$ for both $s$ and $sa$ rectangular assumptions.

For the second claim,

$$\left\|Q^* - \hat{Q}^{\pi^*}\right\|_\infty \le \frac{\gamma}{1-\gamma} \max_{s,a}\left|\kappa_{\hat{\mathcal{P}}_{s,a}}(V^*) - \kappa_{\mathcal{P}_{0,s,a}}(V^*)\right|.$$

we are using a slightly different modification:

$$Q^*(s,a) - \hat{Q}^{\pi^*}(s,a) \overset{(a)}{=} \gamma\kappa_{\mathcal{P}_{0,s,a}}(V^*) - \gamma\kappa_{\hat{\mathcal{P}}_{s,a}}\left(\hat{V}^{\pi^*}\right)$$

$$= \gamma\kappa_{\mathcal{P}_{0,s,a}}(V^*) - \gamma\kappa_{\mathcal{P}_{0,s,a}}\left(\hat{V}^{\pi^*}\right) + \gamma\kappa_{\mathcal{P}_{0,s,a}}\left(\hat{V}^{\pi^*}\right) - \gamma\kappa_{\hat{\mathcal{P}}_{s,a}}\left(\hat{V}^{\pi^*}\right)$$

$$\le \gamma\left\|Q^* - \hat{Q}^{\pi^*}\right\|_\infty + \max_{s,a}\left|\kappa_{\hat{\mathcal{P}}_{s,a}}(V^*) - \kappa_{\mathcal{P}_{0,s,a}}(V^*)\right|$$

using the same arguments as in the first inequality. Solving gives the result. $\qquad\square$

We denote $[V]_\alpha$ as its clipped version by some non-negative vector $\alpha$, namely,

$$[V]_\alpha(s) := \begin{cases} \alpha(s), & \text{if } V(s) > \alpha(s), \\ V(s), & \text{otherwise.} \end{cases} \tag{7}$$

Defining the gradient of $P \mapsto \|P\|$ as $\nabla \|P\|$, $\lambda > 0$, a positive scalar and $\omega$ is the generalized mean defined as the argmin in the definition of the span semi norm in Def.3.1, we derive two optimization lemmas.

**Lemma B.5** (Duality for the minimization problem for $sa$ rectangular case.). *Denoting $\widehat{P}$ the vector $\widehat{P}_{s,a}$ or $P_0$ for $P_{0,s,a}$ ,*

$$\kappa_{\widehat{\mathcal{P}}_{s,a}}(\hat{V}^{\hat\pi}) = \max_{\mu \geq 0}\{\widehat{P}(\hat{V}^{\hat\pi} - \mu) - \beta_{s,a}\mathrm{sp}_q(\hat{V}^{\hat\pi} - \mu)\} = \max_{\mu_{\widehat{P}}^{\lambda,\omega} \in \mathcal{M}_{\widehat{P}}^{\lambda,\omega}}\{\widehat{P}(\hat{V}^{\hat\pi} - \mu_{\widehat{P}}^{\lambda,\omega}) - \beta_{s,a}\mathrm{sp}_q(\hat{V}^{\hat\pi} - \mu_{\widehat{P}}^{\lambda,\omega})\}$$

$$= \max_{\alpha_{\widehat{P}}^{\lambda,\omega} \in \mathrm{A}_{\widehat{P}}^{\lambda,\omega}} \widehat{P}[\hat{V}^{\hat\pi}]_{\alpha_{\widehat{P}}^{\lambda,\omega}} - \beta_{s,a}\mathrm{sp}_q([\hat{V}^{\hat\pi}]_{\alpha_{\widehat{P}}^{\lambda,\omega}}).$$

$$\kappa_{\mathcal{P}_{0,s,a}}(V^*) = \max_{\mu \geq 0}\{P_0(V^* - \mu) - \beta_{s,a}\mathrm{sp}_q(V^* - \mu)\} = \max_{\mu_{P_0}^{\lambda,\omega} \in \mu_{P_0}^{\lambda,\omega}}\{P_0(V^* - \mu_{P_0}^{\lambda,\omega}) - \beta_{s,a}\mathrm{sp}_q(V^* - \mu_{P_0}^{\lambda,\omega})\}$$

$$= \max_{\alpha_{P_0}^{\lambda,\omega} \in \mathrm{A}_{P_0}^{\lambda,\omega}} P_0[V^*]_{\alpha_{P_0}^{\lambda,\omega}} - \beta_{s,a}\mathrm{sp}_q([V^*]_{\alpha_{P_0}^{\lambda,\omega}}).$$

*where*

$$\mathrm{A}_P^{\lambda,\omega} = \{\alpha_P^{\lambda,\omega} : \alpha_P^{\lambda,\omega}(s) = \omega + \lambda|\nabla \|P\|\,|(s) : \lambda > 0, w > 0, P \in \Delta(S), \alpha_P^{\lambda,\omega} \in \left[0, \frac{1}{1-\gamma}\right]^S\} \tag{8}$$

$$\mathcal{M}_P^{\lambda,\omega} = \{\mu_P^{\lambda,\omega} = V - \alpha_P^{\lambda,\omega}, \lambda, \omega \in \mathbb{R}^+, P \in \Delta(S), \mu \in \mathbb{R}_+^S, \mu_P^{\lambda,\omega} = \left[0, \frac{1}{1-\gamma}\right]^S\} \tag{9}$$

$$\tag{10}$$

and with $[V]_\alpha := \begin{cases} \alpha(s), & \text{if } V(s) > \alpha(s) \\ V(s), & \text{otherwise.} \end{cases}$

For $L_1$ or $TV$, case , the vector $\alpha_P^{\lambda,\omega}$ reduces to a 1 dimensional scalar such as $\alpha \in [0, 1/(1-\gamma)]$.

*Proof.* First, we will show that

$$\kappa_{\widehat{\mathcal{P}}_{s,a}}(\hat{V}^{\hat\pi}) = \max_{\mu \geq 0}\{\widehat{P}(\hat{V}^{\hat\pi} - \mu) - \beta_{s,a}\mathrm{sp}_q(\hat{V}^{\hat\pi} - \mu)\}$$

The second equation of this lemma is the same as the first one, replacing the center of the ball constrain $\widehat{P}_{s,a}$ by $P_{0,s,a}$ and $\hat\pi$ by $\pi^*$. By definition,

$$\kappa_{\widehat{\mathcal{P}}_{s,a}}(\hat{V}^{\hat\pi}) = \min_{P \in \Delta_s, \|P - \widehat{P}\|_p \leq \beta_{s,a}} \sum_{s'} P(s')\hat{V}^{\hat\pi}(s') = \widehat{P}_{s,a}\hat{V}^{\hat\pi} + \min_{y, \|y\|_p \leq \beta_{s,a}, 1y=0, y \geq -\hat{P}} \sum_{s'} y(s')\hat{V}^{\hat\pi}(s')$$

where we use the change of variable $y(s') = P(s') - \hat{P}(s')$. Then writing the Lagrangian we get for $\mu \in \mathbb{R}_+^{|S|}, \gamma \in \mathbb{R}$ the Lagrangian variables:

$$\widehat{P}\hat{V}^{\hat\pi} + \max_{\mu \geq 0, \nu \in \mathbb{R}} \min_{y : \|y\|_p \leq \beta_{s,a}} -\sum_{s'} \mu(s)\hat{P}(s') + \sum_{s'}(y(s')(\hat{V}^{\hat\pi}(s') - \mu(s') - \nu) \tag{11}$$

$$\overset{(a)}{=} \widehat{P}\hat{V}^{\hat\pi} + \max_{\mu \geq 0, \nu \in \mathbb{R}} -\sum_{s'} \mu(s')\hat{P}(s') - \beta_{s,a}\left\|(\hat{V}^{\hat\pi}(s') - \mu(s') - \nu)\right\|_q \tag{12}$$

$$\overset{(b)}{=} \max_{\mu \geq 0} \widehat{P}(\hat{V}^{\hat\pi} - \mu) - \beta_{s,a}\mathrm{sp}_q(\hat{V}^{\hat\pi} - \mu) \tag{13}$$

where (a) is true using the equality case of Holder's inequality and (b) is the definition of the span semi-norm (see Def. 3.1). The value that maximizes the inner maximization problem in 12 in $\nu$ is the $q$-mean (see Def. 3.1) by definition denoted $\omega$. Now the aim is to prove that

$$\max_{\mu \geq 0}\{\widehat{P}(\hat{V}^{\hat{\pi}} - \mu) - \beta_{s,a}\mathrm{sp}_q(\hat{V}^{\hat{\pi}} - \mu)\} = \max_{\mu_{\widehat{P}}^{\lambda,\omega} \in \mathcal{M}_{\widehat{P}}^{\lambda,\omega}} \{\widehat{P}(\hat{V}^{\hat{\pi}} - \mu_{\widehat{P}}^{\lambda,\omega}) - \beta_{s,a}\mathrm{sp}_q(\hat{V}^{\hat{\pi}} - \mu_{\widehat{P}}^{\lambda,\omega})\}.$$

First, as the norm is differentiable (which true for $L_p$, $p \geq 2$), we have that the equality (a) comes from the generalized Holder's inequality for arbitrary norms Yang [1991], namely, defining $z = (\hat{V}^{\hat{\pi}} - \mu - \omega)$, it satisfies

$$z = \|z\|_q \nabla \|y\|_p \tag{14}$$

The quantity $\nu$ is replaced by the generalized mean for equality in (b) while (14) comes from Yang [1991]. Using complementary slackness Karush [2013]we define $\mathcal{B} = \{s \in \mathcal{S} : \mu(s) > 0\}$

$$\forall s \in \mathcal{B}: \quad y^*(s) = -\widehat{P}(s), \tag{15}$$

which leads to the following equality by plugging the previous (15) in (14) and defining $z^* = \hat{V}^{\hat{\pi}} - \mu^* - \omega$:

$$\forall s \in \mathcal{B}, \quad z^*(s) = \|z^*\|_q \nabla \left\|\widehat{P}\right\|_p (s) \tag{16}$$

or

$$\forall s \in \mathcal{B}, \quad \hat{V}^{\hat{\pi}}(s) - \mu^*(s) = \omega + \lambda\nabla \left\|\widehat{P}\right\|_p (s)\hat{=}\alpha_{\widehat{P}}^{\lambda,\omega} \tag{17}$$

by letting $\lambda = \|z^*\|_q \in \mathbb{R}^+$ . Note that for $s \in \mathcal{B}$, $\nabla \|y\|_p = \nabla \|P\|_p$ only depends on $P(s)$ and not on other coordinates due to definition of $L_p$ norm.

We can remark that $v - \mu^*$ is $P$ dependent, but if $P$ is known, the best $\mu^*$ is only determined by one 2 dimensional parameters $\lambda = \|v - \mu^* - \nu\|_q$ and $\omega \in \mathbb{R}^+$. Moreover, when $\widehat{P}$ is fixed, the scalar $\omega$ is a constant is fully determined by $P$, $v$ and $\mu^*$. This is why the quantity defined $\alpha_{\widehat{P}}^\lambda$ varies through 2 parameter $\lambda$ and $\omega$. Given this observation, we can rewrite the optimization problem as :

$$\max_{\mu \geq 0}\{\widehat{P}(\hat{V}^{\hat{\pi}} - \mu) - \beta_{s,a}\mathrm{sp}_q(\hat{V}^{\hat{\pi}} - \mu)\} = \max_{\mu_{\widehat{P}}^{\lambda,\omega} \in \mathcal{M}_{\widehat{P}}^{\lambda,\omega}} \{\widehat{P}(\hat{V}^{\hat{\pi}} - \mu_{\widehat{P}}^{\lambda,\omega}) - \beta_{s,a}\mathrm{sp}_q(\hat{V}^{\hat{\pi}} - \mu_{\widehat{P}}^{\lambda,\omega})\} \tag{18}$$

$$= \max_{\alpha_{\widehat{P}}^{\lambda,\omega} \in \mathrm{A}_{\widehat{P}}^{\lambda,\omega}} \widehat{P}[\hat{V}^{\hat{\pi}}]_{\alpha_{\widehat{P}}^{\lambda,\omega}} - \beta_{s,a}\mathrm{sp}_q([\hat{V}^{\hat{\pi}}]_{\alpha_{\widehat{P}}^{\lambda,\omega}}) \tag{19}$$

where we defined the maximization problem on $\mu$ not in $\mathbb{R}^S$ but at the optimal in the variational family denote $\mathcal{M}_P^{\lambda,\omega} = \{\mu_P^{\lambda,\omega} = \hat{V}^{\hat{\pi}} - \alpha_P^{\lambda,\omega}, \lambda, \omega \in \mathbb{R}^+, P \in \Delta(S), \mu \in \mathbb{R}_+^S, \mu_P^{\lambda,\omega} = \left[0, \frac{1}{1-\gamma}\right]^S\}$.

We can rewrite the optimization problem in terms of $\alpha_P$ with

$$[V]_{\alpha_{\widehat{P}}^{\lambda,\omega}}(s) := \begin{cases} \alpha_{\widehat{P}}^{\lambda,\omega}, \text{if } V(s) \geq \alpha_{\widehat{P}}^{\lambda,\omega} \\ V(s), \quad\quad\quad\quad\quad \text{otherwise.} \end{cases}$$

Note that for $TV$ or $L_1$, this lemma holds, but the vector $\alpha_{\widehat{P}}^{\lambda,\omega}$ reduces to a positive scalar denoted $\alpha$ which is equal to $\left\|\hat{V}^{\hat{\pi}} - \mu^*\right\|_\infty$ according to Iyengar [2005]. The thing which is of capital importance is that the second part of the equation $\mathrm{sp}_q([\hat{V}^{\hat{\pi}}]_\alpha)$ does not depend on $\widehat{P}$.

$\square$

**Lemma B.6** (Duality for the minimization problem for $s$ rectangular case.). *Considering a projection matrix associated with a given policy $\pi$ such that $P_s^\pi(s') = \sum_a \pi(a|s) P_{s,a}(s')$ and denoting $\widehat{P}^\pi \in \mathbb{R}^s$ the vector $\widehat{P}_s^\pi(.)$ or $P_0^\pi$ for $P_{0,s}^\pi(.)$, we have:*

$$\kappa_{\widehat{\mathcal{P}}_s}(\hat{V}^{\hat{\pi}}) = \sum_a \hat{\pi}(a|s) \max_{\alpha_{\hat{P}_{s,a}}^{\lambda,\omega} \in A_{\hat{P}_{s,a}}^{\lambda,\omega}} \left( \left( \hat{P}_{s,a}[\hat{V}^{\hat{\pi}}]_{\alpha_{\hat{P}_{s,a}}^{\lambda,\omega}} - \beta_s \|\pi_s\|_q \operatorname{sp}_q([\hat{V}^{\hat{\pi}}]_{\alpha_{\hat{P}_{s,a}}^{\lambda,\omega}}) \right) \right)$$

$$\kappa_{\mathcal{P}_{0,s}}(V^*) = \sum_a \pi(a|s) \max_{\alpha_{P_{0,s,a}}^{\lambda,\omega} \in A_{P_{0,s,a}}^{\lambda,\omega}} \left( \left( P_{0,s,a}[V^*]_{\alpha_{P_{0,s,a}}^{\lambda,\omega}} - \beta_s \|\pi_s\|_q \operatorname{sp}_q([V^*]_{\alpha_{P_{0,s,a}}^{\lambda,\omega}}) \right) \right)$$

*with* $[V]_\alpha(s) := \begin{cases} \alpha(s), & \text{if } V(s) > \alpha \\ V(s), & \text{otherwise.} \end{cases}$

*Proof.* The second equation is the same replacing the center of the ball constrain $\widehat{P}_s^\pi$ by $P_0^\pi$ and $\hat{\pi}$ by $\pi^*$. By definition,

$$\kappa_{\widehat{\mathcal{P}}_s}(\hat{V}^{\hat{\pi}})(s) = \min_{P_s^{\hat{\pi}} \in (\Delta_s), P_s^{\hat{\pi}} \in \hat{\mathcal{P}}_s} P_s^{\hat{\pi}} \hat{V}^{\hat{\pi}}(s)$$

$$\overset{(a)}{=} \sum_a \hat{\pi}(a|s) \hat{P}_{s,a} \hat{V}^{\hat{\pi}} + \min_{\|\beta_{s,a}\|_p \leq \beta_s} \sum_a \hat{\pi}(a|s) \min_{y, \|y\|_p \leq \beta_{s,a}, \mathbf{1}y=0, y \geq -\hat{P}} \sum_{s'} y(s') \hat{V}^{\hat{\pi}}$$

where we use the change of variable $y(s') = P_{s,a}(s') - \hat{P}_{s,a}(s')$ in (a). Then we case use the previous lemma for $sa$ rectangular assumption, Lemma 3.3. Then,

$$\min_{\|\beta_{s,a}\|_p \leq \beta_s} \sum_a \hat{\pi}(a|s) \min_{y, \|y\|_p \leq \beta_{s,a}, \mathbf{1}y=0, y \geq -\hat{P}_{s,a}} \sum_{s'} y(s') \hat{V}^{\hat{\pi}} = \min_{\|\beta_{s,a}\|_p \leq \beta_s} \sum_a \hat{\pi}(a|s) \max_{\mu \geq 0} \left( -\widehat{P}_{s,a}\mu - \beta_{s,a} \operatorname{sp}_q(\hat{V}^{\hat{\pi}} - \mu) \right)$$

$$= \sum_a \max_{\mu \geq 0} \left( \hat{\pi}(a|s)(-\widehat{P}_{s,a}\mu) - \max_{\|\beta_{s,a}\|_p \leq \beta_s} \sum_a \hat{\pi}(a|s) \beta_{s,a} \operatorname{sp}_q(\hat{V}^{\hat{\pi}} - \mu) \right)$$

$$= \sum_a \max_{\mu \geq 0} \left( \hat{\pi}(a|s)(-\widehat{P}_{s,a}\mu) - \beta_s \|\pi_s\|_q \operatorname{sp}_q(\hat{V}^{\hat{\pi}} - \mu) \right)$$

we can exchange the min and the max as we get concave-convex problems in $\beta_{s,a}$ and $\mu$, ([von Neumann, 1928]) in the second line and using Holder's inequality in the last line. Finally, we obtain:

$$\kappa_{\widehat{\mathcal{P}}_s}(\hat{V}^{\hat{\pi}}) = \sum_a \max_{\mu \geq 0} \left( \hat{\pi}(a|s)(\hat{P}_{s,a}(\hat{V}^{\hat{\pi}} - \mu) - \beta_s \|\pi_s\|_q \operatorname{sp}_q(\hat{V}^{\hat{\pi}} - \mu)) \right)$$

$$\overset{(a)}{=} \sum_a \hat{\pi}(a|s) \max_{\alpha_{\hat{P}_{s,a}}^{\lambda,\omega} \in A_{\hat{P}_{s,a}}^{\lambda,\omega}} \left( \left( \hat{P}_{s,a}[\hat{V}^{\hat{\pi}}]_{\alpha_{\hat{P}_{s,a}}^{\lambda,\omega}} - \beta_s \|\pi_s\|_q \operatorname{sp}_q([\hat{V}^{\hat{\pi}}]_{\alpha_{\hat{P}_{s,a}}^{\lambda,\omega}}) \right) \right)$$

where in (a) we use Lemma 3.3. Second claim is the same replacing $\hat{V}^{\hat{\pi}}$ by $V^*$, $\hat{\pi}$ by $\pi^*$ and $\hat{P}$ by $P_0$. Then we derive a new decomposition of the difference the two minimum.

$\square$

**Lemma B.7.** *For s and sa rectangular assumptions,*

$$\left|\kappa_{\hat{\mathcal{P}}_{s,a}}(\hat{V}^{\hat{\pi}}) - \kappa_{\mathcal{P}_{0,s,a}}(\hat{V}^{\hat{\pi}})\right| \leq \max\left\{ \underbrace{\max_{s,a}\left|\max_{\mu\in\mu^{\lambda,\omega}_{P_{0,s,a}}} \left(P_{0,s,a} - \widehat{P}_{0,s,a}\right)(\hat{V}^{\hat{\pi}} - \mu^{\lambda,\omega}_{P_{0,s,a}})\right|}_{=:g_{s,a}(\alpha^{\lambda,\omega}_P,\hat{V}^{\hat{\pi}})}, \right. \tag{20}$$

$$\left. \underbrace{\max_{s,a}\left|\max_{\mu^{\lambda,\omega}_{\hat{P}_{0,s,a}}\in\mathcal{M}^{\lambda,\omega}_{\hat{P}_{0,s,a}}} \left(P_{0,s,a} - \widehat{P}_{0,s,a}\right)(\hat{V}^{\hat{\pi}} - \mu^{\lambda,\omega}_{\hat{P}_{0,s,a}})\right|}_{=:g_{s,a}(\alpha^{\lambda,\omega}_{\hat{P}},\hat{V}^{\hat{\pi}})} \right\} \tag{21}$$

$$\left|\kappa_{\hat{\mathcal{P}}_{s}}(V^*) - \kappa_{\mathcal{P}_{0,s}}(V^*)\right| \leq \max\left\{ \underbrace{\max_{s,a}\left|\max_{\mu\in\mu^{\lambda,\omega}_{P_{0,s,a}}} \left(P_{0,s,a} - \widehat{P}_{0,s,a}\right)(V^* - \mu^{\lambda,\omega}_{P_{0,s,a}})\right|}_{=:g_{s,a}(\alpha^{\lambda,\omega}_P,V^*)}, \right. \tag{22}$$

$$\left. \underbrace{\max_{s,a}\left|\max_{\mu^{\lambda,\omega}_{\hat{P}_{0,s,a}}\in\mathcal{M}^{\lambda,\omega}_{\hat{P}_{0,s,a}}} \left(P_{0,s,a} - \widehat{P}_{0,s,a}\right)(V^* - \mu^{\lambda,\omega}_{\hat{P}_{0,s,a}})\right|}_{=:g_{s,a}(\alpha^{\lambda,\omega}_{\hat{P}},V^*)} \right\} \tag{23}$$

*Proof.*

$$\left|\kappa_{\hat{\mathcal{P}}_{s,a}}(V^*) - \kappa_{\mathcal{P}_{0,s,a}}(V^*)\right| \tag{24}$$

$$= \left| \max_{\mu^{\lambda,\omega}_{P_{0,s,a}}\in\mathcal{M}^{\lambda,\omega}_{P_{0,s,a}}} \left\{P_{0,s,a}(V^* - \mu) - \beta_{s,a}\left(\mathrm{sp}_q((V^* - \mu))\right)\right\} \right.$$

$$\left. - \max_{\mu^{\lambda,\omega}_{\hat{P}_{0,s,a}}\in\mathcal{M}^{\lambda,\omega}_{\hat{P}_{0,s,a}}} \left\{\widehat{P}_{0,s,a}(V^* - \mu^{\lambda,\omega}_{\hat{P}_{0,s,a}}) - \beta_{s,a}\left(\mathrm{sp}_q((V^* - \mu^{\lambda,\omega}_{\hat{P}_{0,s,a}}))\right)\right\} \right|$$

$$\leq \max\left\{ \left| \max_{\mu^{\lambda,\omega}_{P_{0,s,a}}\in\mathcal{M}^{\lambda,\omega}_{P_{0,s,a}}} \left\{P_{0,s,a}(V^* - \mu^{\lambda,\omega}_{P_{0,s,a}}) - \beta_{s,a}\left(\mathrm{sp}_q((V^* - \mu^{\lambda,\omega}_{P_{0,s,a}}))\right)\right\} \right.\right.$$

$$\left.\left. - \max_{\mu^{\lambda,\omega}_{P_{0,s,a}}\in\mathcal{M}^{\lambda,\omega}_{P_{0,s,a}}} \left\{\widehat{P}_{0,s,a}(V^* - \mu^{\lambda,\omega}_{P_{0,s,a}}) - \beta_{s,a}\left(\mathrm{sp}_q((V^* - \mu^{\lambda,\omega}_{P_{0,s,a}}))\right)\right\} \right| ; \right. \tag{25}$$

$$\left| \max_{\mu^{\lambda,\omega}_{\hat{P}_{0,s,a}}\in\mathcal{M}^{\lambda,\omega}_{\hat{P}_{0,s,a}}} \left\{\widehat{P}_{0,s,a}(V^* - \mu^{\lambda,\omega}_{\hat{P}_{0,s,a}}) - \beta_{s,a}\left(\mathrm{sp}_q((V^* - \mu^{\lambda,\omega}_{\hat{P}_{0,s,a}}))\right)\right\} \right. \tag{26}$$

$$\left.\left. - \max_{\mu^{\lambda,\omega}_{\hat{P}_{0,s,a}}\in\mathcal{M}^{\lambda,\omega}_{\hat{P}_{0,s,a}}} \left\{P_{0,s,a}(V^* - \mu^{\lambda,\omega}_{\hat{P}_{0,s,a}}) - \beta_{s,a}\left(\mathrm{sp}_q((V^* - \mu^{\lambda,\omega}_{\hat{P}_{0,s,a}}))\right)\right\} \right| \right\}$$

$$\leq \max\left\{ \underbrace{\left|\max_{\mu\in\mu^{\lambda,\omega}_{P_{0,s,a}}} \left(P_{0,s,a} - \widehat{P}_{0,s,a}\right)(V^* - \mu^{\lambda,\omega}_{P_{0,s,a}})\right|}_{=:g_{s,a}(\alpha^{\lambda,\omega}_P,V^*)}, \underbrace{\left|\max_{\mu^{\lambda,\omega}_{\hat{P}_{0,s,a}}\in\mathcal{M}^{\lambda,\omega}_{\hat{P}_{0,s,a}}} \left(P_{0,s,a} - \widehat{P}_{0,s,a}\right)(V^* - \mu^{\lambda,\omega}_{\hat{P}_{0,s,a}})\right|}_{=:g_{s,a}(\alpha^{\lambda,\omega}_{\hat{P}},V^*)} \right\} \tag{27}$$

where in the first equality we use Lemma B.5. The final inequality is a consequence of the 1-Lipschitzness of the max operator. Taking the supremum over $s, a$ gives the result. Replacing $V^*$ by $\hat{V}^{\hat{\pi}}$ gives the other inequality. The result for $s$

rectangular are the same as

$$\sum_a \pi(a|s) \max\left\{ \underbrace{\left| \max_{\mu \in \mu^{\lambda,\omega}_{P_{0,s,a}}} \left(P_{0,s,a} - \widehat{P}_{0,s,a}\right)(V^* - \mu^{\lambda,\omega}_{P_{0,s,a}}) \right|}_{=:g_{s,a}(\alpha^{\lambda,\omega}_P, V^*)}, \underbrace{\left| \max_{\mu^{\lambda,\omega}_{\hat{P}_{0,s,a}} \in \mathcal{M}^{\lambda,\omega}_{\hat{P}_{0,s,a}}} \left(P_{0,s,a} - \widehat{P}_{0,s,a}\right)(V^* - \mu^{\lambda,\omega}_{\hat{P}_{0,s,a}}) \right|}_{=:g_{s,a}(\alpha^{\lambda,\omega}_{\hat{P}}, V^*)} \right\}$$

(28)

$$\leq \max\left\{ \max_{s,a} \underbrace{\left| \max_{\mu \in \mu^{\lambda,\omega}_{P_{0,s,a}}} \left(P_{0,s,a} - \widehat{P}_{0,s,a}\right)(V^* - \mu^{\lambda,\omega}_{P_{0,s,a}}) \right|}_{=:g_{s,a}(\alpha^{\lambda,\omega}_P, V^*)}, \max_{s,a} \underbrace{\left| \max_{\mu^{\lambda,\omega}_{\hat{P}_{0,s,a}} \in \mathcal{M}^{\lambda,\omega}_{\hat{P}_{0,s,a}}} \left(P_{0,s,a} - \widehat{P}_{0,s,a}\right)(V^* - \mu^{\lambda,\omega}_{\hat{P}_{0,s,a}}) \right|}_{=:g_{s,a}(\alpha^{\lambda,\omega}_{\hat{P}}, V^*)} \right\}$$

(29)

Note that at this point, quantities for $s$ and $sa$ rectangular is the same as the part with span semi norms cancelled. Now, note that the main problem is that we can not apply classical Hoeffding's inequality as $\widehat{P}$ is dependent of data as $\widehat{V}^{\hat{\pi}}$. We need to decouple $\widehat{V}^{\hat{\pi}}$ using $s$ absorbing MDPS as in Agarwal et al. [2020] but using Hoeffding arguments. First, we will use a concentration for $V^*$.

$\square$

**Lemma B.8.** *For $sa$ and $s$-rectangular, with probability $1 - \delta$, it holds:*

$$\left| \kappa_{\hat{\mathcal{P}}_{s,a}}(V^*) - \kappa_{\mathcal{P}_{0,s,a}}(V^*) \right| \leq 2\sqrt{\frac{L}{2N(1-\gamma)^2}} + \frac{2L|S|^{1/q} \|1_S\|_q (p-1)}{N(1-\gamma)}$$

*with $L = \log(18 \|1\|_q SAN/\delta)$*

*Proof.* First, we can use previous Lemma B.7

$$\left| \kappa_{\hat{\mathcal{P}}_{s,a}}(V^*) - \kappa_{\mathcal{P}_{0,s,a}}(V^*) \right|$$

(30)

$$\leq \max\left\{ \underbrace{\left| \max_{\mu \in \mu^{\lambda,\omega}_{P_{0,s,a}}} \left(P_{0,s,a} - \widehat{P}_{0,s,a}\right)(V^* - \mu^{\lambda,\omega}_{P_{0,s,a}}) \right|}_{=:g_{s,a}(\alpha^{\lambda,\omega}_P, V^*)}, \underbrace{\left| \max_{\mu^{\lambda,\omega}_{\hat{P}_{0,s,a}} \in \mathcal{M}^{\lambda,\omega}_{\hat{P}_{0,s,a}}} \left(P_{0,s,a} - \widehat{P}_{0,s,a}\right)(V^* - \mu^{\lambda,\omega}_{\hat{P}_{0,s,a}}) \right|}_{=:g_{s,a}(\alpha^{\lambda,\omega}_{\hat{P}}, V^*)} \right\}$$

(31)

First, we control $g_{s,a}(\alpha^{\lambda,\omega}_P, V^*)$. To do so, we use for a fixed $\alpha^{\lambda,\omega}_P$ and any vector $V^*$ that is independent with $\widehat{P}^0$, the Hoeffding's inequality, one has with probability at least $1 - \delta$ with $sa$-rectangular notations,

$$g_{s,a}(\alpha^{\lambda,\omega}_P, V^*) = \left| \left(P_{0,s,a} - \widehat{P}_{0,s,a}\right)[V^*]_{\alpha^{\lambda,\omega}_P} \right| \leq \sqrt{\frac{\log(\frac{2}{\delta})}{(1-\gamma)^2 2N}}$$

(32)

Once pointwise concentration derived, we will use uniform concentration to yield this lemma. First, union bound, is obtained noticing that $g_{s,a}(\alpha^{\lambda,\omega}_P, V^*)$ is 1-Lipschitz w.r.t. $\lambda$ and $\omega$ as it is linear in $\lambda$ and $\omega$. Moreover, $\lambda^* = \|V^* - \mu^* - \omega\|_q$ obeying $\lambda^* \leq \frac{\|1\|_q}{1-\gamma}$. The quantity $\omega \in [0, 1/(1-\gamma)]$ as it is always smaller that $V^*$ by definition. We construct then a 2-dimensional a $\varepsilon_1$-net $N_{\varepsilon_1}$ over $\lambda^* \in [0, \frac{\|1\|_q}{1-\gamma}]$ and $\omega \in [0, 1/(1-\gamma)]$ whose size satisfies $|N_{\varepsilon_1}| \leq \left(\frac{3\|1\|_q}{\varepsilon_1(1-\gamma)}\right)^2$ [Vershynin, 2017]. Using union bound and (32), it holds with probability at least $1 - \frac{\delta}{SA}$ that for all $\lambda \in N_{\varepsilon_1}$,

$$g_{s,a}(\alpha^{\lambda}_P, V^*) \leq \sqrt{\frac{2\log(\frac{SA|N_{\varepsilon_1}|}{\delta})}{2N(1-\gamma)^2}}.$$

(33)

Using the previous equation and also (27), it results in using notation $\log(\frac{18SAN}{\delta}) = L$,

$$g_{s,a}(\alpha_P^\lambda, V^*) \overset{(a)}{\leq} \sup_{\alpha_P^\lambda \in N_{\varepsilon_1}} \left| \left( P_{0,s,a} - \widehat{P}_{0,s,a} \right) [V^*]_{\alpha_P^\lambda} \right| + \varepsilon_1$$

$$\overset{(b)}{\leq} \sqrt{\frac{\log(\frac{SA|N_{\varepsilon_1}|}{\delta})}{2(1-\gamma)^2 N}} + \varepsilon_1 \tag{34}$$

$$\overset{(c)}{\leq} \sqrt{\frac{\log(\frac{2SA|N_{\varepsilon_1}|}{\delta})}{2N(1-\gamma)^2}} + \frac{\log(\frac{2SA|N_{\varepsilon_1}|}{\delta})}{3N(1-\gamma)}$$

$$\overset{(d)}{\leq} \sqrt{\frac{L}{2N(1-\gamma)2}} + \frac{L}{3N(1-\gamma)}$$

$$\leq 2\sqrt{\frac{L}{2(1-\gamma)^2 N}} \tag{35}$$

where (a) is because the optimal $\alpha^*$ falls into the $\varepsilon_1$-ball centered around some point inside $N_{\varepsilon_1}$ and $g_{s,a}(\alpha_P^\lambda, V^*)$ is 1-Lipschitz with regard to $\lambda$ and $\omega$, (b) is due to Eq. (33), (c) arises from taking $\varepsilon_1 = \frac{\log(\frac{2SA|N_{\varepsilon_1}|}{\delta})}{3N(1-\gamma)}$ , (d) is verified by $|N_{\varepsilon_1}| \leq \left(\frac{3\|1\|_q}{\varepsilon_1(1-\gamma)}\right)^2 \leq 9N \|1\|_q$ and that variance of a ceiling function of a vector is smaller than the variance of non-ceiling vector.

For $L_p$ with $p \geq 2$, contrary to the previous term, the second term $g_{s,a}(\alpha_{\widehat{P}}^\lambda, V)$ is more difficult as we need concentration, but there is an extra dependency in the data thought the parameter $\alpha_{\widehat{P}}^\lambda$. Note that this term does not exist as $\alpha$ is a constant for $TV$. We need to decouple this problem using absorbing MDPs. Then it leads to

$$g_{s,a}(\alpha_{\widehat{P}}^{\lambda,\omega}, V^*) \tag{36}$$

$$= |\max_{\mu_{\widehat{P}_{0,s,a}}^{\lambda,\omega} \in \mathcal{M}_{\widehat{P}_{0,s,a}}^{\lambda,\omega}} \left( P_{0,s,a} - \widehat{P}_{0,s,a} \right) (V^* - \mu_{\widehat{P}_{0,s,a}}^{\lambda,\omega})| \tag{37}$$

$$= |\max_{\mu \in \mathcal{M}_{\widehat{P}_{0,s,a}}^{\lambda,\omega}} \left( P_{0,s,a} - \widehat{P}_{0,s,a} \right) (V^* - \mu_{P_{0,s,a}}^{\lambda,\omega}) + \left( P_{0,s,a} - \widehat{P}_{0,s,a} \right) (\mu_{P_{0,s,a}}^{\lambda,\omega} - \mu_{\widehat{P}_{0,s,a}}^{\lambda,\omega})| \tag{38}$$

$$\leq |\max_{\mu_{P_{0,s,a}}^{\lambda,\omega} \in \mathcal{M}_{P_{0,s,a}}^{\lambda,\omega}} \left( P_{0,s,a} - \widehat{P}_{0,s,a} \right) (V^* - \mu_{P_{0,s,a}}^{\lambda,\omega}) + \max_{\mu_{\widehat{P}_{0,s,a}}^{\lambda,\omega} \in \mathcal{M}_{\widehat{P}_{0,s,a}}^{\lambda,\omega}} \left( P_{0,s,a} - \widehat{P}_{0,s,a} \right) (\mu_{P_{0,s,a}}^{\lambda,\omega} - \mu_{\widehat{P}_{0,s,a}}^{\lambda,\omega})| \tag{39}$$

In the first equality, we add the term $\mu_{P_{0,s,a}}^{\lambda,\omega}$ to retrieve the previous concentration problem, fixing $P_{0,s,a}$ and optimizing $\lambda, \omega$. In the second, we extend the max using triangular inequality. The first term in the last equality is exactly the term we have controlled previously, while the second one needs more attention. We decouple the dependency of the data, and then controlling the difference between the $\mu$. Then using the characterization of the optimal $\mu$ from equation (17):

$$\left( P_{0,s,a} - \widehat{P}_{0,s,a} \right) (\mu_{P_{0,s,a}}^{\lambda,\omega} - \mu_{\widehat{P}_{0,s,a}}^{\lambda,\omega}) = \sum_{s'} \lambda \left( P_{0,s,a}(s') - \widehat{P}_{0,s,a}(s') \right) (\nabla \|P_{0,s,a}\|_p(s') - \nabla \|\widehat{P}_{0,s,a}(s')\|_p)$$

As the norm is $C^2$ for $p \geq 2$, using Mean value theorem, we know that

$$\left\| (\nabla \|P_{0,s,a}\|_p - \nabla \|\widehat{P}_{0,s,a}\|_p) \right\|_2 \leq \sup_{x \in \Delta(S)} \left\| \nabla^2 \|x\|_p \right\|_2 \left\| (P_{0,s,a} - \widehat{P}_{0,s,a}) \right\|_2.$$

For $L_p = \|x\|_p$ norms, $p \geq 2$, we have simple taking derivative twice:

$$\nabla^2 \|x\|_p = \frac{p-1}{L_p} \left( \mathcal{A}^{p-2} - g_p g_p^T \right)$$

with

$$\mathcal{A} = \text{Diag}\left(\frac{\text{abs}(x)}{L_p}\right)$$

$$g_p = \mathcal{A}^{p-2}\left(\frac{x}{L_p}\right).$$

and $L_p$ the norm, where Diag is the diagonal matrix. However, as $x \leq L_p$, $\mathcal{A} \leq I$, we get

$$H \leq \frac{p-1}{\|x\|_p} \leq (p-1)|S|^{1/q} \tag{40}$$

where the $1/L_p$ is minimized for the uniform distribution. Then using Cauchy-Swartz inequality, it holds

$$\left(P_{0,s,a} - \widehat{P}_{0,s,a}\right)\left(\mu_{P_{0,s,a}}^{\lambda,\omega} - \mu_{\widehat{P}_{0,s,a}}^{\lambda,\omega}\right) \leq (p-1)\lambda|S|^{1/q}\left\|\left(P_{0,s,a} - \widehat{P}_{0,s,a}\right)\right\|_2^2. \tag{41}$$

Then the question is how to bound the quantity $\left\|\left(P_{0,s,a} - \widehat{P}_{0,s,a}\right)\right\|_2^2$. To do so, we will use Mac Diarmid inequality.

**Definition B.1.** *Bounded difference property*

*A function $f : \mathcal{X}_1 \times \ldots \mathcal{X}_n \to \mathbb{R}$ satisfies the bounded difference property if for each $i = 1, \ldots, n$ the change of coordinate from $s_i$ to $s_i'$ may change the value of the function at most on $c_i$*

$$\forall i \in [n] : \sup_{x_i' \in \mathcal{X}_i} |f(x_1, \ldots, x_i, \ldots, x_n) - f(x_1, \ldots, x_i', \ldots, x_n)| \leq c_i$$

In our case, we consider $f(X_1, \ldots, X_n) = \|\sum_{k=1}^n X_k\|_2$. Then we can notice that by triangle inequality for any $x_1, \ldots, x_n$ and $x_k'$ with $X_{i,s'} = P_{0,s,a}^i(s') - P_{0,s,a}(s')$ ( index $i$ holds for index of sample generated from the generative model) that

$$f(x_1, \ldots, x_k, \ldots, x_n) = \|x_1 + \ldots + x_n\|_2 \leq \|x_1 + \ldots + x_n - x_k + x_k'\|_2 + \|x_k - x_k'\|_2$$
$$\leq f(x_1, \ldots, x_k', \ldots, x_n) + 2$$

**Theorem B.9.** *(McDiarmid's inequality). McDiarmid et al. [1989] Let $f : \mathcal{X}_1 \times \ldots \mathcal{X}_n \to \mathbb{R}$ be a function satisfying the bounded difference property with bounds $c_1, \ldots, c_n$. Consider independent random variables $X_1, \ldots, X_n, X_i \in \mathcal{X}_i$ for all $i$. Then for any $t > 0$*

$$\mathbb{P}\left[f(X_1, \ldots, X_n) - \mathbb{E}\left[f(X_1, \ldots, X_n)\right] \geq t\right] \leq \exp\left(-\frac{2t^2}{\sum_{i=1}^n c_i^2}\right)$$

Using McDiarmid's inequality and union bound, we can bound the term as here

$$\left\|\left(P_{0,s,a} - \widehat{P}_{0,s,a}\right)\right\|_2^2 - \mathbb{E}[\left\|\left(P_{0,s,a} - \widehat{P}_{0,s,a}\right)\right\|_2^2] \leq \frac{2N\log(|S||A|/\delta))}{N^2}$$

with probability $1 - \delta/(|S||A|)$. Moreover, the additional term can be bounded as follows:

$$\mathbb{E}[\left\|\left(P_{0,s,a} - \widehat{P}_{0,s,a}\right)\right\|_2^2] = \mathbb{E}[\sum_{s'}(P_{0,s,a}(s') - P_{0,s,a}(s'))^2] = \mathbb{E}[\sum_{s'}(\frac{1}{N}\sum_i X_{i,s'})^2]$$

with $X_{i,s'} = P_{0,s,a}^i(s') - P_{0,s,a}(s')$ is one sample sampled from the generative model. Then

$$\mathbb{E}[\left\|\left(P_{0,s,a} - \widehat{P}_{0,s,a}\right)\right\|_2^2] = \frac{1}{N^2} \sum_{s'} \mathsf{Var}(\sum_i X_{i,s}) \stackrel{a}{=} \frac{1}{N^2} \sum_i \sum_{s'} \mathsf{Var}(X_{i,s})$$

$$= \frac{1}{N^2} \sum_i^N \mathbb{E}(\sum_{s'} X_{i,s}^2) \le \frac{4}{N}$$

where (a) the last equality comes from the independence of the random variables and where the last inequality comes from the fact the maximum of two elements in the simplex is bounded by 2. Finally, regrouping the two terms, we obtain with probability $1 - \delta/(|S||A|)$:

$$\left\|\left(P_{0,s,a} - \widehat{P}_{0,s,a}\right)\right\|_2^2 \le \frac{2N\log(|S||A|/(\delta)))}{N^2} + \frac{4}{N} = \frac{2\log(|S||A|/(\delta)))}{N} + \frac{4}{N}$$

$$\le \frac{6\log(|S||A|/(\delta))}{N} = \frac{L'}{N}$$

with $L' = 6\log(|S||A|/(\delta))$. Finally, plugging the previous equation in (41):

$$\max_{\mu \in \mu_{\widehat{P}_{0,s,a}}^\lambda} \left(P_{0,s,a} - \widehat{P}_{0,s,a}\right)(\mu_{P_{0,s,a}}^\lambda - \mu)| \le \max_\lambda \left\|\left(P_{0,s,a} - \widehat{P}_{0,s,a}\right)\right\|_2^2 S^{1/q}(p-1)\lambda.$$

This term can be easily controlled by taking the supremum over $\lambda$ which is a 1 dimensional parameter. Then we can bound $\lambda \in [0, H\left\|1_S\right\|_q]$. Indeed,

$$\lambda^* = \|V^* - \mu^* - \omega\|_q \le \|V^*\|_q \le H\left\|1_S\right\|_q.$$

Finally, we obtain:

$$\max_\lambda \left\|\left(P_{0,s,a} - \widehat{P}_{0,s,a}\right)\right\|_2^2 S^{1/q}\lambda \le \frac{L'|S|^{1/q}\left\|1_S\right\|_q (p-1)}{N(1-\gamma)}.$$

Regrouping all terms:

$$g_{s,a}(\alpha_{\widehat{P}}^\lambda, V^*) \le |\max_{\mu_{P_{0,s,a}}^\lambda \in \mathcal{M}_{\widehat{P}_{0,s,a}}^\lambda} \left(P_{0,s,a} - \widehat{P}_{0,s,a}\right)(V^* - \mu_{P_{0,s,a}}^\lambda) + \max_{\mu_{P_{0,s,a}}^\lambda \in \mathcal{M}_{\widehat{P}_{0,s,a}}^\lambda} \left(P_{0,s,a} - \widehat{P}_{0,s,a}\right)(\mu_{P_{0,s,a}}^\lambda - \mu_{\widehat{P}_{0,s,a}}^\lambda)|$$

$$\le 2\sqrt{\frac{L}{2N(1-\gamma)^2}} + \frac{L'|S|^{1/q}\left\|1_S\right\|_q (p-1)}{N(1-\gamma)} \le 2\sqrt{\frac{L}{2N(1-\gamma)^2}} + \frac{2L|S|^{1/q}\left\|1_S\right\|_q (p-1)}{N(1-\gamma)} \qquad (42)$$

$$(43)$$

For the specific case of $TV$ which is not $C^2$ smooth, this lemma still holds as in (27), we only need to control one term without the dependency on data in the supremum as $\alpha_{\widehat{P}}^\lambda$ reduces to a scalar $\alpha$ which does not depend on $P$. Then extra decomposition using smoothness of the norm is not needed, as the only remaining term in the max in (27) is the left hand side term.

$\square$

**Lemma B.10** ($s$-absorbing MDPs for Hoeffding's concentration Inequalities).

As in Agarwal paper Agarwal et al. [2020], we define for a state $s$ and a scalar $u$, the MDP called $M_{s,u}$ such that: $M_{s,u}$ is identical to $M$ except that state $s$ is absorbing in $M_{s,u}$, i.e. $P_{M_{s,u}}(s \mid s, a) = 1$ for all $a$, and the reward at state $s$ in $M_{s,u}$ is $(1-\gamma)u$. The remainder of the transition model and reward function are identical to those in $M$. In the following, we will use $V_{s,u}^\pi$ to denote the value function $V_{M_{s,u}}^\pi$ and correspondingly for $Q$ and reward and transition functions to avoid notational clutter. Then, we have that for all policies $\pi$ :

$$V_{s,u}^\pi(s) = u$$

because $s$ is absorbing with reward $(1-\gamma)u$. For some state $s$, we will only consider the MDP $M_{s,u}$ for $u$ in a finite set $U_s$ with

$$U_s \subset [V^\star(s) - \Delta_{\delta,N} V^\star(s) + \Delta_{\delta,N}].$$

with $\Delta_{\delta,N} := \frac{\gamma}{(1-\gamma)^2}\left(2\sqrt{\frac{L}{2N}} + \frac{2L|S|^{1/q}\|1_S\|_q(p-1)}{N}\right)$ The set $U_s$ consists of evenly spaced elements in this interval, where we set the size of $|U_s|$ appropriately later on. As before, we let $\widehat{M}_{s,u}$ denote the MDP that uses the empirical model $\widehat{P}$ instead of $P$, at all non-absorbing states and abbreviate the value functions in $\widehat{M}_{s,u}$ as $\widehat{V}^\pi_{s,u}$. Then we have for a fix a state $s$, action $a$, a finite set $U_s$, and $\delta \geq 0$, that for all $u \in U_s$: with probability greater than $1-\delta$, it holds :

$$|(\widehat{P}_{s,a} - P_{0,s,a})[V^{\widehat{\pi}}_u]_{\alpha^{\lambda,\omega}_P}| \leq \frac{1}{(1-\gamma)}\left(2\sqrt{\frac{\log(\frac{18SAN|U_s|\|1\|_q}{\delta})}{2N}} + \frac{2\log(\frac{18SAN|U_s|\|1\|_q}{\delta})|S|^{1/q}\|1_S\|_q(p-1)}{N}\right) \quad (44)$$

This is exactly B.8 in equation (27) to the finite set $U_s$ as now $V^{\widehat{\pi}}_u$ and $\widehat{P}_{s,a}$ are now independent.

**Lemma B.11** (Agarwal et al. [2020], Lemma 7). *Let $u^* = V^\star_M(s)$ and $u^\pi = V^\pi_M(s)$. We have*

$$V^\star_M = V^\star_{s,u^\star}, \quad \text{and for all policies } \pi, \quad V^\pi_M = V^\pi_{M^\pi_{s,u^\pi}}$$

Proof can be found in Agarwal et al. [2020], Lemma 7.

**Lemma B.12.** *For any $u, u', s$ and policy $\pi$:*

$$\left\|Q^\pi_{s,u} - Q^\pi_{s,u'}\right\|_\infty \leq |u - u'|$$

*Proof.* To obtain the result in our robust MDP setting, we need a similar stability property like in Lemma 8 of Agarwal et al. [2020], but for the robust value functions. It turns out that this a direct consequence of the property for classical MDP. Agarwal in Agarwal et al. [2020] show equation 45 for classical MPDs, then we have for RMDPs:

$$|Q^\pi_{M_{s,u}}(s,a) - Q^\pi_{M_{s,u'}}(s,a)| \leq \frac{1}{1-\gamma}|u - u'| \quad (45)$$

$$\Rightarrow |\inf_M Q^\pi_{M_{s,u}}(s,a) - \inf_M Q^\pi_{M_{s,u}}(s,a)| \leq \frac{1}{1-\gamma}|u - u'| \quad (46)$$

$$\Rightarrow |\sup_\pi \inf_M Q^\pi_{M_{s,u}}(s,a) - \sup_\pi \inf_M Q^\pi_{M_{s,u}}(s,a)| \leq \frac{1}{1-\gamma}|u - u'|. \quad (47)$$

which concludes the proof for RMDPs. $\qquad\square$

**Lemma B.13** (Hoeffding's Concentration for dependent variables). *Removing $s, a$ notations for kernels,*

$$\left|(P_0 - \widehat{P}) \cdot [\widehat{V}^\star]_{\alpha^{\lambda,\omega}_P}\right| \leq \frac{1}{(1-\gamma)}\left(2\sqrt{\frac{\log(\frac{18SAN|U_s|\|1\|_q}{\delta})}{2N}} + \frac{2\log(\frac{18SAN|U_s|\|1\|_q}{\delta})|S|^{1/q}\|1_S\|_q(p-1)}{N}\right) \quad (48)$$

$$+ 2\min_{u \in U_s}\left|\widehat{V}^\star(s) - u\right| \quad (49)$$

*Proof.*

$$\left| \left( P_0 - \widehat{P} \right) \cdot [\widehat{V}^\star]_{\alpha_P^{\lambda,\omega}} \right| = \left| \left( P_0 - \widehat{P} \right) \cdot \left( [\widehat{V}^\star]_{\alpha_P^{\lambda,\omega}} - [V_{s,u}^\star]_{\alpha_P^{\lambda,\omega}} + [V_{s,u}^\star]_{\alpha_P^{\lambda,\omega}} \right) \right| \tag{50}$$

$$\leq \left| \left( P_0 - \widehat{P} \right) \cdot \left( [\widehat{V}^\star]_{\alpha_P^{\lambda,\omega}} - [V_{s,u}^\star]_{\alpha_P^{\lambda,\omega}} \right) \right| + \left| \left( P_0 - \widehat{P} \right) \cdot \left( [V_{s,u}^\star]_{\alpha_P^{\lambda,\omega}} \right) \right| \tag{51}$$

$$\overset{(a)}{\leq} \frac{1}{(1-\gamma)} \left( 2\sqrt{\frac{\log(\frac{18SAN|U_s|\|1\|_q}{\delta})}{2N}} + \frac{2\log(\frac{18SAN|U_s|\|1\|_q}{\delta})|S|^{1/q}\|1_S\|_q(p-1)}{N} \right) \tag{52}$$

$$+ 2\left\| \widehat{V}^\star - V_{s,u}^\star \right\|_\infty \tag{53}$$

$$\overset{(b)}{\leq} \frac{1}{(1-\gamma)} \left( 2\sqrt{\frac{\log(\frac{18SAN|U_s|\|1\|_q}{\delta})}{2N}} + \frac{2\log(\frac{18SAN|U_s|\|1\|_q}{\delta})|S|^{1/q}\|1_S\|_q(p-1)}{N} \right) \tag{54}$$

$$+ 2\left| \widehat{V}^\star(s) - u \right| \tag{55}$$

$$\tag{56}$$

where $(a)$ is 44 or Hoeffding's inequality for s-absorbing MDPs. By Lemmas B.11 and B.12,

$$\left\| [\widehat{V}^\star]_{\alpha_P^{\lambda,\omega}} - [V_{s,u}^\star]_{\alpha_P^{\lambda,\omega}} \right\|_\infty \leq \left\| [\widehat{V}^\star - V_{s,u}^\star]_{\alpha_P^{\lambda,\omega}} \right\|_\infty \leq \left\| \widehat{V}^\star - V_{s,u}^\star \right\|_\infty = \left\| \widehat{V}_{s,\widehat{V}^\star(s)}^\star - V_{s,u}^\star \right\|_\infty \leq \left| \widehat{V}^\star(s) - u \right|.$$

which is point $(b)$. The last $\min$ operator in the result comes from the fact that the previous equation holds for all $u \in U_s$, we take the best possible choice, which completes the proof of the first claim. $\qquad\square$

**Lemma B.14** (Crude bound for Robust MDPs). *This lemma is needed for next Lemma B.15 but the proof differs from the classical MDP setting. For s and sa rectangular assumptions,*

$$\left\| Q^* - \widehat{Q}^{\pi^*} \right\|_\infty \leq \Delta_{\delta,N} \text{ and } \left\| Q^* - \widehat{Q}^* \right\|_\infty \leq \Delta_{\delta,N} \quad \text{with} \quad \Delta_{\delta,N} = \frac{\gamma}{(1-\gamma)^2}\left( 2\sqrt{\frac{L}{2N}} + \frac{2L|S|^{1/q}\|1_S\|_q(p-1)}{N} \right)$$

*Proof.* For the first claim :

$$\left\| Q^\pi - \widehat{Q}^\pi \right\|_\infty = \max_{s,a} \left| \gamma \left( \kappa_{\mathcal{P}_{0,s,a}}(V^\pi) - \kappa_{\widehat{\mathcal{P}}_{s,a}}(V^\pi) \right) + \gamma \left( \kappa_{\widehat{\mathcal{P}}_{s,a}}(V^\pi) - \kappa_{\widehat{\mathcal{P}}_{s,a}}(\widehat{V}^\pi) \right) \right|$$

$$\overset{(b)}{\leq} \max_{s,a} \left| \gamma \left( \kappa_{\mathcal{P}_{0,s,a}}(V^\pi) - \kappa_{\widehat{\mathcal{P}}_{s,a}}(V^\pi) \right) \right| + \gamma \left\| V^\pi - \widehat{V}^\pi \right\|_\infty$$

$$\overset{(b)}{\leq} \gamma \max_{s,a} \left| \kappa_{\widehat{\mathcal{P}}_{s,a}}(V^\pi) - \kappa_{\mathcal{P}_{0,s,a}}(V^\pi) \right| + \gamma \left\| Q^\pi - \widehat{Q}^\pi \right\|_\infty.$$

$$\square$$

where we use contraction of $\kappa$, lemma B.3 in (a) and $\left\| Q^\pi - \widehat{Q}^\pi \right\|_\infty \leq \left\| V^\pi - \widehat{V}^\pi \right\|_\infty$ in (c) for any $\pi$. Solving we get :

$$\left\| Q^\pi - \widehat{Q}^\pi \right\|_\infty \leq \frac{\gamma}{1-\gamma} \max_{s,a} \left| \kappa_{\widehat{\mathcal{P}}_{s,a}}(V^\pi) - \kappa_{\mathcal{P}_{0,s,a}}(V^\pi) \right|$$

Then using Lemma B.7, we obtain :

$$\left\| Q^\pi - \widehat{Q}^\pi \right\|_\infty \leq \frac{\gamma}{1-\gamma} \max_{s,a} \left| \kappa_{\widehat{\mathcal{P}}_{s,a}}(V^\pi) - \kappa_{\mathcal{P}_{0,s,a}}(V^\pi) \right|$$

Taking $\pi = \pi^*$, $V^{\pi^*}$ is independent of the data and we can use Lemma B.8. Finally, we have

$$\left\| Q^* - \widehat{Q}^{\pi^*} \right\|_\infty \leq \frac{\gamma}{1-\gamma} \left\| (\widehat{P} - P_0)V^\pi \right\|_\infty \leq \frac{\gamma}{1-\gamma} \left( 2\sqrt{\frac{L}{2N(1-\gamma)^2}} + \frac{2L|S|^{1/q}\|1_S\|_q(p-1)}{N(1-\gamma)} \right)$$

For the second point, using $s$ or $sa$ rectangular assumptions,

$$
\begin{aligned}
\left\| Q^* - \hat{Q}^* \right\|_\infty &\leq \left\| \mathcal{T}_{\mathcal{U}_p^{sa}}^{\pi^*} Q^* - \hat{\mathcal{T}}_{\mathcal{U}_p^{sa}}^{\hat{\pi}^*} Q^* + \hat{\mathcal{T}}_{\mathcal{U}_p^{sa}}^{\hat{\pi}^*} Q^* - \hat{\mathcal{T}}_{\mathcal{U}_p^{sa}}^{\hat{\pi}^*} \hat{Q}^* \right\|_\infty \\
&\leq \left\| \mathcal{T}_{\mathcal{U}_p^{sa}}^{\pi^*} Q^* - \hat{\mathcal{T}}_{\mathcal{U}_p^{sa}}^{\hat{\pi}^*} Q^* \right\|_\infty + \left\| \hat{\mathcal{T}}_{\mathcal{U}_p^{sa}}^{\hat{\pi}^*} Q^* - \hat{\mathcal{T}}_{\mathcal{U}_p^{sa}}^{\hat{\pi}^*} \hat{Q}^* \right\|_\infty \\
&\overset{(a)}{\leq} \left\| \mathcal{T}_{\mathcal{U}_p^{sa}}^{\pi^*} Q^* - \hat{\mathcal{T}}_{\mathcal{U}_p^{sa}}^{\hat{\pi}^*} Q^* \right\|_\infty + \gamma \left\| Q^* - \hat{Q}^* \right\|_\infty \\
&\overset{(b)}{\leq} \left\| \kappa_{\hat{\mathcal{P}}_{s,a}}(V^*) - \kappa_{\mathcal{P}_{0,s,a}}(V^*) \right\|_\infty + \gamma \left\| Q^* - \hat{Q}^* \right\|_\infty
\end{aligned}
$$

Then using Lemma B.7, and solving we get :

$$
\left\| Q^* - \hat{Q}^* \right\|_\infty \frac{\gamma}{1-\gamma} \left\| \kappa_{\hat{\mathcal{P}}_{s,a}}(V^*) - \kappa_{\mathcal{P}_{0,s,a}}(V^*) \right\|_\infty
$$

Finally using Lemma B.8, we obtain

$$
\left\| Q^* - \hat{Q}^* \right\|_\infty \leq \frac{\gamma}{(1-\gamma)^2} \left( 2\sqrt{\frac{L}{2N}} + \frac{2L|S|^{1/q} \left\| 1_S \right\|_q (p-1)}{N} \right)
$$

which concludes the proof.

**Lemma B.15** (Similar to Agarwal, Agarwal et al. [2020] lemma 9 but for RMPDs). *With probability $1 - \delta$, we have:*

$$
\min_{u \in U_s} \left| \widehat{V}^\star(s) - u \right| \leq 4\gamma \left( 2\sqrt{\frac{L}{2N}} + \frac{2L|S|^{1/q} \left\| 1_S \right\|_q (p-1)}{N} \right)
$$

*Proof.* The proof can be found in Agarwal et al. [2020] and is similar for RMDs than for classical MPDs and consists in choosing $U_s$ to be the evenly spaced elements in the interval $\left[ V^\star(s) - \Delta_{\delta/2,N} V^\star(s) + \Delta_{\delta/2,N} \right]$, then finally the size of $U_s$ is chosen to be $|U_s| = \frac{1}{(1-\gamma)^2}$. Using lemma , with probability greater than $1 - \delta/2$, we have $\widehat{V}^\star(s) \in \left[ V^\star(s) - \Delta_{\delta/2,N} V^\star(s) + \Delta_{\delta/2,N} \right]$ for all $s$ according to Lemma B.14. This implies using that that $\hat{V}^{\pi^*}$ will land in one of $|U_s| - 1$ evenly sized sub-intervals of length $2\Delta_{\delta/2,N}$ :

$$
\begin{aligned}
\min_{u \in U_s} \left| \widehat{V}^\star(s) - u \right| &\leq \frac{2\Delta_{\delta/2,N}}{|U_s| - 1} = \frac{2}{|U_s| - 1} \frac{\gamma}{(1-\gamma)^2} \left( 2\sqrt{\frac{L}{2N}} + \frac{2L|S|^{1/q} \left\| 1_S \right\|_q}{N} \right) \\
&\leq 4\gamma \left( 2\sqrt{\frac{L}{2N}} + \frac{2L|S|^{1/q} \left\| 1_S \right\|_q (p-1)}{N} \right)
\end{aligned}
$$

$\square$

**Lemma B.16** (Relation between concentration of robust and non-robust MDPs). *With probability $1 - \delta$, we get:*

$$
\max_{s,a} \left| \kappa_{\hat{\mathcal{P}}_{s,a}}(V^{\hat{\pi}}) - \kappa_{\mathcal{P}_{0,s,a}}(V^{\hat{\pi}}) \right| \leq \frac{10}{(1-\gamma)} \left( \sqrt{\frac{L''}{2N}} + \frac{L''|S|^{1/q} \left\| 1_S \right\|_q (p-1)}{N} \right) + 2\epsilon_{opt}.
$$

$$
\max_{s,a} \left| \kappa_{\hat{\mathcal{P}}_{s,a}}(V^*) - \kappa_{\mathcal{P}_{0,s,a}}(V^*) \right| \leq \frac{10}{(1-\gamma)} \left( \sqrt{\frac{L''}{2N}} + \frac{L''|S|^{1/q} \left\| 1_S \right\|_q (p-1)}{N} \right).
$$

*with $L'' = \log\left( \frac{32SAN \|1\|_q}{\delta(1-\gamma)} \right)$*

*Proof.* Using Lemma B.7, we directly have the first inequality equality part of the first statement:

$$
\max_{s,a} \left| \kappa_{\hat{\mathcal{P}}_{s,a}}(\hat{V}^{\hat{\pi}}) - \kappa_{\mathcal{P}_{0,s,a}}(\hat{V}^{\hat{\pi}}) \right|
$$

is bounded by either by

$$\max_{(s,a)} \max_{\alpha_P^{\lambda,\omega} \in A_P^{\lambda,\omega}} \left| \left( P_0 - \widehat{P} \right) [\hat{V}^{\hat{\pi}}]_{\alpha_P^{\lambda,\omega}} \right|$$

or

$$\max_{(s,a)} \max_{\alpha_{\hat{P}}^{\lambda,\omega} \in A_{\hat{P}}^{\lambda,\omega}} \left| \left( P_0 - \widehat{P} \right) [\hat{V}^{\hat{\pi}}]_{\alpha_{\hat{P}}^{\lambda,\omega}} \right|.$$

We know that in both cases that

$$\max_{(s,a)} \left| \left( P_0 - \widehat{P} \right) [\hat{V}^{\hat{\pi}}]_{\alpha_P^{\lambda,\omega}} \right| \leq \max_{(s,a)} |(P_0 - \widehat{P})([\hat{V}^{\hat{\pi}}]_{\alpha_P^{\lambda,\omega}} - [\hat{V}^*]_{\alpha_P^{\lambda,\omega}})| + \max_{(s,a)} |(P_0 - \widehat{P})[\hat{V}^*]_{\alpha_P^{\lambda,\omega}}|,$$

using $|[\hat{V}^{\hat{\pi}}]_{\alpha_P^{\lambda,\omega}}| - |[\hat{V}^*]_{\alpha_P^{\lambda,\omega}}| \leq |([\hat{V}^{\hat{\pi}} - \hat{V}^*]_{\alpha_P^{\lambda,\omega}})| \leq |(\hat{V}^{\hat{\pi}} - \hat{V}^*)|$ and combining Lemma B.13 and B.15, for $|U_s| = \frac{1}{(1-\gamma)^2}$, with probability $1 - \delta$, we have :

$$\left| \left( P_0 - \widehat{P} \right) [\hat{V}^{\hat{\pi}}]_{\alpha_P^{\lambda,\omega}} \right| \leq 4\gamma \left( 2\sqrt{\frac{L''}{2N}} + \frac{2LS^{1/q} \|1_S\|_q}{N} \right) + \frac{1}{(1-\gamma)} \left( 2\sqrt{\frac{L''}{2N}} + \frac{2L''S^{1/q} \|1_S\|_q}{N} \right) + 2\epsilon_{opt}.$$

$$\leq \frac{10}{(1-\gamma)} \left( \sqrt{\frac{L''}{2N}} + \frac{L''|S|^{1/q} \|1_S\|_q (p-1)}{N} \right) + 2\epsilon_{opt}.$$

The proof is exactly the same by replacing $\hat{\pi}$ by $\pi^*$ but without the $2\epsilon_{opt}$, which gives the second stated result. Again, this proof is written for the $sa$-rectangular assumption, it is also true for the $s$-rectangular case with slightly different notations, replacing $\mathcal{D} = \mathcal{P}_{0,s,a}$ by $\mathcal{D} = \mathcal{P}_{0,s}$. $\qquad \square$

These two inequalities are the core of our proof, as the closed form solution of the $\min$ problem in the robust setting only depends on $\alpha, \beta$ and the current value function.

**Theorem B.17.** *Suppose $\delta > 0$, $\epsilon > 0$ and $\beta > 0$, let $\hat{\pi}$ be any $\epsilon_{opt}$-optimal policy for $\widehat{M}$, i.e. $\left\| \widehat{Q}^{\hat{\pi}} - \widehat{Q}^\star \right\|_\infty \leq \epsilon_{opt}$. If*

$$N \geq \frac{C\gamma^2 L''}{(1-\gamma)^4 \epsilon^2},$$

*we get*

$$\left\| Q^* - Q^{\hat{\pi}} \right\|_\infty \leq \epsilon + \frac{3\gamma\epsilon_{opt}}{1-\gamma}$$

*with probability at least $1 - \delta$, where $C$ is an absolute constant. Finally, for $N_{total} = N|\mathcal{S}||\mathcal{A}|$ and $H = 1/(1-\gamma)$, we get an overall complexity of*

$$N_{total} = \tilde{O} \left( \frac{H^4 |S||A|}{\epsilon^2} \right).$$

*Proof.*

$$\begin{aligned}
\left\| Q^* - Q^{\hat{\pi}} \right\|_\infty &\overset{(a)}{\leq} \left\| Q^* - \hat{Q}^* \right\|_\infty + \left\| \hat{Q}^* - \hat{Q}^{\hat{\pi}} \right\|_\infty + \left\| \hat{Q}^{\hat{\pi}} - Q^{\hat{\pi}} \right\|_\infty \\
&\overset{(b)}{\leq} \epsilon_{opt} + \frac{\gamma}{(1-\gamma)} \left( \max_{s,a} \left| \kappa_{\hat{\mathcal{P}}_{s,a}} (V^*) - \kappa_{\mathcal{P}_{s,a}} (V^*) \right| + \max_{s,a} \left| \kappa_{\mathcal{P}_{s,a}} (V^{\hat{\pi}}) - \kappa_{\mathcal{P}_{s,a}} (V^{\hat{\pi}}) \right| \right) \\
&\overset{(c)}{\leq} \frac{20\gamma}{(1-\gamma)^2} \left( \sqrt{\frac{L''}{2N}} + \frac{L''|S|^{1/q} \|1_S\|_q (p-1)}{N} \right) + \epsilon_{opt} + \frac{2\gamma\epsilon_{opt}}{1-\gamma} \\
&\leq \frac{20\gamma}{(1-\gamma)^2} \left( \sqrt{\frac{L''}{2N}} + \frac{L''|S|^{1/q} \|1_S\|_q (p-1)}{N} \right) + \epsilon_{opt} + \frac{2\gamma\epsilon_{opt}}{1-\gamma} \\
&\overset{(d)}{\leq} \epsilon + \frac{3\gamma\epsilon_{opt}}{1-\gamma}
\end{aligned}$$

Inequality (a) is due to Lemma B.1. Inequality (b) comes from Lemma B.4. Finally, inequality (c) comes from Lemma B.16 and inequality (d) from the form of $N$ in the theorem. For $N \geq H^4 SA$, the second term proportional to $1/N$ is very small compared to the asymptotic term in $1/\sqrt{N}$ for small $\epsilon$. Note that $S^{1/q} \|1_S\|_q = |S|$ for $L_2$ norm for example. This proof holds for both $s$- and $sa$-rectangular assumptions. □

## C  TOWARDS MINIMAX OPTIMAL BOUNDS

We start from the same decomposition as the proof of Theorem 4.1 proved in Lemma B.1:

$$\left\| Q^* - Q^{\hat{\pi}} \right\|_\infty \leq \left\| Q^* - \hat{Q}^{\pi^*} \right\|_\infty + \left\| \hat{Q}^{\pi^*} - \hat{Q}^{\hat{\pi}} \right\|_\infty + \left\| \hat{Q}^{\hat{\pi}} - Q^{\hat{\pi}} \right\|_\infty.$$

However, we need tighter concentration arguments for this proof.

In the following, we will frequently use the fact that, for any policy $\pi$, written below for the $s$-rectangular case (a similar expression can be obtained for the $sa$-rectangular case, adapting the regularized reward),

Recall, the fix point equation for $Q^\pi$ can be written as :

$$Q^\pi = (I - \gamma P_0^\pi)^{-1} \left( R_0 - \alpha_s \left( \pi_s / \|\pi_s\|_q \right)^{q-1} + \gamma \inf_{P^\pi \in \mathcal{P}_s} P^\pi V^\pi \right) \tag{57}$$

It will be applied notably to $\hat{\pi}$ and $\pi^*$ (recall that $Q^* = Q^{\pi^*}$), in the RMDP but also in the empirical one.

**Lemma C.1.** *For s-rectangular we have*

$$
\begin{aligned}
(I - \gamma P_0^\pi)^{-1} r_{\hat{Q}_s^\pi}^s - \left( I - \gamma \widehat{P}^\pi \right)^{-1} r_{\hat{Q}_s^\pi}^s &\overset{(a)}{=} (I - \gamma P_0^\pi)^{-1} \left( \left( I - \gamma \widehat{P}^\pi \right) - (I - \gamma P_0^\pi) \right) \widehat{Q}_s^\pi \\
&= \gamma (I - \gamma P_0^\pi)^{-1} \left( P_0^\pi - \widehat{P}^\pi \right) \widehat{Q}_s^\pi \\
&= \gamma (I - \gamma P_0^\pi)^{-1} (P_0 - \widehat{P}) \widehat{V}_s^\pi
\end{aligned}
$$

*and for optimal policy*

$$\left( I - \gamma P_0^{\pi^*} \right)^{-1} r_{\hat{Q}_s^{\pi^*}}^s - \left( I - \gamma \hat{P}^{\pi^*} \right)^{-1} r_{\hat{Q}_s^{\pi^*}}^s = \gamma \left( I - \gamma P_0^{\pi^*} \right)^{-1} (P_0 - \widehat{P}) \hat{V}_s^{\pi^*} \tag{58}$$

$$\left( I - \gamma P_0^{\hat{\pi}} \right)^{-1} r_{\hat{Q}_s^{\hat{\pi}}}^s - \left( I - \gamma \hat{P}^{\hat{\pi}} \right)^{-1} r_{\hat{Q}_s^{\hat{\pi}}}^s = \gamma \left( I - \gamma P_0^{\hat{\pi}} \right)^{-1} (P_0 - \widehat{P}) \hat{V}_s^{\hat{\pi}} \tag{59}$$

The solution is a bit different as $r_{\hat{Q}_s^\pi}^s$ is the regularized form of the $L_p$ optimization problem with simplex constraints which correspond to $r_{\hat{Q}_s^\pi}^s = R_0 - \left( \frac{\pi_s^*}{\|\pi_s^*\|_q} \right)^{q-1} \alpha_s + \gamma \inf_{P^\pi \in \mathcal{P}_s} P^\pi \hat{V}^\pi$ or for $sa$ case : $r_{\hat{Q}_{sa}^\pi}^{(s,a)} = R_0 - \alpha_{sa} + \gamma \inf_{P^\pi \in \mathcal{P}_s} P^\pi \hat{V}^\pi$

Indeed, even without close form, we can write the problem with an expectation over the nominal and the infimum problem.

**Lemma C.2** (Upper bound on $Q^* - \hat{Q}^{\pi^*}$ and on $Q^{\hat{\pi}} - \hat{Q}^{\hat{\pi}}$, all Q values are now with robust under simplex constraints.)**.**

$$\left\| Q^* - \hat{Q}^{\pi^*} \right\|_\infty \leq \gamma \left\| (I - \gamma P_0^{\pi^*})^{-1} (P_0 - \widehat{P}) \hat{V}^{\pi^*} \right\|_\infty + \frac{2\gamma\beta|S|^{1/q}}{1 - \gamma} \left\| Q^* - \hat{Q}^{\pi^*} \right\|_\infty$$

$$\left\| Q^{\hat{\pi}} - \hat{Q}^{\hat{\pi}} \right\|_\infty \leq \gamma \left\| (I - \gamma P_0^{\hat{\pi}})^{-1} (P_0 - \widehat{P}) \hat{V}^{\hat{\pi}} \right\|_\infty + \frac{2\gamma\beta|S|^{1/q}}{1 - \gamma} \left\| Q^{\hat{\pi}} - \hat{Q}^{\hat{\pi}} \right\|_\infty$$

*Proof.*

$$Q^* - \hat{Q}^{\pi^*}$$

$$= \left(I - \gamma P_0^{\pi^*}\right)^{-1} \left(R_0 - \left(\frac{\pi_s^*}{\|\pi_s^*\|_q}\right)^{q-1} \alpha_s + \inf_{P^\pi \in \mathcal{P}_s} P^\pi V^*\right)$$

$$- \left(I - \gamma \hat{P}^{\pi^*}\right)^{-1} \left(R_0 - \left(\frac{\pi_s^*}{\|\pi_s^*\|_q}\right)^{q-1} \alpha_s + \inf_{P^\pi \in \mathcal{P}_s} P^\pi \hat{V}^{\pi^*}\right)$$

$$= \left(I - \gamma P_0^{\pi^*}\right)^{-1} \left(R_0 - \left(\frac{\pi_s^*}{\|\pi_s^*\|_q}\right)^{q-1} \alpha_s + \gamma \inf_{P^\pi \in \mathcal{P}_s} P^\pi V^*\right)$$

$$- \left(I - \gamma P_0^{\pi^*}\right)^{-1} \left(R_0 - \left(\frac{\pi_s^*}{\|\pi_s^*\|_q}\right)^{q-1} \alpha_s + \gamma \inf_{P^\pi \in \mathcal{P}_s} P^\pi \hat{V}^{\pi^*}\right)$$

$$+ \left(I - \gamma P_0^{\pi^*}\right)^{-1} \left(R_0 - \left(\frac{\pi_s^*}{\|\pi_s^*\|_q}\right)^{q-1} \alpha_s + \gamma \inf_{P^\pi \in \mathcal{P}_s} P^\pi \hat{V}^{\pi^*}\right)$$

$$- \left(I - \gamma \hat{P}^{\pi^*}\right)^{-1} \left(R_0 - \left(\frac{\pi_s^*}{\|\pi_s^*\|_q}\right)^{q-1} \alpha_s + \gamma \inf_{P^\pi \in \mathcal{P}_s} P^\pi \hat{V}^{\pi^*}\right)$$

$$\overset{(a)}{=} \gamma \left(I - \gamma P_0^{\pi^*}\right)^{-1} (P_0 - \widehat{P})\hat{V}^{\pi^*} + \left(I - \gamma P_0^{\pi^*}\right)^{-1} \gamma \left(\inf_{P^\pi \in \mathcal{P}_s} P^\pi V^* - \inf_{P^\pi \in \mathcal{P}_s} P^\pi \hat{V}^{\pi^*}\right)$$

where in (a) we use previous Lemma C.1. □

Hence, taking the supremum norm $\|.\|_\infty$,

$$\left\|Q^* - \hat{Q}^{\pi^*}\right\|_\infty =$$

$$\left\|\gamma \left(I - \gamma P_0^{\pi^*}\right)^{-1} (P_0 - \widehat{P})\hat{V}^{\pi^*} + \left(I - \gamma P_0^{\pi^*}\right)^{-1} \gamma \left(\inf_{P^\pi \in \mathcal{P}_s} P^\pi V^* - \inf_{P^\pi \in \mathcal{P}_s} P^\pi \hat{V}^{\pi^*}\right)\right\|_\infty$$

$$\overset{(b)}{\leq} \left\|\gamma \left(I - \gamma P_0^{\pi^*}\right)^{-1} (P_0 - \widehat{P})\hat{V}^{\pi^*}\right\|_\infty + \left\|\left(I - \gamma P_0^{\pi^*}\right)^{-1} \gamma \left(\inf_{P^\pi \in \mathcal{P}_s} P^\pi V^* - \inf_{P^\pi \in \mathcal{P}_s} P^\pi \hat{V}^{\pi^*}\right)\right\|_\infty$$

$$\overset{(c)}{\leq} \left\|\gamma \left(I - \gamma P_0^{\pi^*}\right)^{-1} (P_0 - \widehat{P})\hat{V}^{\pi^*}\right\|_\infty + \frac{\gamma}{1-\gamma} \left|\inf_{P^\pi \in \mathcal{P}_s} P^\pi V^* - \inf_{P^\pi \in \mathcal{P}_s} P^\pi \hat{V}^{\pi^*}\right|$$

$$\overset{(d)}{\leq} \left\|\gamma \left(I - \gamma P_0^{\pi^*}\right)^{-1} (P_0 - \widehat{P})\hat{V}^{\pi^*}\right\|_\infty + \frac{\gamma}{1-\gamma} \sup_{P^\pi \in \mathcal{P}_s} P^\pi \left|V^* - \hat{V}^{\pi^*}\right|$$

$$\overset{(e)}{\leq} \left\|\gamma \left(I - \gamma P_0^{\pi^*}\right)^{-1} (P_0 - \widehat{P})\hat{V}^{\pi^*}\right\|_\infty + \frac{\gamma}{1-\gamma} \sup_{P:\|P\|_p \leq \beta_s, \sum_s P(s)=0} P \left|V^* - \hat{V}^{\pi^*}\right|$$

$$\overset{(f)}{\leq} \left\|\gamma \left(I - \gamma P_0^{\pi^*}\right)^{-1} (P_0 - \widehat{P})\hat{V}^{\pi^*}\right\|_\infty - \frac{\gamma}{1-\gamma} \inf_{P:\|P\|_p \leq \beta_s, \sum_s P(s)=0} -P \left|V^* - \hat{V}^{\pi^*}\right|$$

$$\overset{(g)}{\leq} \left\|\gamma \left(I - \gamma P_0^{\pi^*}\right)^{-1} (P_0 - \widehat{P})\hat{V}^{\pi^*}\right\|_\infty + \frac{\gamma\beta|S|^{1/q}}{1-\gamma} \mathrm{sp}_{q,\pi^*}(Q^* - \hat{Q}^{\pi^*})$$

$$\overset{(h)}{\leq} \left\|\gamma \left(I - \gamma P_0^{\pi^*}\right)^{-1} (P_0 - \widehat{P})\hat{V}^{\pi^*}\right\|_\infty + \frac{2\gamma\beta|S|^{1/q}}{1-\gamma} \left\|Q^* - \hat{Q}^{\pi^*}\right\|_\infty$$

where (b) is the triangular inequality, (c) Eq. (4), (d) is the triangular inequality for seminorms, (d) is $|\inf_A f - \inf_A g| \leq \sup_A |f - g|.$, (e) is a relaxation (f) is the relation between sup and inf, (g) is lemma 1 of Kumar et al. [2022]), (h) is inequality for seminorms and norms (5).

For brevity in the remaining analysis, let us define the shorthand:

$$L = \log(8|\mathcal{S}||\mathcal{A}|/((1-\gamma)\delta)).$$

Recall, slightly abusing the notation, for $V \in \mathbb{R}^S$, we define the vector $\mathrm{Var}_P(V) \in \mathbb{R}^{\mathcal{S} \times A}$ as $\mathrm{Var}_P(V) = P(V)^2 - (PV)^2$.

**Lemma C.3** (Agarwal et al. [2020], Lemma 9). *With probability greater than $1 - \delta$,*

$$\left| (P_0 - \widehat{P}) \widehat{V}^\star \right| \leq \sqrt{\frac{8L}{N}} \sqrt{\mathrm{Var}_{P_0} \left( \widehat{V}^\star \right)} + \Delta'_{\delta,N} \mathbb{I}$$

$$\left| (P_0 - \widehat{P}) \widehat{V}^{\pi^\star} \right| \leq \sqrt{\frac{8L}{N}} \sqrt{\mathrm{Var}_{P_0} \left( \widehat{V}^{\pi^\star} \right)} + \Delta'_{\delta,N} \mathbb{I}$$

*where* $\Delta'_{\delta,N} = \sqrt{\dfrac{cL}{N}} + \dfrac{cL}{(1-\gamma)N}$ *and $c$ is a universal constant smaller than* 16.

*Proof.* The proof of Agarwal et al. [2020] holds for classical MDP but can be adapted to the robust setting using all lemmas proved for the bound in $H^4$ previously. Lemma B.11,B.12 ,B.14,B.15,45 are needed but the main difference is that we are using Berstein's inequality and not Hoeffding's inequality. The idea is first, as in the previous proof, to apply Berstein's inequality to independent variables using $s$ absorbing MDPs then using Lemma B.15.

*Proof.* Similar to Agarwal et al. [2020], we first show that

$$\left| \left( P_0 - \widehat{P} \right) \cdot \widehat{V}^\star \right| \leq \sqrt{\frac{2 \log \left( 4 \left| U_s \right| / \delta \right)}{N}} \sqrt{\mathrm{Var}_{P_0} \left( \widehat{V}^\star \right)}$$

$$+ \min_{u \in U_s} \left| \widehat{V}^\star(s) - u \right| \left( 1 + \sqrt{\frac{2 \log \left( 4 \left| U_s \right| / \delta \right)}{N}} \right) + \frac{2 \log \left( 4 \left| U_s \right| / \delta \right)}{(1 - \gamma)3N}$$

$$\left| \left( P_0 - \widehat{P} \right) \cdot \widehat{V}^{\pi^\star} \right| \leq \sqrt{\frac{2 \log \left( 4 \left| U_s \right| / \delta \right)}{N}} \sqrt{\mathrm{Var}_{P_0} \left( \widehat{V}^{\pi^\star} \right)}$$

$$+ \min_{u \in U_s} \left| \widehat{V}^{\pi^\star}(s) - u \right| \left( 1 + \sqrt{\frac{2 \log \left( 4 \left| U_s \right| / \delta \right)}{N}} \right) + \frac{2 \log \left( 4 \left| U_s \right| / \delta \right)}{(1 - \gamma)3N}$$

First, with probability greater than $1 - \delta$, we have that for all $u \in U_s$.

$$\left| \left( P_0 - \widehat{P} \right) \cdot \widehat{V}^\star \right| = \left| \left( P_0 - \widehat{P} \right) \cdot \left( \widehat{V}^\star - V^\star_{s,u} + V^\star_{s,u} \right) \right|$$

$$\overset{(a)}{\leq} \left| \left( P_0 - \widehat{P} \right) \cdot \left( \widehat{V}^\star - V^\star_{s,u} \right) \right| + \left| \left( P_0 - \widehat{P} \right) \cdot \left( V^\star_{s,u} \right) \right|$$

$$\overset{(b)}{\leq} \left\| \widehat{V}^\star - V^\star_{s,u} \right\|_\infty + \sqrt{\frac{2 \log \left( 4 \left| U_s \right| / \delta \right)}{N}} \sqrt{\mathrm{Var}_{P_0} \left( V^\star_{s,u} \right)} + \frac{2 \log \left( 4 \left| U_s \right| / \delta \right)}{(1 - \gamma)3N}$$

$$\overset{(c)}{=} \left\| \widehat{V}^\star - V^\star_{s,u} \right\|_\infty + \sqrt{\frac{2 \log \left( 4 \left| U_s \right| / \delta \right)}{N}} \sqrt{\mathrm{Var}_{P_0} \left( \widehat{V}^\star - V^\star_{s,u} - \widehat{V}^\star \right)} + \frac{2 \log \left( 4 \left| U_s \right| / \delta \right)}{(1 - \gamma)3N}$$

$$\overset{(d)}{\leq} \left\| \widehat{V}^\star - V^\star_{\widehat{M}_{s,u}} \right\|_\infty \left( 1 + \sqrt{\frac{2 \log \left( 4 \left| U_s \right| / \delta \right)}{N}} \right) + \sqrt{\frac{2 \log \left( 4 \left| U_s \right| / \delta \right)}{N}} \sqrt{\mathrm{Var}_{P_0} \left( \widehat{V}^\star \right)} + \frac{2 \log \left( 4 \left| U_s \right| / \delta \right)}{(1 - \gamma)3N}$$

using the triangle inequality in (a), (b) classical Berstein's inequality, (d) for variance and Lemmas B.11 and B.12 such as

$$\left\| \widehat{V}^\star - V^\star_{s,u} \right\|_\infty = \left\| \widehat{V}^\star_{s,\widehat{V}^\star(s)} - V^\star_{s,u} \right\|_\infty \leq \left| \widehat{V}^\star(s) - u \right|.$$

It is true for $u \in U_s$, so we take the best possible choice, which completes the proof of the first claim. The proof of the second claim is similar. Then using Lemma B.15 gives the final concentration theorem. $\square$

$\square$

**Lemma C.4** (Azar et al. [2013], Lemma 7). *This is an adaptation of Azar et al. [2013] to RMDPs. For any policy $\pi$,*

$$\left\| (I - \gamma P^\pi_0)^{-1} \sqrt{\mathrm{Var}_{P_0} (V^\pi)} \right\|_\infty \leq \sqrt{\frac{2}{(1 - \gamma)^3}},$$

*where $P_0$ is the nominal transition model of $M$.*

*Proof.* This proof is exactly the same for Robust and non robust MDPs, as it uses only standard computations such as the Jensen inequality and no robust form which are specific to this problem. The main difference is that we are doing the proof on the nominal of our robust set $P_0$, considering the regularized robust Bellman operator and associated regularized reward functions.

Azar et al. [2013] introduce the variance of the sum of discounted rewards starting at state-action $(s, a)$,

$$\Sigma^\pi(s, a) := \mathbb{E}[|\sum_{t \geq 0} \gamma^t R_0(s_t, a_t) - Q^\pi(s, a)|^2 | s_0 = s, a_0 = a],$$

and we defined the same variance for robust MDPs using robust rewards $r_{Q_{sa}^\pi}^{(s,a)}$ and $r_{Q_s^\pi}^s$ and using robust Q-function instead of classical Q-function in the definition of $\Sigma$. Then, in their Lemma 6 they show that, for any $\pi$:

$$\Sigma^\pi = \operatorname{Var}_{P_0}(V^\pi) + \gamma^2 P_0^\pi \Sigma^\pi,$$

which is, in fact, a Bellman equation for the variance. The proof is exactly the same for RMDPs considering our robust reward $r_{Q_{sa}^\pi}^{(s,a)}$ or $r_{Q_s^\pi}^s$ and not classical $R_0$. Note that this is thanks to the regularized form of robust RMDPs. Finally, Lemma C.4 is the same as their Lemma 7 considering robust rewards. This lemma is usually called the total variance lemma. This completes the proof. $\qquad \square$

**Lemma C.5.** *The following upper bound holds with probability $1 - \delta$:*

$$\left\| Q^{\widehat{\pi}} - \widehat{Q}^{\widehat{\pi}} \right\|_\infty < (C_N + C_\beta) \left\| Q^{\widehat{\pi}} - \hat{Q}^{\widehat{\pi}} \right\|_\infty + \gamma 4 \sqrt{\frac{L}{N(1-\gamma)^3}} + \frac{\gamma \Delta'_{\delta, N}}{1 - \gamma} + \frac{\gamma \epsilon_{\mathrm{opt}}}{1 - \gamma} \left( 2 + \sqrt{\frac{8L}{N}} \right) \qquad (60)$$

*with $C_N = \frac{\gamma}{1-\gamma} \sqrt{\frac{8L}{N}}$ and $C_\beta = \frac{2\gamma \beta |S|^{1/q}}{1 - \gamma}$.*

*Proof.*

$$\left\|Q^{\widehat{\pi}} - \widehat{Q}^{\widehat{\pi}}\right\|_\infty$$

$$\overset{(a)}{\leq} \gamma \left\|\left(I - \gamma P_0^{\widehat{\pi}}\right)^{-1}(P_0 - \widehat{P})\widehat{V}^{\widehat{\pi}}\right\|_\infty + \frac{2\gamma\beta|S|^{1/q}}{1-\gamma}\left\|Q^{\widehat{\pi}} - \widehat{Q}^{\widehat{\pi}}\right\|_\infty$$

$$\overset{(b)}{\leq} \gamma\left\|\left(I - \gamma P^{\widehat{\pi}}\right)^{-1}(P_0 - \widehat{P})\widehat{V}^\star\right\|_\infty + \gamma\left\|(I - \gamma P_0^\pi)^{-1}(P_0 - \widehat{P})\left(\widehat{V}^{\widehat{\pi}} - \widehat{V}^\star\right)\right\|_\infty + \frac{2\gamma\beta|S|^{1/q}}{1-\gamma}\left\|Q^{\widehat{\pi}} - \widehat{Q}^{\widehat{\pi}}\right\|_\infty$$

$$\overset{(c)}{\leq} \gamma\left\|\left(I - \gamma P_0^{\widehat{\pi}}\right)^{-1}(P_0 - \widehat{P})\widehat{V}^\star\right\|_\infty + \frac{2\gamma\epsilon_{\text{opt}}}{1-\gamma} + \frac{2\gamma\beta|S|^{1/q}}{1-\gamma}\left\|Q^{\widehat{\pi}} - \widehat{Q}^{\widehat{\pi}}\right\|_\infty$$

$$\overset{(d)}{\leq} \gamma\left\|\left(I - \gamma P_0^{\widehat{\pi}}\right)^{-1}\left|(P_0 - \widehat{P})\widehat{V}^\star\right|\right\|_\infty + \frac{2\gamma\epsilon_{\text{opt}}}{1-\gamma} + \frac{2\gamma\beta|S|^{1/q}}{1-\gamma}\left\|Q^{\widehat{\pi}} - \widehat{Q}^{\widehat{\pi}}\right\|_\infty$$

$$\overset{(e)}{\leq} \gamma\sqrt{\frac{8L}{N}}\left\|\left(I - \gamma P_0^{\widehat{\pi}}\right)^{-1}\sqrt{\mathrm{Var}_{P_0}\left(\widehat{V}^\star\right)}\right\|_\infty + 2\frac{\gamma\Delta'_{\delta,N}}{1-\gamma} + \frac{2\gamma\epsilon_{\text{opt}}}{1-\gamma} + \frac{2\gamma\beta|S|^{1/q}}{1-\gamma}\left\|Q^{\widehat{\pi}} - \widehat{Q}^{\widehat{\pi}}\right\|_\infty$$

$$\overset{(f)}{\leq} \gamma\sqrt{\frac{8L}{N}}\left\|\left(I - \gamma P_0^{\widehat{\pi}}\right)^{-1}\left(\sqrt{\mathrm{Var}_{P_0}\left(V^{\widehat{\pi}}\right)} + \sqrt{\mathrm{Var}_{P_0}\left(V^{\widehat{\pi}} - \widehat{V}^{\widehat{\pi}}\right)} + \sqrt{\mathrm{Var}_{P_0}\left(\widehat{V}^{\widehat{\pi}} - \widehat{V}^\star\right)}\right)\right\|_\infty$$

$$+ \frac{\gamma\Delta'_{\delta,N}}{1-\gamma} + \frac{2\gamma\epsilon_{\text{opt}}}{1-\gamma} + \frac{2\gamma\beta}{1-\gamma}\left\|Q^{\widehat{\pi}} - \widehat{Q}^{\widehat{\pi}}\right\|_\infty$$

$$\overset{(g)}{\leq} \gamma\sqrt{\frac{8L}{N}}\left(\sqrt{\frac{2}{(1-\gamma)^3}} + \frac{\sqrt{\left\|V^{\widehat{\pi}} - \widehat{V}^{\widehat{\pi}}\right\|_\infty^2}}{1-\gamma} + \frac{2\epsilon_{\text{opt}}}{1-\gamma}\right) + \frac{\gamma\Delta'_{\delta,N}}{1-\gamma} + \frac{2\gamma\epsilon_{\text{opt}}}{1-\gamma} + \frac{2\gamma\beta}{1-\gamma}\left\|Q^{\widehat{\pi}} - \widehat{Q}^{\widehat{\pi}}\right\|_\infty$$

$$\overset{(h)}{\leq} \gamma\sqrt{\frac{8L}{N}}\left(\sqrt{\frac{2}{(1-\gamma)^3}} + \frac{\left\|Q^{\widehat{\pi}} - \widehat{Q}^{\widehat{\pi}}\right\|_\infty}{1-\gamma} + \frac{2\epsilon_{\text{opt}}}{1-\gamma}\right) + \frac{\gamma\Delta'_{\delta,N}}{1-\gamma} + \frac{2\gamma\epsilon_{\text{opt}}}{1-\gamma} + \frac{2\gamma\beta|S|^{1/q}}{1-\gamma}\left\|Q^{\widehat{\pi}} - \widehat{Q}^{\widehat{\pi}}\right\|_\infty$$

$$= \gamma\sqrt{\frac{8L}{N}}\left(\sqrt{\frac{2}{(1-\gamma)^3}} + \frac{\left\|Q^{\widehat{\pi}} - \widehat{Q}^{\widehat{\pi}}\right\|_\infty}{1-\gamma}\right) + \frac{\gamma\Delta'_{\delta,N}}{1-\gamma} + \frac{\gamma\epsilon_{\text{opt}}}{1-\gamma}\left(2 + \sqrt{\frac{8L}{N}}\right) + \frac{2\gamma\beta|S|^{1/q}}{1-\gamma}\left\|Q^{\widehat{\pi}} - \widehat{Q}^{\widehat{\pi}}\right\|_\infty$$

$$= (C_N + C_\beta)\left\|Q^{\widehat{\pi}} - \widehat{Q}^{\widehat{\pi}}\right\|_\infty + 4\gamma\sqrt{\frac{L}{N(1-\gamma)^3}} + \frac{\gamma\Delta'_{\delta,N}}{1-\gamma} + \frac{\gamma\epsilon_{\text{opt}}}{1-\gamma}\left(2 + \sqrt{\frac{8L}{N}}\right)$$

with $C_N = \frac{\gamma}{1-\gamma}\sqrt{\frac{8L}{N}}$ and $C_\beta = \frac{2\gamma\beta|S|^{1/q}}{1-\gamma}$.

We have that (a) is true by Lemma C.2, (b) is by the triangular inequality using $\widehat{V}^{\widehat{\pi}} = \widehat{V}^{\widehat{\pi}} + \widehat{V}^\star - \widehat{V}^\star$, (c) is from the definition of $\epsilon_{\text{opt}}$ and Eq. (4), (d) is by positivity of the classic horizon inverse matrix, that is $(I - \gamma P)^{-1} = \sum_{t>0}\gamma^t P^t > 0$, (e) is by Lemma C.3, (f) is by the triangular inequality for the variance (which is, in fact, a seminorm) and decomposing $\widehat{V}^\star = \widehat{V}^\star + \widehat{V}^{\widehat{\pi}} - \widehat{V}^{\widehat{\pi}} + V^{\widehat{\pi}} - V^{\widehat{\pi}}$, (g) is by Lemma C.4, uses the definition of $\epsilon_{\text{opt}}$ and takes the sup over $(s, a)$ of the variance in the second term, and eventually (h) is because we have that $\|V^\pi - \widehat{V}^\pi\|_\infty \leq \|Q^\pi - \widehat{Q}^\pi\|_\infty$ for any $\pi$.

$\square$

**Lemma C.6.** *The following upper bound holds with probability $1 - \delta$:*

$$\left\|Q^* - \widehat{Q}^{\pi^*}\right\|_\infty < (C_N + C_\beta)\left\|Q^* - \widehat{Q}^{\pi^*}\right\|_\infty + \gamma 4\sqrt{\frac{L}{N(1-\gamma)^3}} + \frac{\gamma\Delta'_{\delta,N}}{1-\gamma}. \tag{61}$$

*with $C_N = \frac{\gamma}{1-\gamma}\sqrt{\frac{8L}{N}}$ and $C_\beta = \frac{2\gamma\beta|S|^{1/q}}{1-\gamma}$.*

*Proof.*

$$
\left\|Q^* - \hat{Q}^{\pi^*}\right\|_\infty \overset{(a)}{\le} \gamma \left\|\left(I - \gamma P_0^{\pi^*}\right)^{-1}(P_0 - \widehat{P})\hat{V}^{\pi^*}\right\|_\infty + \frac{2\gamma\beta|S|^{1/q}}{1-\gamma}\left\|Q^* - \hat{Q}^{\pi^*}\right\|_\infty
$$

$$
\overset{(b)}{\le} \gamma \left\|\left(I - \gamma P_0^{\pi^*}\right)^{-1}\left|(P_0 - \widehat{P})\hat{V}^{\pi^*}\right|\right\|_\infty + \frac{2\gamma\beta|S|^{1/q}}{1-\gamma}\left\|Q^* - \hat{Q}^{\pi^*}\right\|_\infty
$$

$$
\overset{(c)}{\le} \gamma\sqrt{\frac{8L}{N}}\left\|\left(I - \gamma P_0^{\pi^*}\right)^{-1}\sqrt{\mathrm{Var}_{P_0}\left(\hat{V}^{\pi^*}\right)}\right\|_\infty + 2\frac{\gamma\Delta'_{\delta,N}}{1-\gamma} + \frac{2\gamma\beta|S|^{1/q}}{1-\gamma}\left\|Q^* - \hat{Q}^{\pi^*}\right\|_\infty
$$

$$
\overset{(d)}{\le} \gamma\sqrt{\frac{8L}{N}}\left\|\left(I - \gamma P_0^{\pi^*}\right)^{-1}\left(\sqrt{\mathrm{Var}_{P_0}(V^*)} + \sqrt{\mathrm{Var}_{P_0}\left(V^* - \hat{V}^{\pi^*}\right)}\right)\right\|_\infty
$$
$$
+ \frac{\gamma\Delta'_{\delta,N}}{1-\gamma} + \frac{2\gamma\beta|S|^{1/q}}{1-\gamma}\left\|Q^* - \hat{Q}^{\pi^*}\right\|_\infty
$$

$$
\overset{(e)}{\le} \gamma\sqrt{\frac{8L}{N}}\left(\sqrt{\frac{2}{(1-\gamma)^3}} + \frac{\sqrt{\left\|V^* - \hat{V}^{\pi^*}\right\|_\infty^2}}{1-\gamma}\right) + \frac{\gamma\Delta'_{\delta,N}}{1-\gamma} + \frac{2\gamma\beta|S|^{1/q}}{1-\gamma}\left\|Q^* - \hat{Q}^{\pi^*}\right\|_\infty
$$

$$
\le \gamma\sqrt{\frac{8L}{N}}\left(\sqrt{\frac{2}{(1-\gamma)^3}} + \frac{\left\|Q^* - \hat{Q}^{\pi^*}\right\|_\infty}{1-\gamma}\right) + \frac{\gamma\Delta'_{\delta,N}}{1-\gamma} + \frac{2\gamma\beta|S|^{1/q}}{1-\gamma}\left\|Q^* - \hat{Q}^{\pi^*}\right\|_\infty
$$

$$
= (C_N + C_\beta)\left\|Q^* - \hat{Q}^{\pi^*}\right\|_\infty + 4\gamma\sqrt{\frac{L}{N(1-\gamma)^3}} + \frac{\gamma\Delta'_{\delta,N}}{1-\gamma}
$$

with $C_N = \frac{\gamma}{1-\gamma}\sqrt{\frac{8L}{N}}$ and $C_\beta = \frac{2\gamma\beta|S|^{1/q}}{1-\gamma}$.

We have that (a) is true by Lemma C.2, (b) is by the positivity of the classic horizon inverse matrix, (c) is by Lemma (C.3), (d) is by the triangular inequality for the variance (which is a seminorm), (e) is by Lemma C.4 and taking the sup over $(s,a)$ of the variance in the second term, and eventually (h) is because $\|V^\pi - \hat{V}^\pi\|_\infty \le \|Q^\pi - \hat{Q}^\pi\|_\infty$ for any $\pi$.

$\square$

As the event on which $\Delta'_{\delta,N}$ is the same in the two previous Lemma C.5 and Lemma C.6, we can obtain the following.

**Theorem C.7.** *For $0 < C_\beta \le 1/2$ and $0 < C_N + C_\beta < 1$, with probability $1 - \delta$, we get:*

$$
\|Q^* - Q^{\hat{\pi}}\|_\infty < \frac{1}{1 - (C_N + C_\beta)}\left(8\gamma\sqrt{\frac{L}{N(1-\gamma)^3}} + \frac{2\gamma\Delta'_{\delta,N}}{1-\gamma} + \frac{\gamma\epsilon_{\mathrm{opt}}}{1-\gamma}\left(2 + \sqrt{\frac{8L}{N}}\right)\right) + \epsilon_{\mathrm{opt}}.
$$

*Proof.* This result is obtained by combining the two previous Lemmas C.5 and C.6 and passing the term in $(C_N + C_\beta)$ to the left-hand side. $\square$

Note that $C_\beta + C_N < 1$ implies $C_\beta = \frac{2\gamma\beta|S|^{1/q}}{1-\gamma} < 1$ and hence $\beta < \frac{1-\gamma}{2\gamma|S|^{1/q}}$. Now we need to pick $C_N < 1 - C_\beta$. Let $C_N \le 1 - C_\beta - \eta$, for any $0 < \eta < 1 - C_\beta$ the previous inequality becomes

$$
\|Q^* - Q^{\hat{\pi}}\|_\infty < \frac{8}{\eta}\gamma\sqrt{\frac{L}{N(1-\gamma)^3}} + \frac{2\gamma\Delta'_{\delta,N}}{\eta(1-\gamma)} + \frac{\gamma\epsilon_{\mathrm{opt}}}{\eta(1-\gamma)}\left(2 + \sqrt{\frac{8L}{N}}\right) + \epsilon_{\mathrm{opt}}.
$$

As $\Delta'_{\delta,N} = \sqrt{\frac{cL}{N}} + \frac{cL}{(1-\gamma)N}$, the term in $1/\sqrt{N}$ is given by $\frac{8\gamma\sqrt{L}H^{3/2}}{\eta\sqrt{N}}\left(1 + 1/4\sqrt{c/H}\right)$ and is smaller than $\epsilon$ whenever

$$
N \ge \frac{64\gamma^2 LH^3(1 + 1/4\sqrt{c/H})^2}{\eta^2\epsilon^2}.
$$

We will use $c < 16$ and $H \geq 1$ and use the stronger constraint

$$N \geq \frac{256\gamma^2 LH^3}{\eta^2 \epsilon^2}.$$

Along the same line, the term in $1/N$ is $\frac{2\gamma cLH^2}{\eta N}$ which is smaller than $\epsilon$ whenever

$$N \geq \frac{2\gamma cLH^2}{\epsilon}.$$

Now, $C_N < 1 - \eta - C_\beta$ means

$$\frac{\gamma}{1-\gamma}\sqrt{\frac{8L}{N}} < 1 - \eta - C_\beta$$

hence

$$N > \frac{8L\gamma^2 H^2}{(1 - \eta - C_\beta)^2}.$$

We deduce that whenever

$$N \geq \max\left(\frac{256\gamma^2 LH^3}{\eta^2 \epsilon^2}, \frac{2\gamma cLH^2}{\epsilon}, \frac{8L\gamma^2 H^2}{(1 - \eta - C_\beta)^2}\right)$$

$$= \frac{256\gamma^2 LH^3}{\eta^2} \max\left(\frac{1}{\epsilon^2}, \frac{c\eta}{128H\gamma\epsilon}, \frac{\eta^2}{64H(1 - \eta - C_\beta)^2}\right)$$

the error is smaller than $2\epsilon$ up to the $\epsilon_{\text{opt}}$ terms.

This bounds reduces to

$$N \geq \frac{C\gamma^2 LH^3}{\epsilon^2}$$

with $C = 256/\eta^2$ if

$$\epsilon \leq \min\left(\frac{128H}{\eta}, \sqrt{64H}\frac{1 - \eta - C_\beta}{\eta}\right).$$

Note that $\epsilon \in [0, H)$ and $\eta < 1$ so that the previous condition simplifies to

$$\epsilon \leq \sqrt{64H}\frac{1 - \eta - C_\beta}{\eta} = \epsilon_0.$$

If we want to obtain an arbitrary $\epsilon_0$, it suffices thus to take $\eta$ arbitrarily small leading to the constant $C = 256/\eta^2$ to be arbitrarily large.

Note that if $\epsilon_0 \geq O(H^{1/2+\delta})$ then $1/\eta > O(H^\delta)$ which adds a $H^{2\delta}$ factor to the bound on $N$.

However, for any $\kappa\sqrt{H}$ and for any $C_\beta$, it exists an $\eta$ independent of $H$ so that $\epsilon_0 = 8\sqrt{H}\frac{1-\eta-C_\beta}{\eta} = \kappa\sqrt{H}$, hence the result stated in Theorem 5.1.

Now, as $L = \log(8|\mathcal{S}||\mathcal{A}|/((1 - \gamma)\delta))$, the previous condition can be summarized by

$$N_{\text{total}} = N|\mathcal{S}||\mathcal{A}| = \tilde{O}\left(\frac{H^3|\mathcal{S}||\mathcal{A}|}{\epsilon^2}\right)$$

provided $\epsilon < \epsilon_0$.

Finally, taking $\beta_0 = \frac{1-\gamma}{8\gamma}$ which gives $C_\beta = 1/4$ and $\eta = 1/2$ so that $C_N \leq 1/4$, we obtain $C = 1024$ and $\epsilon_0 = \sqrt{16H}$.