# OpenReview forum: "Towards Minimax Optimality of Model-based Robust Reinforcement Learning"
_auai.org/UAI/2024/Conference — UAI 2024 oral_

### Official Review · Reviewer_2K6m · 2024-02-29

**Q2-1 Originality-Novelty:** 4
**Q2-2 Correctness-Technical Quality:** 3
**Q2-5 Clarity Of Writing:** 3

**Q1 Summary And Contributions:**

This work presents the theoretical analysis on the Model-based Robust Reinforcement Learning. It provides the minimax optimality sample complexity of such algorithm for the $L_p$-ball uncertainty set. Then the proposed algorithm DRVI $L_p$ achieves such bound. When the uncertainty set is sufficiently small, the result recovers the sample complexity of the non-robust case.

**Q2-3 Extent To Which Claims Are Supported By Evidence:**

3: Good: the main claims are supported by convincing evidence (in the form of adequate experimental evaluation, proofs, (pseudo-)code, references, assumptions).

**Q2-4 Reproducibility:**

3: Good: key resources (e.g. proofs, code, data) are available and key details (e.g. proofs, experimental setup) are sufficiently well-described for competent researchers to confidently reproduce the main results.

**Q3 Main Strengths:**

This work presents a solid theoretical understanding of Model-based Robust Reinforcement Learning and explicitly measures how much the uncertainty will affect the sample complexity (by a factor $H$ as indicated by this paper). This result is new and has not been shown by others.

Regarding the presentation, this work is sufficiently complete; it provides the best possible complexity and presents the algorithm that achieves such complexity. And each step in the proof is well-explained.

**Q4 Main Weakness:**

This work considers the $L_p$-ball, which generalizes the total variation distance. I have concerns about this setting: (1) Many work will consider the KL-divergence instead of $L_p$-ball, so the $L_p$-ball is not sufficiently general to include these settings; (2) $L_p$-ball is simpler than other divergences since a recent result [Kumar2023] indicates that the worst-kernel is a rank-one perturbation of the nominal; other uncertainty set may not have such simplified structure.

[Kumar2023] Kumar, Navdeep, et al. "Policy Gradient for Rectangular Robust Markov Decision Processes." Advances in Neural Information Processing Systems 36 (2024).

These concerns make me feel the result of this paper might be somehow restricted.

**Q5 Detailed Comments To The Authors:**

1. I suggest the author include [Kumar2023] in the main context since it also considers the $L_p$-ball setting.
2. As explained in the Q4 section, I would like to hear the author further clarify the motivation for studying $L_p$-ball instead of more common KL-divergence. From my perspective, $L_p$-ball is simpler but less representative.

**Q9 Complying With Reviewing Instructions:**

Yes

---

> ### Author Rebuttal · Authors · 2024-04-03
>
> We thank the reviewer for their comments and provide answers to their questions and remarks.
>
> *Q4 Main Weakness:
> This work considers the $L_p$-ball, which generalizes the total variation distance. I have concerns about this setting: (1) Many works will consider the KL-divergence instead of $L_p$-ball, so the $L_p$-ball is not sufficiently general to include these settings; and
> 2. As explained in the Q4 section, I would like to hear the author further clarify the motivation for studying $L_p$-ball instead of more common KL-divergence. From my perspective, $L_p$-ball is simpler but less representative.*
>
> We are studying $L_p$ balls because they are a generalization of TV and because $L_p$ norms are finite quantities compared to divergences, which can go to infinity when the support of the adversarial is different from the nominal. It is acknowledged that studying KL would be of interest, however, this is left for future work.
>
> *$L_p$-ball is simpler than other divergences since a recent result [Kumar2023] indicates that the worst-kernel is a rank-one perturbation of the nominal; other uncertainty set may not have such simplified structure.
> [Kumar2023] Kumar, Navdeep, et al. "Policy Gradient for Rectangular Robust Markov Decision Processes." Advances in Neural Information Processing Systems 36 (2024). I suggest the author include [Kumar2023] in the main context since it also considers the $L_p$-ball setting.*
>
> We will include this paper as it deals with $L_p$ RMDPs, however, their setting did not assume adversarial to belong to the simplex, which leads to simple rank one perturbation, which is not the case for robust transitions belonging to the simplex.

---

### Official Review · Reviewer_JVFM · 2024-03-21

**Q2-1 Originality-Novelty:** 2
**Q2-2 Correctness-Technical Quality:** 3
**Q2-5 Clarity Of Writing:** 3

**Q1 Summary And Contributions:**

This paper studies the sample complexity of learning an $\epsilon$-optimal policy in robust MDPs with a generative model. In particular, this paper considers robust MDPs with sa- and s- rectangular uncertainty sets with $L_p$-balls and studies the model-based algorithm. In the general case, this paper proves a sample complexity of $\frac{H^4|S||A|}{\epsilon^2}$, which leads to improvements of $|S|$ and $|A|$. When the radius of the uncertainty set is small enough, this paper proves an improved sample complexity of $\frac{H^3|S||A|}{\epsilon^2}$, which matches the lower bound in the non-robust case for the first time.

**Q2-3 Extent To Which Claims Are Supported By Evidence:**

3: Good: the main claims are supported by convincing evidence (in the form of adequate experimental evaluation, proofs, (pseudo-)code, references, assumptions).

**Q2-4 Reproducibility:**

3: Good: key resources (e.g. proofs, code, data) are available and key details (e.g. proofs, experimental setup) are sufficiently well-described for competent researchers to confidently reproduce the main results.

**Q3 Main Strengths:**

1. This work takes a notable step towards sharper analysis for robust MDPs. This paper achieves improved sample complexities of learning an $\epsilon$-optimal policy in robust MDPs with a generative model. Importantly, this paper is the first work that achieves the minimax sample complexity in solving robust MDPs, thus providing a complete understanding of the sample complexity of solving robust MDPs.
2. This paper is overall well-written and provides a clear summary of its main contributions. It is easy for me to follow this paper.

**Q4 Main Weakness:**

The analysis methods used in this paper seem to be not very novel.  The main analysis techniques used in this paper have been developed in previous works of Agarwal et al. [2020] and Azar et al. [2013].

**Q5 Detailed Comments To The Authors:**

1. In the definition of $\mathcal{P}\_\{s, a}$ in Assumption 3.1, $P_0$ should be $P\_\{0, s, a\}$ which represents the transition probability at the pair $(s, a)$? Similarly, in Assumption 3.2, $P_0$ should be $P_{0, s}$?
2. In the first line of page 6, $\epsilon_{ops}$ should be $\epsilon_{opt}$.

**Q9 Complying With Reviewing Instructions:**

Yes

---

> ### Author Rebuttal · Authors · 2024-04-03
>
> We would like to thank the reviewer for all his comments and provide some responses to his questions and concerns.
>
> *Q4 Main Weakness:
> The analysis methods used in this paper seem to be not very novel. The main analysis techniques used in this paper have been developed in previous works of Agarwal et al. [2020] and Azar et al. [2013].*
>
> We agree that a more technical discussion would be helpful to point novel contributions. We can derive better bounds with two significant contributions which are:
>
> 1) Bernstein's arguments and leave-one-out absorbing MDP ideas from  Agarwal et al. [2020] can be transferred to RMPDs (with some work, but this not straightforward at all) to reduce from $H^4$ to $H^3$ the sample complexity.
>
> 2) We have derived a new optimization lemma for rectangular $L_P$ RMDPS with sa and s. (B5 and B6) These lemma aims to improve the optimization process and achieve better results as Penalization terms in the dual for B5 and B6 do not depend on the current transition matrix as it only spans semi-norms. This allows vanishing of these terms in lemma B7. This is central in our proof before using concentration theory.
>
> *Q5 Detailed Comments To The Authors
> 1 : In the definition of $P_{s, a}$ in Assumption 3.1, $P_0$ should be $P_{0, s, a}$ which represents the transition probability at the pair $(s, a)$ ? Similarly, in Assumption 3.2, $P_0$ should be $P_{0, s}$ ?
> 2 : In the first line of page $6, \epsilon_{o p s}$ should be $\epsilon_{\text {opt }}$.*
>
> We thank the reviewer for pointing out these typos, which will be corrected in the final version.

---

### Official Review · Reviewer_SSdQ · 2024-03-25

**Q2-1 Originality-Novelty:** 3
**Q2-2 Correctness-Technical Quality:** 2
**Q2-5 Clarity Of Writing:** 3

**Q1 Summary And Contributions:**

This work studies model-based RL for robust MDP with a generative model. Similar to vanilla MDPs, the authors proposed algorithms based on the plug-in approach: first estimating the transition model, then selecting the optimal policy w.r.t. the estimated model as its policy output. The main contribution of this work is an improvement of the sample complexity of finding the optimal complexity, as it does not depend on the robustness parameter $\beta$ and the total sample complexity achieves the minimax lower bound when $\beta$ is small. The main technical contribution is the use of the dual form of the robust Bellman operator (Lemma 4.2), which is of independent interest.

**Q2-3 Extent To Which Claims Are Supported By Evidence:**

3: Good: the main claims are supported by convincing evidence (in the form of adequate experimental evaluation, proofs, (pseudo-)code, references, assumptions).

**Q2-4 Reproducibility:**

3: Good: key resources (e.g. proofs, code, data) are available and key details (e.g. proofs, experimental setup) are sufficiently well-described for competent researchers to confidently reproduce the main results.

**Q3 Main Strengths:**

The result itself is important, as previous works mainly focus on specific robust algorithms that find the optimal policies. Instead, this work uses a plug-in approach that makes it viable for any near-optimal planner.

The proof looks technically sound to me, although I did not check every detail.

**Q4 Main Weakness:**

There are no numerical experiments. I suggest the authors add some basic experiments to suggest that the plug-in approach is as good as the robust VI approach-based algorithms. Experiments on some small-scale MDPs are fine.

**Q5 Detailed Comments To The Authors:**

See Q3 and Q4. Besides,

I assume that the sample complexity should still depend on $\beta$, otherwise it seems quite strange since a very large $\beta$ may even ruin the original MDP structure. Is it possible to provide a sample complexity with a dependence on $\beta$?

**Q9 Complying With Reviewing Instructions:**

Yes

---

> ### Author Rebuttal · Authors · 2024-04-03
>
> We thank the reviewer for their comments and provide answers to their questions and remarks.
>
> *Q4 Main Weakness:
> There are no numerical experiments. I suggest the authors add some basic experiments to suggest that the plug-in approach is as good as the robust VI approach-based algorithms. Experiments on some small-scale MDPs are fine.*
>
> We agree with the reviewer that we could illustrate our result using sample finite MDPs to understand empirically our results. It is being considered to conduct an experiment in the final version of the paper.
>
> *Q5 Detailed Comments To The Authors:
> See Q3 and Q4. Besides,
> I assume that the sample complexity should still depend on $\beta$, otherwise it seems quite strange since a very large $\beta$ may even ruin the original MDP structure. Is it possible to provide a sample complexity with a dependence on $\beta$ ?*
>
> This is an interesting question, and we are currently working on it. As you mentioned, when $\beta$ increases, it leads to a very conservative solution or constant value functions (as every kernel in the simplex can be chosen) and the sample complexity should maybe be lower. This point will be discussed in the final version of the manuscript.

---

### Official Review · Reviewer_Wmke · 2024-03-26

**Q2-1 Originality-Novelty:** 2
**Q2-2 Correctness-Technical Quality:** 3
**Q2-5 Clarity Of Writing:** 2

**Q1 Summary And Contributions:**

The paper provides a new sample complexity bound for robust MDPs that in the general case improves the state-of-the-art by a factor of $SA$ and $S$ respectively for $sa$ and $s$ uncertainty sets. The authors improve it even further in a special case and achieve the optimal lower bound.

**Q2-3 Extent To Which Claims Are Supported By Evidence:**

3: Good: the main claims are supported by convincing evidence (in the form of adequate experimental evaluation, proofs, (pseudo-)code, references, assumptions).

**Q2-4 Reproducibility:**

3: Good: key resources (e.g. proofs, code, data) are available and key details (e.g. proofs, experimental setup) are sufficiently well-described for competent researchers to confidently reproduce the main results.

**Q3 Main Strengths:**

1. New state-of-the-art.
2. Theoretical results are sound.

**Q4 Main Weakness:**

1. the writing is poor and hard to follow in many places.
2. It seems that the algorithm is pretty standard, and improvement is achieved mainly by new insights in the analysis. Although the authors have discussed in the introduction why the new analysis improves the state-of-the-art, the discussion is superficial. A more technical discussion would be helpful. For example, while discussing the proof sketch of Theorem 4.1, it would be helpful to point out why the analysis is difficult despite most of the techniques being pre-existing in the literature.

**Q5 Detailed Comments To The Authors:**

1. Notations are hard to follow. The presence of multiple typos makes it even worse.

2. The definition of robust Bellman operator seems to have a typo. Specifically, the LHS in the relevant equation should be $(\mathcal{T}^{\pi}_{\mathcal{U}}V)(s)$.

3. The sentence “…where $P$ and $R$ belong to the uncertainty set, $\mathcal{U}$'' is not clear. Is $\mathcal{U}$ the uncertainty set for the pair $(P, R)$?

4. The meaning of the assertion $\mathcal{U}\in\Delta_{s}$ (page 4) is not clear. Is there any typo? A slightly modified assertion $\mathcal{U}\subset\Delta_{S}$ also does not seem to be mathematically correct if $\mathcal{U}$ is the uncertainty set of the pair $(P, R)$.

5. In the definition of $\mathcal{P}_{s, a}$ (Assumption 3.1), one should use $P_0(s, a)$ instead of $P_0$. A similar correction applies to the definition of $\mathcal{P}_s$ (Assumption 3.2).

6. Definition 3.1 and its subsequent discussion are incomprehensible. It is not clear where the function $\kappa_q$ is used. Is the domain of $\omega_q:\mathcal{S}\rightarrow \mathbb{R}$ consistent with its definition?

7. In theorem 4.1, it is unclear how the factors $\alpha$ and $\beta$ influence the final sample complexity. If they do not affect the result (which seems to be the case), would the result still hold if $\alpha$ and $\beta$ are infinite?

**Q9 Complying With Reviewing Instructions:**

Yes

---

> ### Author Rebuttal · Authors · 2024-04-03
>
> We would like to thank the reviewer for carefully reading our paper and providing relevant comments. We are providing details in response to his feedback:
>
> *It seems that the algorithm is pretty standard, and improvement is achieved mainly by new insights in the analysis. Although the authors have discussed in the introduction why the new analysis improves the state-of-the-art, the discussion is superficial. A more technical discussion would be helpful. For example, while discussing the proof sketch of Theorem 4.1, it would be helpful to point out why the analysis is difficult despite most of the techniques being preexisting in the literature.*
>
> We agree that a more technical discussion would be helpful, especially why we can derive better bounds with two points :
>
> 1)Bernstein's arguments and leave-one-out absorbing MDP ideas from  Agarwal et al. [2020] can be transferred to RMPDs (with some work, but this not straightforward at all) to reduce from $H^4$ to $H^3$ the sample complexity.
>
> 2 We have derived a new optimization lemma for rectangular $L_P$ RMDPS with sa and s. (B5 and B6) These lemma aims to improve the optimization process and achieve better results as Penalization terms in the dual for B5 and B6 do not depend on the current transition matrix as it only spans semi-norms. This allows vanishing of these terms in lemma B7. This is central in our proof before using concentration theory.
>
> These two points significantly improve the sample complexity of RMDPs.
>
>
>
> Thanks for pointing out these typos and carefully reading the proofs.
>
> *1) Notations are hard to follow. The presence of multiple typos makes it even worse.
> 2) The definition of robust Bellman operator seems to have a typo. Specifically, the LHS in the relevant equation should be ...
> 3) The sentence "...where $P$ and $R$ belong to the uncertainty set, $\mathcal{U}$ " is not clear. Is $\mathcal{U}$ the uncertainty set for the pair $(P, R)$ ?
> 4) The meaning of the assertion $\mathcal{U} \in \Delta_s$ (page 4) is not clear. Is there any typo? A slightly modified assertion $\mathcal{U} \subset \Delta_S$ also does not seem to be mathematically correct if $\mathcal{U}$ is the uncertainty set of the pair $(P, R)$.
> 5) In the definition of $\mathcal{P}_{s, a}$ (Assumption 3.1), one should use $P_0(s, a)$ instead of $P_0$. A similar correction applies to the definition of $\mathcal{P}_s$ (Assumption 3.2).*
>
>
> Thanks for point these typo and carefully reading proof  This typographical error will be corrected in the final version of the manuscript.
>
>
> *6. Definition 3.1 and its subsequent discussion are incomprehensible. It is not clear where the function $\kappa_q$ is used. Is the domain of $\omega_q: \mathcal{S} \rightarrow \mathbb{R}$ consistent with its definition?*
>
> This typo will be corrected in the final version of the manuscript as q-variance is equivalent to span semi-norm and is denoted $sp_q(v)$.
>
> *7. In theorem 4.1, it is unclear how the factors $\alpha$ and $\beta$ influence the final sample complexity. If they do not affect the result (which seems to be the case), would the result still hold if $\alpha$ and $\beta$ are infinite?*
>
> Thank you for pointing this out, in fact the sample complexity does not depend on $\alpha$ as we can change the definition of the reward (or robustify the MDPs to reward mispecifications) without changing the sample complexity. However, it depends on $\beta$ for many f-divergences such as $\chi^2$. [1] In fact, our results also depend on $\beta$. For the regime where beta is less than or equal to $\beta_0$, we observe a smaller sample complexity.
>
> [1] The Curious Price of Distributional Robustness in Reinforcement Learning with a Generative Model
> Laixi Shi, Gen Li, Yuting Wei, Yuxin Chen, Matthieu Geist, Yuejie Chi

---

### Official Review · Reviewer_yH4i · 2024-03-26

**Q2-1 Originality-Novelty:** 3
**Q2-2 Correctness-Technical Quality:** 4
**Q2-5 Clarity Of Writing:** 3

**Q1 Summary And Contributions:**

This paper studies the sample complexity of obtaining an $\varepsilon$-optimal policy in RMDPs, with access to a sampling oracle for the nominal model and any planning algorithm with high-probability guarantees. The analysis establishes state-of-the-art sample complexity $\widetilde{\mathcal{O}}\bigl( \frac{H^4 |S| |A|}{\varepsilon^2} \bigr)$ in the general case, for both sa- and s-rectangular RMDPs. It is also shown that the sample complexity can be further improved to $\widetilde{\mathcal{O}}\bigl( \frac{H^3 |S| |A|}{\varepsilon^2} \bigr)$ when the ambiguity set is small enough, recovering the lower-bound for the non-robust case. Finally, the DRVI algorithm is proposed to solve the planning problem for $L_p$-constrained RMDPs, adding a missing piece to the complexity results.

**Q2-3 Extent To Which Claims Are Supported By Evidence:**

4: Excellent: all claims are supported by very convincing evidence (in the form of comprehensive experimental evaluation, rigorous mathematical proofs, detailed (pseudo-)code, precise references, well-motivated and realistic assumptions) and the authors deliver what they promise.

**Q2-4 Reproducibility:**

4: Excellent: key resources (e.g. proofs, code, data) are available and key details (e.g. proof sketches, experimental setup) are comprehensively described for competent researchers to confidently and easily reproduce the main results.

**Q3 Main Strengths:**

* This paper is well-motivated and fits into the literature and on-going research well.
* A comprehensive literature review is presented, with table comparing known sample complexities, which familiarizes the readers with the current progress and challenges in this direction.
* All theoretical results are concrete and rigorously written, presented with proof sketch, discussion and comparison against related works ( I appreciate this part a lot). The proofs in the appendix are detailed and not too hard to follow (though indeed very technical).
* The flow of the paper is friendly to the readers.

**Q4 Main Weakness:**

* Some of the notations are a little confusing.
  * The transition kernel $P$ is both understood as a function and a matrix, and element-wise operations are used in a few places (e.g., the definition of $\mathrm{Var}_P(V)$. I understand this is common in literature (e.g., Agarwal et al. (2019)), but it may still confuse the readers.
  * A section in the appendix to explain the use of notations is recommended. For example I had some difficulties in finding the definition for $[V]_{\alpha}$, which is "hidden" in Lemma A.7.
  * In Theorem 5.1, $C$ is not a universal constant, but rather, depends on $\varepsilon_0$ and $\beta_0$. I would suggest using $C(\beta_0, \varepsilon_0)$ to make the dependency clearer.
* There are a few typos in the paper. For example:
  * On page 7, right column, there is an unfinished sentence “it is still an open question”, and I'm not sure what is left as future work.
  * In Appendix A.2, Assumption A.1/A.3 and Theorem A.2/A.4 seem to be exactly the same, which obviously shouldn't be the case.

**Q5 Detailed Comments To The Authors:**

The paper is overall pleasing to read. The following is just some random thoughts that came into my mind while reading. It is standard to bypass the exploration issue with a generative model, but I still find it interesting to ask what would be a proper setting to talk about on-policy learning with exploration for RMDPs. Any intuitions or thoughts on future work would be appreciated.

**Q9 Complying With Reviewing Instructions:**

Yes

---

> ### Author Rebuttal · Authors · 2024-04-03
>
> We appreciate the reviewer's positive feedback and the time they spent reviewing our work, as well as their relevant comments. Here are responses to the comments :
>
> *Some of the notations are a little confusing.
> The transition kernel $P$ is both understood as a function and a matrix, and element-wise operations are used in a few places (e.g., the definition of $\operatorname{Var}_P(V)$. I understand this is common in literature (e.g., Agarwal et al. (2019)), but it may still confuse the readers.*
>
> We will clarify this definition by writing $Var_{P_{s,a}}(V)$ when it is just a scalar and  $Var_P(V)$ when it is a matrix.
>
> *A section in the appendix to explain the use of notations is recommended. For example I had some difficulties in finding the definition for $[V]_\alpha$, which is "hidden" in Lemma A.7.*
>
> The definition will be provided in the main paper and this point will be clarified.
>
> *In Theorem 5.1, $C$ is not a universal constant, but rather, depends on $\varepsilon_0$ and $\beta_0$. I would suggest using $C\left(\beta_0, \varepsilon_0\right)$ to make the dependency clearer.*
>
> It is true that the constant is dependent on these two parameters, therefore we will rename the constant.
>
> *There are a few typos in the paper. For example:
> On page 7, right column, there is an unfinished sentence "it is still an open question", and I'm not sure what is left as future work.
> We will correct this sentence, the meaning was that there is no current result for $L_p$ norms currently with sample complexity smaller than classical MDPs.
> In Appendix A.2, Assumption A.1/A. 3 and Theorem A.2/A. 4 seem to be exactly the same, which obviously shouldn't be the case.*
>
> This point will be corrected in the final version of the manuscript.
>
> *The paper is overall pleasing to read. The following is just some random thoughts that came into my mind while reading. It is standard to bypass the exploration issue with a generative model, but I still find it interesting to ask what would be a proper setting to talk about on-policy learning with exploration for RMDPs. Any intuitions or thoughts on future work would be appreciated.*
>
>   The combination of robust Markov decision processes (MDPs) and exploration problems presents an interesting challenge. Robust MDPs rely on a pessimistic approach, while exploration problems typically use optimism. It is not immediately clear how to effectively merge these two approaches. However, this presents an opportunity for further exploration and research.

---

### Meta-Review · Area_Chair_VBBg · 2024-04-14

This paper studies the near-optimal sample complexity of model-based robust MDPs, when a generative model is available. It reaches a consensus that the paper is well-written, contains novel technical contributions, and is solid in theory. Given the minimax nature of robust MDP, it would also be helpful for people to understand the technical challenge/novelty by comparing with the near-optimal sample complexity of model-based approaches in "zero-sum Markov games" (under the "generative model" setting), where the tightness in H can be obtained, see e.g., [1,2,3].

[1] Kaiqing Zhang, Sham M. Kakade, Tamer Basar, and Lin F. Yang. "Model-Based Multi-Agent RL in Zero-Sum Markov Games with Near-Optimal Sample Complexity." Journal of Machine Learning Research 24, no. 175 (2023): 1-53.
[2] Gen Li, Yuejie Chi, Yuting Wei, and Yuxin Chen. "Minimax-optimal multi-agent RL in Markov games with a generative model." Advances in Neural Information Processing Systems 35 (2022): 15353-15367.
[3] Yuling Yan, Gen Li, Yuxin Chen, and Jianqing Fan. "Model-Based Reinforcement Learning for Offline Zero-Sum Markov Games." Operations Research (2024).

Overall, this is a good paper, and I encourage the authors to incorporate the feedback from the reviewers in preparing the next version of the paper.